# The Crk4-Cyc4 complex regulates G$_2$/M transition in *Toxoplasma gondii*

Lauren M Hawkins[1], Chengqi Wang[2], Dale Chaput[3], Mrinalini Batra[1], Clem Marsilia [1], Danya Awshah[1] & Elena S Suvorova [1✉]

## Abstract

A versatile division of apicomplexan parasites and a dearth of conserved regulators have hindered the progress of apicomplexan cell cycle studies. While most apicomplexans divide in a multi-nuclear fashion, *Toxoplasma gondii* tachyzoites divide in the traditional binary mode. We previously identified five *Toxoplasma* CDK-related kinases (Crk). Here, we investigated TgCrk4 and its cyclin partner TgCyc4. We demonstrated that TgCrk4 regulates conventional G$_2$ phase processes, such as repression of chromosome rereplication and centrosome reduplication, and acts upstream of the spindle assembly checkpoint. The spatial TgCyc4 dynamics supported the TgCrk4–TgCyc4 complex role in the coordination of chromosome and centrosome cycles. We also identified a dominant TgCrk4–TgCyc4 complex interactor, TgiRD1 protein, related to DNA replication licensing factor CDT1 but played no role in licensing DNA replication in the G$_1$ phase. Our results showed that TgiRD1 also plays a role in controlling chromosome and centrosome reduplication. Global phosphoproteome analyses identified TgCrk4 substrates, including TgORC4, TgCdc20, TgGCP2, and TgPP2ACA. Importantly, the phylogenetic and structural studies suggest the Crk4–Cyc4 complex is limited to a minor group of the binary dividing apicomplexans.

**Keywords** Apicomplexa; *Toxoplasma gondii*; Cell Cycle; CDK; G$_2$/M Checkpoint

**Subject Categories** Cell Cycle; Microbiology, Virology & Host Pathogen Interaction

## Introduction

Apicomplexan parasites are opportunistic intracellular pathogens of humans and animals. They cause many important diseases, such as malaria, toxoplasmosis, and cryptosporidiosis. Apicomplexan cell divisions are remarkably versatile and vastly differ from the division mechanisms of their host cells (Francia et al, 2012; White and

Suvorova, 2018). Some species of apicomplexan parasites duplicate their genome once in a binary fashion, resulting in progenies of two, and others replicate their genomes multiple times, resulting in thousands of daughter cells in a single division round (Gubbels et al, 2020). Duplicated apicomplexan genomes can be segregated after each round of chromosome replication or after multiple rounds, which would include intermediate stages with multiple nuclei, or a DNA syncytium. Depending on the site of daughter bud assembly, parasites can form buds internally or from the surface of the mother cell. There are also species such as *Toxoplasma gondii* that can switch division modes in different hosts (Gubbels et al, 2021; Gubbels et al, 2022). To date, two types of apicomplexan cell division have mainly been studied: the most abundant multinuclear division called schizogony, and the rare binary division endodyogeny.

Cell division is regulated by a cell cycle program, which is poorly understood in apicomplexan parasites. There are several liabilities that prevent fast progress of these studies, such as the complexity of cell division modes, low conservation of cell cycle regulators, and compound parasite-specific internal mitotic structures. Nevertheless, major cell cycle phases and their associated processes have been identified. There are distinguishable growth (G$_1$), DNA replication (S), DNA segregation (M or mitosis) phases, and cytokinesis (budding) (Alvarez and Suvorova, 2017; Behnke et al, 2010; Francia and Striepen, 2014; Ganter et al, 2017; Gubbels et al, 2008; Hawkins et al, 2022; Matthews et al, 2018; Nishi et al, 2008; Wall et al, 2018; White and Suvorova, 2018). However, the Gap 2 (G$_2$) period that temporarily separates chromosome replication and segregation is presumed missing in apicomplexan cell cycles. To achieve higher progeny, apicomplexan parasites repress cytokinesis and allow chromosomes to re-replicate, which seems to be linked to the binary structure of apicomplexan centrosomes (Chen and Gubbels, 2019; Suvorova et al, 2015). Mitosis is closed and runs concurrently with the assembly of daughter cytoskeletons (cytokinesis or budding), but multinuclear divisions have additional mitotic mechanisms uncoupled from budding that operate during amplification of nuclear content (Francia and Striepen, 2014). Thus, the representative *Plasmodium* spp. engage both mitotic mechanisms to produce multiple daughters, while *T. gondii* tachyzoites involve only coupled mitosis and cytokinesis during endodyogeny (White and Suvorova, 2018).

[1]Division of Infectious Diseases, Department of Internal Medicine, Morsani College of Medicine, University of South Florida, Tampa, FL 33612, USA. [2]College of Public Health, University of South Florida, Tampa, FL 33612, USA. [3]Proteomics Core, College of Arts and Sciences, University of South Florida, Tampa, FL 33612, USA. ✉E-mail: suvorova@usf.edu

The complete sequence of apicomplexan cell cycle events has never been established and the regulation of cycle transitions remains unknown because only a few major players in the cell cycle pathways of conventional eukaryotes are identifiable in apicomplexan genomes (Matthews et al, 2018; White and Suvorova, 2018). Most *T. gondii* CDK-related kinases (Crks) and cyclins have limited similarity to conventional eukaryotic counterparts, but we have recently discovered that multiple Crks are required to progress through the *T. gondii* tachyzoite cell cycle (Alvarez and Suvorova, 2017). Mapping *T. gondii* Crk activities revealed several points of regulation, including conventional spindle assembly checkpoint (SAC) regulated by the novel TgCrk6-TgCyc1 complex (Hawkins et al, 2022). While $G_1$ kinase TgCrk2, S-phase kinase TgCrk5, and mitotic kinase TgCrk6 are conserved among apicomplexans, not all species encode TgCrk4 orthologs (Alvarez and Suvorova, 2017). Previous examination of TgCrk4 deficiency using a tet-OFF model suggested a role for TgCrk4 in centrosome duplication, but neither the mechanism nor the cell cycle stage it regulates was determined (Alvarez and Suvorova, 2017). We also could not detect a cyclin partner for TgCrk4.

In this study, we identified the cyclin TgCyc4 that specifically interacts with TgCrk4, and together they regulate the presumed missing $G_2$ phase of apicomplexan endodyogeny. We demonstrated that the primary TgCrk4 role is to suppress DNA rereplication and centrosome reduplication. We also found that TgCrk4–TgCyc4 complex interacts with TgiRD1 protein genotypically distantly related to eukaryotic DNA replication licensing factor CDT1, and phenotypically related to the inhibitor of replication Geminin. The limited conservation of TgCyc4 in the genomes of binary dividing parasites, and the specifics of TgiRD1 function suggest the dominant multinuclear division mode was targeted for repression in these apicomplexan subgroups.

## Results

### Coccidian CDK-related kinase TgCrk4 forms complex with atypical TgCyc4

Our analysis of CDK-related kinases in *T. gondii* demonstrated the role of the novel TgCrk4 kinase in dividing tachyzoites (Alvarez and Suvorova, 2017; White and Suvorova, 2018). We previously examined TgCrk4 in a tet-OFF model of conditional expression which sufficiently downregulated the kinase after 6–8 h. Gradual downregulation over a period of one division cycle (7 h) produced a tainted phenotype and resulted in a rough estimation of TgCrk4-dependent processes. To define processes regulated by TgCrk4, we have now engineered a model of acute proteolytic degradation of TgCrk4. The TgCrk4 auxin-induced degradation (AID) mutant was created by placing an AID-3xHA epitope at the 3'-end of the kinase genomic locus (Fig. EV1A,C). Consistent with previous findings, TgCrk4$^{AID-HA}$ kinase was low in abundance and Western blot analysis confirmed downregulation of TgCrk4$^{AID-HA}$ within a 10-min indole 3-acetic acid (IAA or auxin) treatment (Fig. 1A). Contrary to the TgCrk4 tet-OFF model, the robust proteolytic degradation of TgCrk4$^{AID-HA}$ resulted in complete growth arrest of the tachyzoites. The RH TgCrk4$^{AID-HA}$ transgenic tachyzoites grew normally in the absence of auxin but failed to form plaques in the presence of auxin (Fig. 1B).

Although TgCrk4 has a recognizable cyclin-binding domain, we previously could not detect a cyclin partner (Alvarez and Suvorova, 2017). Phylogenetic analyses of the newly annotated cyclin-domain-containing proteins revealed that *Toxoplasma* genomes encoded more cyclin-like factors than initially anticipated (Hawkins et al, 2022). To find out if TgCrk4 interacts with any of these novel cyclin-like proteins, we examined TgCrk4$^{AID-HA}$ protein complexes (Figs. 1E and EV1E). The mass-spectrometry analysis detected a single cyclin-domain-containing protein, TgCyc4 (TGME49_249880), and SAINT analysis confirmed a high probability of TgCrk4–TgCyc4 interaction (Fig. EV1G; Dataset EV3).

To test whether TgCyc4 is a major partner of the cell cycle kinase TgCrk4, we made several attempts to establish an AID conditional model for TgCyc4. Despite successful incorporation of the AID-3xHA nucleotide sequence at the 3'-end of the *TgCyc4* genomic locus, we could not detect TgCyc4$^{AID-HA}$ protein, suggesting proteolytic processing at the C-terminus. Corroborating this finding, we successfully placed 3xHA epitope at the N-terminus of TgCyc4 and under control of the tetracycline regulatable promoter within the *TgCyc4* genomic locus (Fig. EV1B,D). Western blot analysis detected two bands corresponding to the full-length (Fig. 1C, red star) and truncated TgCyc4, further corroborating TgCyc4 processing. This observation, together with the lack of protein expression in TgCyc4$^{AID-HA}$ with a C-terminus tag, suggests that the TgCyc4 C-terminus is highly unstable.

Testing TgCyc4 tet-OFF model, we determined that TgCyc4 was essential for tachyzoites, and a 6-h treatment with anhydrotetracycline (ATc) efficiently downregulated TgCyc4 (Fig. 1C,D). To confirm TgCrk4-TgCyc4 complex assembly in vivo, we isolated $^{HA}$TgCyc4 complexes and detected TgCrk4 as a major TgCyc4 interactor (Figs. 1E and EV1F,H). To examine whether the complex subunits affect the expression of one another, we introduced TgCrk4$^{myc}$ into TgCyc4 tet-OFF parasites. We found that ATc-induced TgCyc4 downregulation was accompanied by a significant reduction of TgCrk4 expression, suggesting co-dependent stabilization of the TgCrk4-TgCyc4 complex (Fig. 1C). Altogether, our results confirmed that TgCrk4 forms a specific complex with novel cyclin TgCyc4 that is essential for tachyzoite division.

### TgCyc4 expression is dynamic across endodyogeny

We reasoned that if TgCrk4 forms a complex with TgCyc4, then their expression should spatiotemporally overlap. To test this hypothesis, we endogenously tagged TgCrk4 with Spaghetti Monster 10xMyc (SM-Myc) epitope, which significantly improved visualization of this low-abundant kinase. Co-staining with a set of cell cycle markers showed constitutive and ubiquitous TgCrk4 expression with a negligible decline in the tachyzoites' last (late budding) and first ($G_1$) cell cycle stages (Fig. 1F,G, blue schematics). Examining TgCyc4 expression revealed that, contrary to its partner kinase, TgCyc4 was highly dynamic and dramatically changed localization in a cell cycle-dependent manner. Quantification of TgCyc4-positive tachyzoites with single centrosomes (TgCentrin1: single dot in $G_1$ cells) showed that TgCyc4 first emerged as a nuclear factor in mid-$G_1$ phase (Fig. 1H). This brief period of nuclear expression was followed by a pronounced accumulation of TgCyc4 on duplicated centrosomes (S-phase) that gradually faded by mid-mitosis. Previous studies showed that mitotic marker TgMORN1 facilitates separation of the subphases of

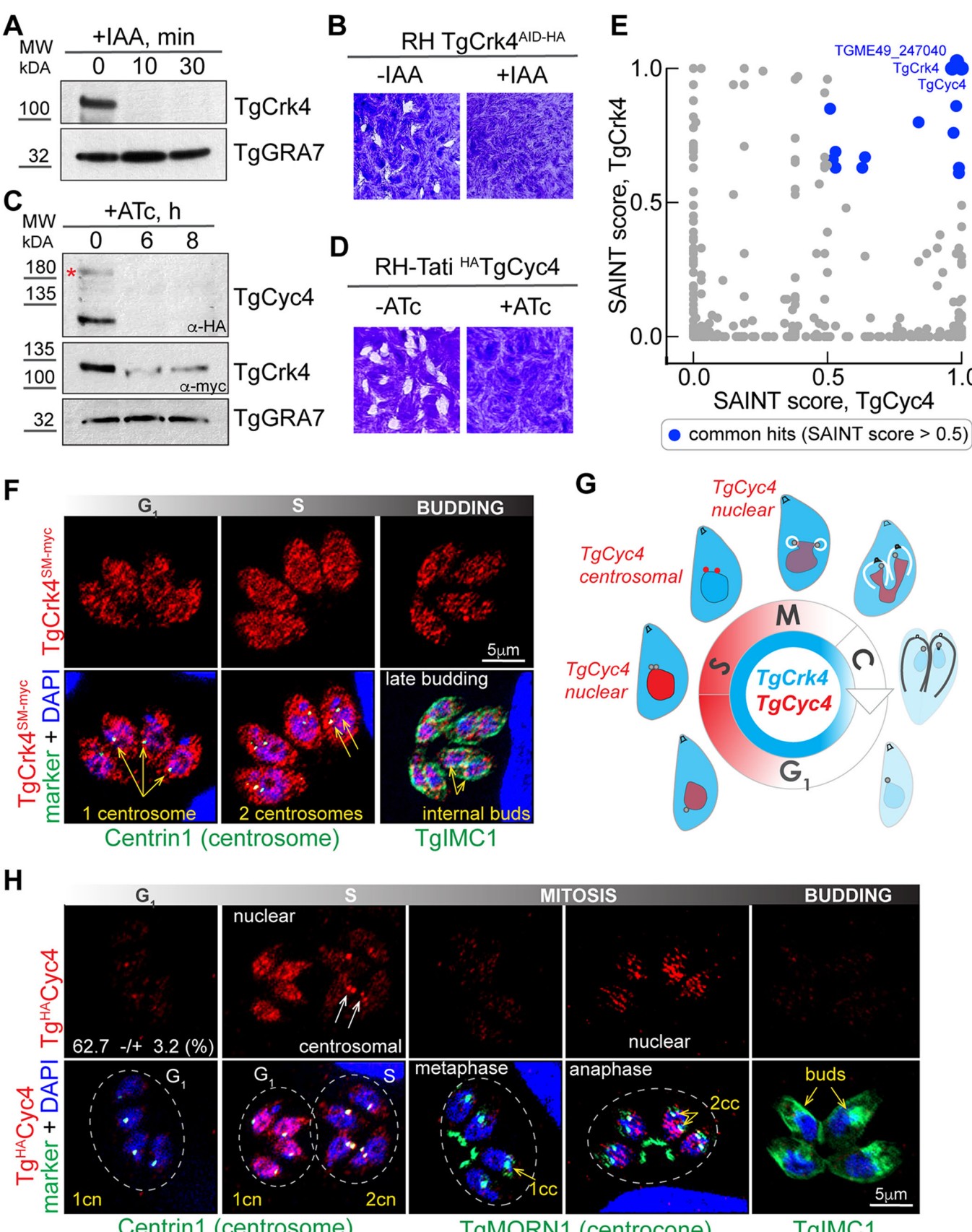

**Figure 1.  TgCrk4 forms complex with dynamically expressed TgCyc4.**

(A) The total protein extracts of the RHΔ*Ku80TIR1* tachyzoites expressing TgCrk4[AID-HA] were non-treated or treated with 500 µM IAA for 10 and 30 min and analyzed by western blot analysis using α-HA (α-rat IgG-HRP) to detect TgCrk4, and with α-GRA7 (α-mouse IgG-HRP) to confirm equal loading of the total lysates. (B) The images of the stained HFF monolayers infected with RH TgCrk4[AID-HA] tachyzoites and grown with or without 500 µM IAA for 7 days. Note that only TgCrk4-expressing tachyzoites (−IAA) formed viable plaques. The experiment has been performed in three biological replicates. (C) Western blot analysis of RHΔ*Ku80Tati* tachyzoites co-expressing tetracycline regulatable [HA]TgCyc4 and TgCrk4[myc]. Total extracts were prepared from parasites grown in the absence or presence of 1µg/ml ATc (anhydrotetracycline) for 6 and 8 h. Epitope-fused proteins were detected with α-HA (α-rat IgG-HRP) or α-myc (α-mouse IgG-HRP) antibodies. The equal protein loading was confirmed with α-GRA7 (α-mouse IgG-HRP) antibodies. The red asterisk indicates the full-length [HA]TgCyc4 protein. (D) The images of the stained HFF monolayers infected with RHΔ*Ku80Tati* [HA]TgCyc4 tachyzoites and grown with or without 1 µg/ml ATc for 7 days. The experiment has been performed in three biological replicates. (E) Comparison of TgCrk4 and TgCyc4 proteomes using SAINT scores. The common hits with SAINT score >0.5 are highlighted in blue. Note that only three factors, TgCrk4, TgCyc4 and TGME49_247040 had a score of 1, indicating a high probability to form a stable complex. (F) Immunofluorescent microscopy analysis of TgCrk4[SM-myc] cell cycle expression. The TgCrk4[SM-myc] (α-myc/α-rabbit IgG Fluor 568) co-staining with Centrin1 (α-Centrin1/α-mouse IgG Fluor 488) was used to distinguish $G_1$ (1 centrosome) and S-phase (2 centrosomes). The budding parasites were identified by co-staining with alveolar protein TgIMC1(α-IMC1/α-mouse IgG Fluor 488). The blue staining represents nucleus (DAPI). Cell cycle phases were determined based on the number of the reference structures and morphology of the nucleus. The experiment has been performed in three biological replicates. (G) Schematics of the tachyzoite cell cycle depicts the relative expression of TgCrk4 (blue) and TgCyc4 (red) deduced from immunofluorescent microscopy studies (F, H). Drawings around shows morphological changes of the dividing tachyzoite and associated changes in TgCrk4-TgCyc4 localization. (H) Immunofluorescent microscopy analysis of [HA]TgCyc4 cell cycle expression. The [HA]TgCyc4 (α-HA/α-rat IgG Fluor 568) was co-stained with Centrin1 (α-Centrin1/α-mouse IgG Fluor 488) was used to distinguish $G_1$ and S-phases and with TgMORN1 (α-MORN1/α-rabbit IgG Flour 488) to identify parasites in metaphase and anaphase of mitosis. The budding parasites were visualized with antibodies against alveolar protein TgIMC1(α-IMC1/α-rabbit IgG Fluor 488). The experiment has been performed in three biological replicates. A minimum hundred random vacuoles per experiment were evaluated. Cell cycle phases were determined based on the number and morphology of the reference structures and morphology of the nucleus (DAPI, blue). The numbers in the Centrin panel represent the percentage of the TgCyc4-expressing parasites that contain a single centrosome (unpaired two-sided $t$ test value 0.00065). Source data are available online for this figure.

mitosis (Alvarez and Suvorova, 2017; Hawkins et al, 2022). Besides in the mother basal complex (mBC), TgMORN1 was present on the daughter basal complexes (dBC), known as MORN-rings. In metaphase, dBCs were associated with a single intranuclear compartment centrocone that resolves into two compartments in anaphase (Fig. 1H) (Hawkins et al, 2022). We detected a second peak of TgCyc4 expression at the spindle assembly checkpoint operating during metaphase-to-anaphase transition. Co-staining with TgMORN1 revealed that TgCyc4 briefly accumulated in the nucleus of anaphase cells (2dBCs and two centrocones) and was completely gone by mid-budding (Fig. 1H,G). The spatiotemporal expression of TgCrk4 and TgCyc4 echoed that of conventional cell cycle CDK-cyclin complexes that contain a constitutively expressed kinase and an oscillating cyclin (Nasmyth, 2001). The alternating cell cycle localization of TgCyc4 suggests that TgCrk4–TgCyc4 complex controls events in the nucleus and on the centrosome.

## TgCrk4 regulates cell cycle events located upstream of the spindle assembly checkpoint

The timing of the TgCrk4-TgCyc4 complex expression indicates its role takes place in post-$G_1$ cell cycle phases. To determine the timing of TgCrk4-regulated processes, we compared TgCrk4-dependent cell cycle arrest with the previously examined spindle assembly checkpoint (SAC) regulated by TgCrk6–TgCyc1 complex (Hawkins et al, 2022). Thereby, we established a semi-synchronization approach and monitored the cell cycle progression of parasites enriched at a specific checkpoint. First, we determined the minimal period of TgCrk4 knockdown that parasites tolerated without losing viability. We found that, like TgCrk6, about 80% of parasites reverted to division after 4 h of TgCrk4 deficiency (+IAA), while a longer treatment severely affected parasite survival (Fig. EV2A; Dataset EV4). Western blot analysis verified the reversibility of this auxin-induced block; it showed that TgCrk4- and TgCrk6-deficient parasites could restore their expression within 1 h of auxin removal (Fig. 2A).

We treated TgCrk4 and TgCrk6 AID parasites with auxin for 4 h (block) and monitored the release from block over 5 h. Our results showed that retention of asynchronously dividing tachyzoites at a checkpoint for half of a division cycle led to significant enrichment of a specific cell cycle population (Fig. 2B). To determine the timing of TgCrk4-dependent arrest, we quantified parasites progressing through the budding stage. The asynchronous population of tachyzoites contained around 20% of parasites undergoing budding, which was detected with the budding marker TgIMC1. Contrary to TgCrk6-dependent arrest that led to a twofold increase, TgCrk4-dependent arrest reduced the number of budding parasites. Quantification showed that less than 10% TgCrk4-deficient parasites had internal buds, suggesting that TgCrk4 block occurs earlier than the TgCrk6 arrest. After TgCrk4 and TgCrk6 activities were restored (within an hour), the arrested parasites progressed to the next stage of the cell cycle in near synchrony (Fig. 2B). According to the budding dynamics, our semi-synchronization approach resulted in a 10-fold enrichment of cell cycle populations. Quantifying budding parasites upon release from block confirmed the difference in the timing of TgCrk4- and TgCrk6-regulated processes. Tachyzoites with restored TgCrk4 activity reached the budding peak an hour later than parasites with restored TgCrk6 expression (Fig. 2B,C). Based on our findings, we placed the TgCrk4-dependent process ~1 h (1/7 of cell division duration) upstream of the SAC that controls metaphase-to-anaphase transition in mitosis. The relative positions of these regulatory steps suggests that TgCrk4 is an ideal candidate to regulate entry into mitosis at the $G_2$ period that was presumed missing in apicomplexan cell cycles.

## Knockdown of TgCrk4 prevents segregation of duplicated centrosomes and spindle assembly

To determine what processes are regulated by the TgCrk4/TgCyc4 complex, we examined the phenotypes of tachyzoites lacking subunits of this complex. Since downregulation of TgCyc4 in the

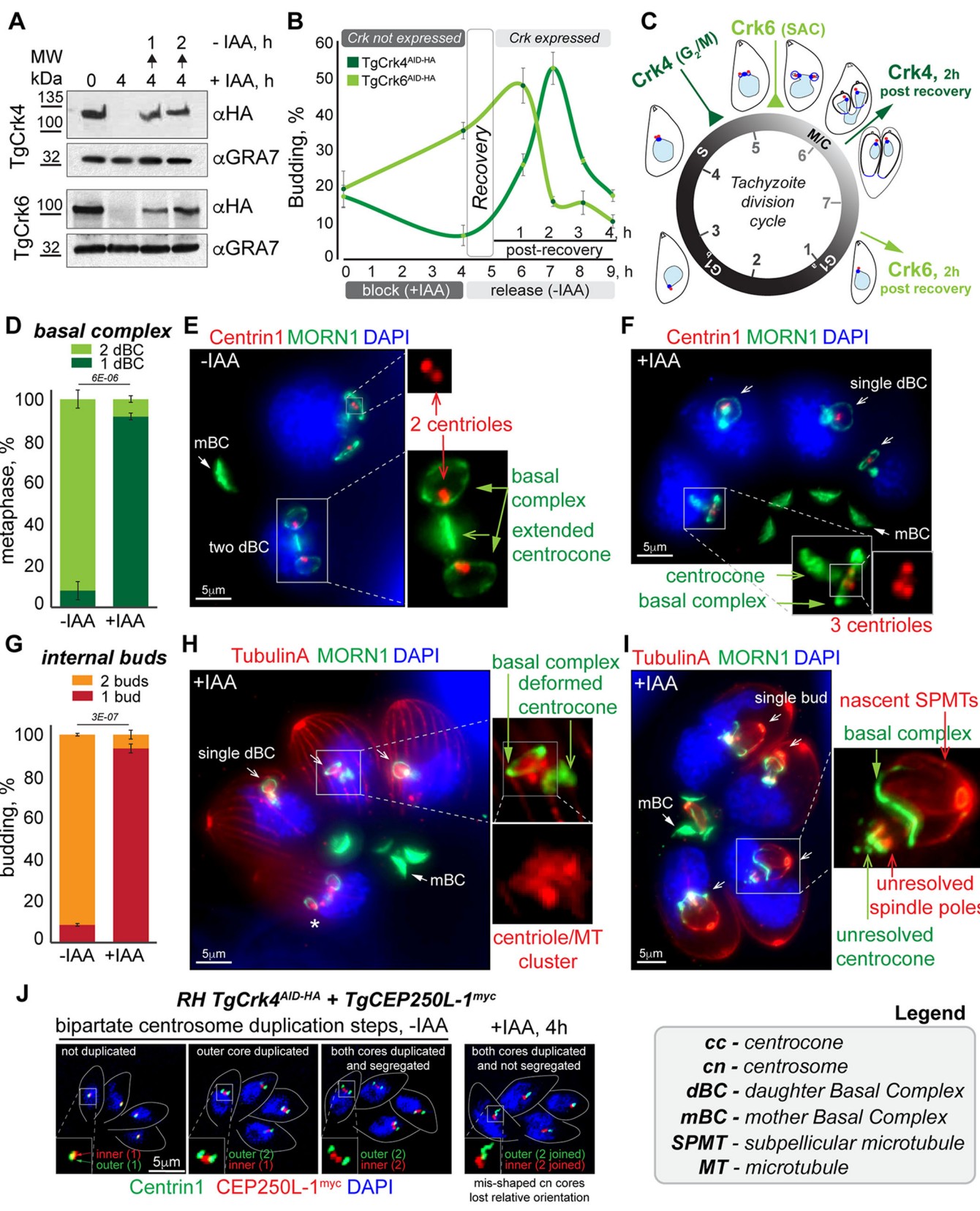

**Figure 2.  TgCrk4 regulates centrosome duplication and budding events upstream of the spindle assembly checkpoint.**

(A) Western blot analysis of the total lysates of the RHΔKu80TIR1 tachyzoites expressing TgCrk4[AID-HA] (upper panels) or TgCrk6[AID-HA] (bottom panels). Lysates of non-treated parasites, parasites treated with 500 μM IAA for 4 h and chased for 1 and 2 h were analyzed. Western blots were probed with α-HA (α-rat IgG-HRP) and with α-GRA7 (α-mouse IgG-HRP) to confirm equal loading of the total lysates. (B) Quantification of the budding populations of TgCrk4[AID-HA] (dark green) or TgCrk6[AID-HA] (light green) expressing tachyzoites during block (4 h with 500 μM IAA) and release (5 h followed IAA removal). A hundred random vacuoles of the parasites were examined for α-TgIMC1-positive staining of the internal buds in three independent experiments. The recovery period marks first hour after IAA was removed from the growth medium. The mean −/+ SD values of three independent experiments are plotted on the graph. (C) The cell cycle diagram indicates relative positions of TgCrk6- and TgCrk4-regulated blocks at SAC and G₂/M checkpoints and the timing of a 2-h recovery from each block calculated from semi-synchronization experiments shown in (B). (D) Quantification of the primary defect caused by RH TgCrk4[AID-HA] deficiency. The centrocones connected to 2 (2dBC: light green) or 1 daughter basal complex (1dBC: dark green) were quantified in non-treated and treated with 500 μM IAA for 4 h parasites using TgMORN1 staining. A hundred random vacuoles of the parasites were examined in three independent experiments. The mean −/+ SD values are plotted on the graph. The unpaired two-sided *t* test value is 0.000006. (E, F) The ultra-expansion microscopy analysis of RHΔKu80TIR1 TgCrk4[AID-HA] expressing (E) and deficient (F) tachyzoites. (E) Co-staining of Centrin1 (α-Centrin1/α-mouse IgG Fluor 568) and TgMORN1 (α-MORN1/α-rabbit IgG Flour 488) shows duplicated centrosomes containing two centrioles, extended centrocone and two daughter basal complexes (dBC) organization in the metaphase. TgMORN1 localization in the mother basal complex (mBC) is indicated. Nucleus stained with DAPI (blue). (F) Co-staining of the TgCrk4-deficient parasites (4 h, 500 μM IAA) revealed a single dBC with multiple centrioles connected to deformed centrocone. The experiment has been performed in three biological replicates. (G) Quantification of the budding defect caused by RH TgCrk4[AID-HA] deficiency. The number of the buds per cell were quantified in non-treated and treated with 500 μM IAA for 8 h using TgIMC1 staining. The longer IAA treatment allowed the development of the bigger buds to aid quantifications. A hundred random vacuoles of the parasites were examined in three independent experiments. The mean −/+ SD values are plotted on the graph. The unpaired two-sided *t* test value is 0.0000003. (H, I) The ultra-expansion microscopy analysis of RHΔKu80TIR1 TgCrk4[AID-HA]-deficient tachyzoites. Staining with Tubulin A (α-TubulinA/α-mouse IgG Fluor 568) depicts subpellicular, centriolar microtubules (MTs) and the lack of spindle MTs. The TgMORN1 (α-MORN1/α-rabbit IgG Flour 488) staining shows changed morphology of the centrocone and number of the daughter basal complexes (dBC). Nucleus stained with DAPI (blue). The experiment has been performed in three biological replicates. (J) Immunofluorescent microscopy analysis of RHΔKu80TIR1 tachyzoites co-expressing TgCrk4[AID-HA] and TgCEP250-L1[myc]. Parasites were co-stained with α-Centrin1 (α-mouse IgG Fluor 488) and α-myc (α-rabbit IgG Flour 568) antibodies to visualize the outer (Centrin1) and inner (TgCEP250-L1) cores of the bipartite centrosome. Three representative images of non-treated tachyzoites (−IAA) show the normal duplication of the centrosomal cores and morphological changes caused by TgCrk4 deficiency (+IAA, 4 h) are shown on the right panel. The experiment has been performed in three biological replicates. Source data are available online for this figure.

tet-OFF model was too lengthy to isolate the primary phenotype, we focused on TgCrk4 deficiency that can be achieved within 10 min of auxin treatment of TgCrk4 AID parasites. Given that the G₂ period was considered missing in apicomplexans, there are no stage-specific markers. Therefore, we used a combination of Centrin1, TgMORN1, Tubulin A, and TgIMC1 markers to resolve the neighboring S-phase and mitosis concurrent with budding (Engelberg et al, 2022; Hawkins et al, 2022; Suvorova et al, 2015; Tomasina et al, 2022). Examination of TgMORN1-positive structures by ultra-expansion microscopy (U-ExM) showed that over 90% of TgCrk4-deficient parasites had altered the ratio of centrocone-to-daughter basal complexes (dBC) (Fig. 2E,F,H,I). Normal progression throughout tachyzoite S-phase and mitosis involves the expansion of the intranuclear compartment centrocone, followed by the development of two dBCs either attached or near the centrocone (Fig. 2E) (Engelberg et al, 2022). As the parasite transitions to anaphase in mitosis, the spindle, which is assembled in the centrocone, breaks, leading to centrocone resolution into two compartments, each of which is associated with a single dBC (Engelberg et al, 2022; Hawkins et al, 2022). We found that, instead of the typical two dBCs associated with the single centrocone, TgCrk4-depleted parasites had a single dBC linked to a deformed centrocone (Figs. 2D–F and EV2D). Although it was similar to the ratio seen in anaphase, TgCrk4-deficient tachyzoites had a single unresolved centrocone linked to a single dBC (Fig. 2H, asterisk indicates normal anaphase). TgCrk4 deficiency also affected apicoplast segregation and fission that occurs concurrently with mitosis (Fig. EV2B,C; Dataset EV4). Co-staining of the centrocone and dBC (MORN1) with microtubules (Tubulin A) or with the inner membrane complex (TgIMC1) confirmed the reduction of daughter cytoskeletal structures (Figs. 2H,I and EV3). Nearly all budding parasites began assembling a single internal bud templated on a single dBC (Figs. 2G–I and EV2D,E; Dataset EV4). Downregulation of the

partner cyclin TgCyc4 resulted in a similar increase of the populations producing a single internal bud (Fig. EV2F,G).

The appearance of a single centrocone linked to a single dBC can result from a failed centrosome duplication or duplicated centrosomes that failed to segregate. Our observations strongly support the latter. Staining with α-Tubulin A and α-Centrin1 antibodies revealed centrioles and possibly short microtubule (MT) fibers accumulated near the unresolved centrocone (Fig. 2F–I). The presence of more than two centrioles in the cluster confirmed that TgCrk4-deficient parasites can successfully duplicate centrosomes. The clustered centrioles near a single centrocone confirmed that duplicated centrosomes did not segregate even in budding parasites. The *T. gondii* centrosome is composed of inner and outer cores with differential protein composition (Chen and Gubbels, 2019; Courjol and Gissot, 2018; Suvorova et al, 2015). TgCentrin1 is localized at the outer, centriole-containing core while TgCEP250-L1 is preferentially expressed in the inner centrosomal core (Tomasina et al, 2022). To determine if TgCrk4 down-regulation affects the inner core, we placed a 3xmyc epitope tag on endogenous TgCEP250-L1 protein in the RH TgCrk4[AID-HA] transgenic line (Fig. 2J). IFA analysis of TgCrk4-expressing tachyzoites confirmed the expected stepwise duplication of both outer and inner cores of the centrosome. Examination of tachyzoites treated with 500 μM IAA for 4 h revealed that centrosomal cores were affected by TgCrk4 deficiency. Judging by their size and morphology, both cores appeared to be duplicated but failed to segregate. In addition, the centrosomal cores lost their proper orientation relative to one another, often placed orthogonally rather than parallel to each other.

The inner core of the centrosome is tightly aligned with the centrocone (Suvorova et al, 2015). In the normal *T. gondii* mitosis, centrocone extension coincides with spindle pole growth (Figs. 2E and EV3A) (Engelberg et al, 2022; Hawkins et al, 2022; Suvorova et al, 2015). Examination of TgCrk4-deficient parasites revealed

that deformed centrocones did not incorporate an extended pole-like structure, implying that TgCrk4-deficient parasites fail to assemble a bipolar spindle (Figs. 2H,I and EV3B). Thus, we concluded that TgCrk4-deficient parasites could neither segregate duplicated centrosomes nor form a bipolar spindle. In the conventional cell cycle, segregation of duplicated centrosomes is a prerequisite for bipolar spindle assembly, which signifies the entry into mitosis, also known as the $G_2/M$ transition (Agircan et al, 2014). The similar regulation of these processes and the spatiotemporal localization of the TgCrk4-TgCyc4 complex suggest that, contrary to previous assumptions, the $G_2$ period does operate in apicomplexan cell cycles. In *T. gondii* endodyogeny, the $G_2/M$ transition is regulated by the novel TgCrk4-TgCyc4 complex.

## TgCrk4 represses multinuclear division by blocking centrosome and chromosome reduplication

To further verify the role of TgCrk4 in centrosome segregation and spindle assembly, we examined the recovery of TgCrk4-deficient tachyzoites during the first 3 h of restored TgCrk4 expression (Fig. 3). In good agreement with the predicted role of TgCrk4 in centrosome segregation, tachyzoites released from TgCrk4-dependent block began to separate their centrosomes. The number of separated Centrin1-positive centrosomes increased from 10 to 50% (Fig. 3A–D). Surprisingly, we noticed the emergence of parasites with three or more centrosomes concurrently to an increase of parasites with multiple internal buds (Fig. 3D,E).

FACScan analysis of their DNA content confirmed that release from block led to decrease of $G_1$ and a concurrent increase of populations that duplicated and overduplicated DNA (Fig. 3H). During 2 h of post-block recovery, the number of parasites whose DNA content exceeded 2N had increased, suggesting that a fraction of TgCrk4-deficient parasites reduplicated their chromosomes. IFA analysis of the nuclei showed that DNA mis-segregation had amplified during recovery, confirming that it was a secondary effect of TgCrk4 deficiency, in particular, the outcome of centrosome reduplication (Fig. 3F). The phenotypic analysis of parasites emerging from TgCrk4-induced block showed that TgCrk4-deficient parasites broke the "once per cell cycle" rule of DNA and centrosome duplication (Nigg and Holland, 2018; Sclafani and Holzen, 2007).

In the conventional cell cycle, centrosome duplication is tightly linked to chromosome replication (Arbi et al, 2018). Centrosome duplication happens at the $G_1/S$ transition and is blocked for the rest of the division cycle (Nigg and Holland, 2018). Likewise, licensing of DNA replication takes place in $G_1$ and is forbidden in S and $G_2$ phases (Parker et al, 2017). The major goal of $G_2$ period is to repress an unlawful DNA rereplication and centrosome reduplication (Reusswig and Pfander, 2019). The fact that TgCrk4-deficient parasites had multiple centrosomes and over-replicated chromosomes indicated that S-phase events were reinitiated, which happens only in multinuclear divisions such as those employed by apicomplexan schizogony. The fraction of parasites with multiple centrosomes or buds likely represented a population that spent the longest time with arrested cell cycles (Fig. 3G). It is possible that non-segregated centrosomes serve as a signal of incomplete DNA replication, allowing relicensing and refiring of DNA replication origins, and reinitiating centrosome duplication. The quick reduction of the number of multinuclear

dividing parasites released from the TgCrk4-dependent block further confirmed the primary role of TgCrk4 in the repression of multinuclear and promotion of binary division. The accompanied growth of mis-segregation defects suggests that the switch from binary to multinuclear division involves multiple regulators, some of which may not be expressed in tachyzoites programmed for binary division.

## Global profiling of TgCrk4-induced cell cycle block

To determine the cellular effect of TgCrk4 deprivation, we examined changes in global protein expression and phosphorylation in tachyzoites that were held in a TgCrk4-induced block for 4 h (Fig. 4A,B). Our results showed specific changes in the protein landscape that corroborated our immunofluorescent microscopy analyses of TgCrk4-deficient tachyzoites (Figs. 4A, 2, and 3). We detected a nearly complete tachyzoite proteome (4219 proteins), out of which the TgCrk4-deprived tachyzoites had 58 upregulated and 126 downregulated proteins (Dataset EV3). In good agreement with the predicted role of TgCrk4 in repressing chromosome rereplication, GO term analysis detected upregulation of DNA replication licensing factors TgORC4, TgORC5, and DNA polymerase. Since DNA repair is a process that takes place in $G_2$ phase, the increase in expression of DNA repair proteins in TgCrk4-deficient tachyzoites further confirms the role of TgCrk4 as a $G_2$ phase regulator. Corroborating TgCrk4-dependent suppression of budding, we detected decreased expression of inner membrane complex proteins that are enriched on daughter buds, such as IMC16, IMC30, IMC33, and IMC34 (Back et al, 2023; Butler et al, 2014; Dos Santos Pacheco et al, 2021). Regulated proteolysis plays a vital role in the progression through S, $G_2$ phase, and mitosis (King et al, 1996). The TgCrk4-induced block led to the downregulation of deubiquitinase TgOTUD3A and caspase TgMCA1. Interestingly, previous studies showed that TgOTUD3A knockdown induced a switch from binary to multinuclear division, which resembles the TgCrk4-deficiency phenotype and suggests a functional link between these factors (Dhara et al, 2017).

The comparative global phosphoproteome of asynchronous and TgCrk4-arrested tachyzoites contained 23,480 phosphopeptides (Dataset EV3). TgCrk4-dependent cell cycle arrest resulted in increased phosphorylation of 1112 and decreased phosphorylation on 883 phospho-sites (Fig. 4B). Examining short- (30 min) and long-term (4 h) TgCrk4 deprivation revealed five patterns (classes) of temporal phosphorylation (Fig. 4C). We performed a GO term enrichment analysis of class I and V proteins that showed the loss or steady increase of phosphorylation, respectively (Fig. 4D). TgCrk4 knockdown significantly affected the protein expression machinery that controls transcription, translation, and regulated proteolysis. A dominating group of nucleic acid binding proteins included 16 AP2 DNA binding factors, among which are three transcriptional repressors AP2XII-1, AP2XI-2, and AP2IX-9, which function is linked to parasite development (Antunes et al, 2024; Radke et al, 2013). A few histone modifiers and nucleosome assembly factors also appeared to be regulated by alternative phosphorylation at the $G_2$ block. Protein kinases constituted the second large group affected by TgCrk4 downregulation and included essential for tachyzoite replication kinases, MAPK1 and TKL2 (Brown et al, 2014; Suvorova et al, 2015; Varberg et al, 2018). We had previously shown that activating a temperature-sensitive

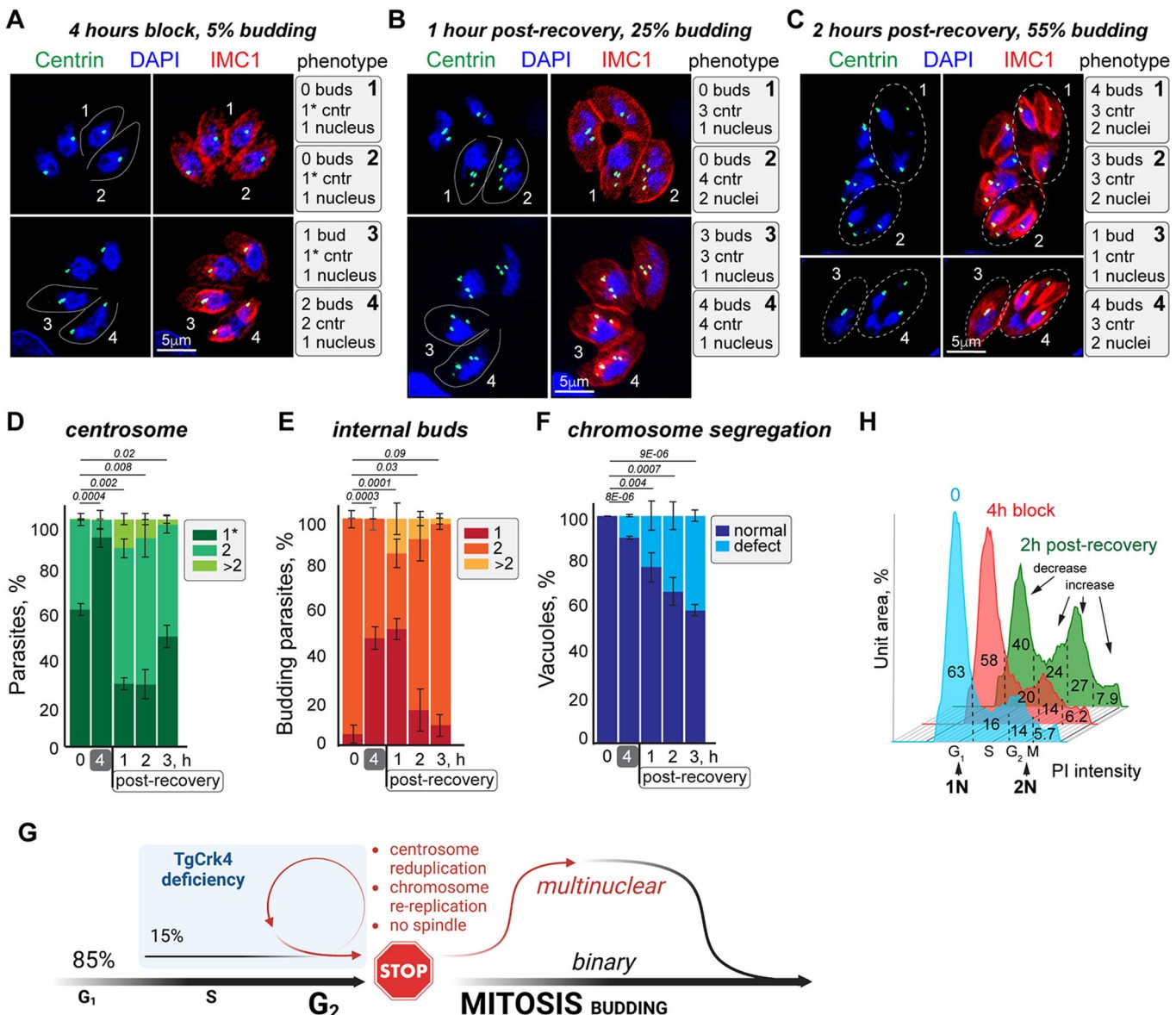

**Figure 3. TgCrk4 deficiency leads to centrosome reduplication and DNA rereplication.**

(A–C) Immunofluorescent microscopy analysis of RHΔ*Ku80TIR1* TgCrk4^AID-HA tachyzoites incubated with 500 µM IAA for 4 h (A) [block] and then without IAA for 2 (B) [1-h post-recovery] or 3 h (C) [2 h post-recovery]. Parasites were co-stained with α-Centrin1 (α-mouse IgG Fluor 488), α-TgIMC1 (α-rabbit IgG Fluor 488) antibodies and DAPI to estimate the ratio of centrosome, nuclei and internal buds per parasite. Analyzed cells are numbered and the summary is shown on the right of each panel. The experiment has been performed in three biological replicates. (D–F) Number of centrosomes (D) and internal buds (E) per parasite, and DNA mis-segregation defects (F) were quantified in asynchronously growing (0 h) RHΔ*Ku80TIR1* TgCrk4^AID-HA tachyzoites as well as in the arrested (4 h with 500 µM IAA, gray block), and recovered (1, 2, and 3 h after kinase expression was restored) populations. A hundred random vacuoles of the parasites were examined in three independent experiments. The mean −/+ SD values are plotted on the graphs. The unpaired two-sided *t* test values for duplicated centrosomes (D), two internal buds (E) and chromosome segregation defects are shown. The *t* test values for overduplicated centrosomes (D) and assembly of more than two buds (E) are listed in the Dataset EV4. (H) FACScan analysis of DNA content of TgCrk4^AID-HA expressing (0, blue plot), TgCrk4^AID-HA deficient (4 h with 500 µM IAA, red plot) and recovered from TgCrk4^AID-HA deficiency (3 h after IAA removal, green plot) parasites. The results of one of three independent experiments are shown. Dashed lines show the gates used to segregate parasites containing non-replicated (G$_1$: <1 − 1N), replicating (S: 1 − <2N), replicated (G$_2$ + M: 2N) and over-replicated (>2N) DNA. (G) Cell cycle effect of the TgCrk4 deficiency. The schematics shows the cell cycle progression of TgiRD1-deficient tachyzoites. Most of the parasites quickly recover from TgCrk4-dependent G$_2$ arrest (red STOP sign). About 15% of the tachyzoites that spent a longer time in the block re-enter S-phase (blue block: centrosome and chromosome reduplication), which led to one-cycle multinuclear division. The surviving DNA mis-segregation defect population quickly reverted to binary division upon restoration of TgCrk4 activity. Source data are available online for this figure.

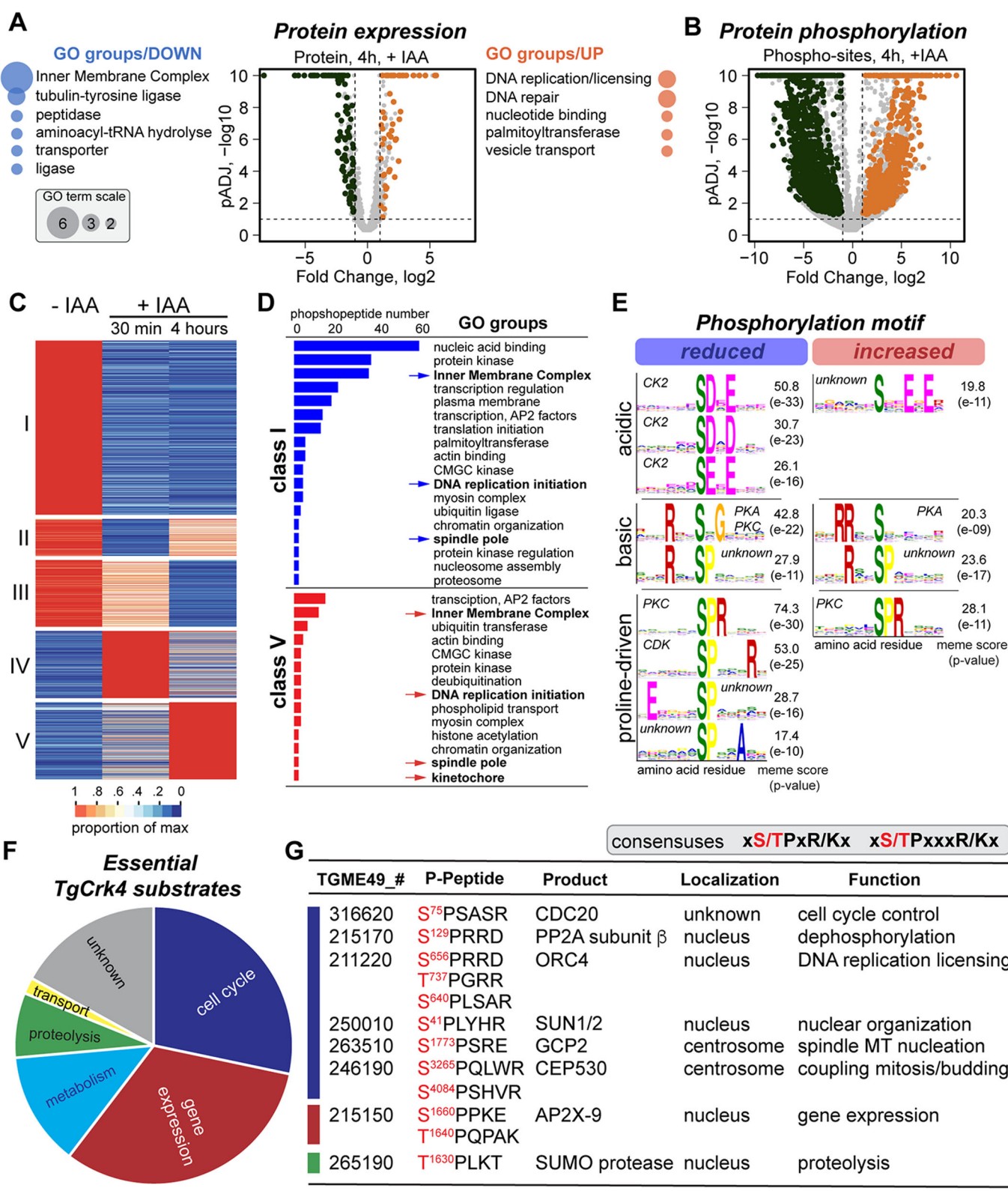

◀  **Figure 4.   Global proteomic and phosphoproteomic analysis of *T. gondii* G$_2$/M checkpoint.**

(A, B) Volcano plots show changes of the protein expression level (A) and of the phospho-sites intensity (B) at the checkpoint block induced by TgCrk4$^{AID-HA}$ downregulation 4 h. The GO term enrichment analysis performed on upregulated (orange) and downregulated (blue) groups identified in global proteome. Results are shown on the sides of each plot. The bubbles reflect the size of individual pools. The experiment has been performed in two biological replicates. (C) The heatmap displays global changes of the protein phosphorylation caused by auxin-induced TgCrk4$^{AID-HA}$ degradation for 30 min and 4 h. The proteins are organized according to the similarity in phosphorylation by *K*-means and combined into five classes based on the temporal dynamic of phosphorylation. (D) The graphic representation of the GO term analysis of class I (reduced phosphorylation) and V (increased phosphorylation) phosphopeptides. The groups of particular interest are indicated with arrows and shown in bold. (E) The significant phosphorylation motifs affected by 30 min TgCrk4$^{AID-HA}$ degradation were deduced using Soft Momo 5.1.1. software. Three-letter motifs with *P* values >e-09 are shown. Responsible kinase family and scores are indicated on the corresponding plots. (F) Pie chart of the putative TgCrk4 substrate proteins. The essential proteins that reduced or lost intensity of TgCrk4-dependent phosphorylation were categorized. (G) A selected set of putative TgCrk4 substrates that were identified based on the reduction of phosphorylation intensity within a Proline-driven motif (consensus) caused by the lack of TgCrk4 for 30 min and 4 h (Class I, (C)). Affected phospho-Serine or phospho-Threonine residues are shown in red. Source data are available online for this figure.

allele of TgMAPK1 led to centrosome amplification, which placed TgMAPK1 into the broader G$_2$ network (Suvorova et al, 2015). We detected 34 IMC proteins that lost and 11 that gained phospho-modifications, which likely reflects the repressed budding in TgCrk4-deficient tachyzoites. In good agreement with TgCrk4 role in regulating G$_2$, both class I and class V proteins included DNA replication licensing factors MCM2, 4, 6, and 7, phosphorylation of which may be a part of the mechanism that represses relicensing of DNA replication. Furthermore, TgCrk4 depletion resulted in the altered phosphorylation of γ-tubulin ring complex proteins GCP2 and GCP4 that nucleate spindle MTs, and increased phosphorylation of the apicomplexan-specific kinetochore proteins AKIT1 and AKIT6, which suggests they may regulate repression of the spindle and kinetochore assembly (Brusini et al, 2022).

To find out what kinases operate in the *T. gondii* G$_2$ phase, we determined phosphorylation motifs that were affected in the brief absence of TgCrk4 (30 min) (Fig. 4E). The MEME analysis revealed a specific reduction of phosphorylation within 33 motifs and increased phosphorylation within 14 motifs representing three major groups: acidic, basic, and proline-driven motifs. The loss of TgCrk4 affected casein kinase 2 (CK2), Cdk-, and PKC/PKA-dependent phosphorylation. While the activity of PKC and PKA seem to be differentially modulated, CK2 and Cdk activities were specifically reduced in response to TgCrk4 removal. Interestingly, TgCrk4 degradation largely affected proteins containing an uncommonly extended Cdk motif [(S/T)PxxxK/R], adding another nonconventional feature to TgCrk4. In addition, we detected several unknown or highly deviated phosphorylation motifs that implicate the activity of novel kinases in the control of G$_2$ phase.

## Search for TgCrk4 substrates

To identify immediate TgCrk4 effectors, we interrogated phospho-peptides that contain the most common (S/T)*PxR/K and extended (S/T)*PxxxR/K Cdk motifs which had lost or reduced phosphor-ylation both immediately (30 min, +IAA) and after prolonged TgCrk4 downregulation (4 h, +IAA) (Fig. 4E) (Errico et al, 2010). Out of 109 peptides that matched the criteria, 54 were predicted to be essential proteins (Dataset EV3). Based on ToxoDB database annotations, the most abundant categories of putative TgCrk4 substrates were proteins involved in cell cycle regulation (15 peptides from 12 proteins) and gene expression control (17 peptides from 15 proteins) (Fig. 4F). Validating the role of TgCrk4 in chromosome rereplication, three residues of the DNA replication licensing factor TgORC4 (TGME49_211220), S$^{656}$, S$^{737}$ and T$^{737}$

were predicted to be dependent on TgCrk4 phosphorylation (Fig. 4G). Coincidently, global protein expression analysis of TgCrk4-deficient tachyzoites showed increased TgORC4 expression (Fig. 4A; Dataset EV6). This suggests that one of the mechanisms controlling chromosome rereplication in G$_2$ phase could be affected by reduced stability of phosphorylated TgORC4. The list of putative substrates included the repressor of the anaphase-promoting complex/cyclosome (APC/C) Cdc20 (TGME49_316620) (Fig. 4G). In the conventional cell cycle, the Cdc20 phosphorylation occurs at the G$_2$/M transition, which further confirms that TgCrk4 regulates G$_2$ phase in tachyzoites. Furthermore, our results suggest that TgCrk4 likely controls initiation of spindle pole assembly by directly phosphorylating TgGCP2 (TGME49_263510), a compo-nent of the γ-tubulin ring complex responsible for nucleating spindle microtubules (Bohler et al, 2021). Interestingly, we also detected factors that were not previously implicated in G$_2$ phase control, such as the nuclear membrane protein SUN1/2 (TGME49_250010), bipartite centrosome protein TgCEP530 (TGME49_246190) and DNA binding factor TgAP2X-9 (TGME49_215150), suggesting novel routes of G$_2$ regulation (Courjol and Gissot, 2018) (Fig. 4G). It was recently shown that TgAP2X-9 transcription is directly regulated by another AP2 factor, TgAP2IX-5 implicated in the control of binary division in *T. gondii* tachyzoites (Khelifa et al, 2021). Lastly, the progression through the G$_2$/M transition involved specific changes in protein expression and phosphorylation that could be dependent on TgCrk4 phosphorylating the TgSUMO protease (TGME49_265190) and a regulatory β-subunit of TgPP2A phosphatase (TGME49_215170).

## Functional orthologs of TgCrk4-TgCyc4 complex are only present in binary dividing apicomplexans

Previous phylogenetic analyses of apicomplexan Cdk-related kinases showed that Crk4 orthologs exist in apicomplexans and ancestral alveolates (Alvarez and Suvorova, 2017). We performed a follow-up examination of the latest annotated genomes and confirmed the relatively broad inheritance of Crk4. TgCrk4 orthologs were found in three core Apicomplexa groups, cocci-dians, piroplasmids, and cryptosporidians, as well as in chrompo-delids closest to the Apicomplexa lineage, which suggests that Crk4 kinase was inherited from apicomplexan ancestors (Fig. 5A). Contrary to its partner kinase, TgCyc4 had a limited presence in the analyzed genomes. Given the lack of detectable TgCyc4 orthologs in chrompodelids, it is likely that apicomplexans evolved rather

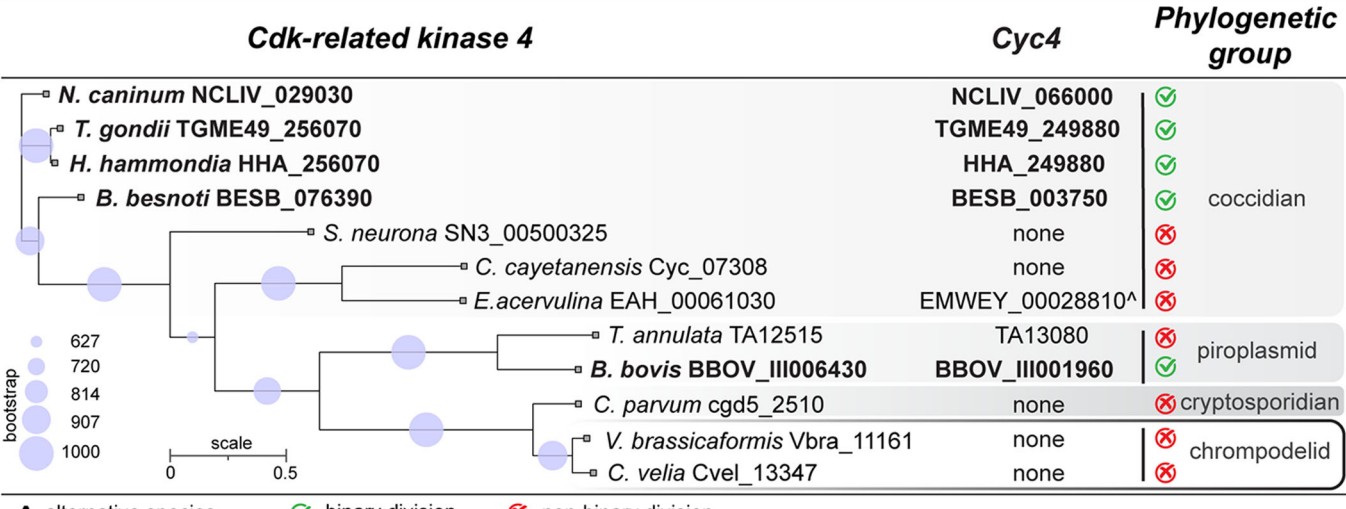

**Figure 5. Structural and phylogenetic analysis of TgCrk4–TgCyc4 complex in apicomplexans.**

(A) Summary of the phylogenetic analyses of the TgCrk4-related kinases and the orthology search of TgCyc4-related factors among Apicomplexa groups and chromopodelids. (B) Folding prediction of the cyclin domains of *T. gondii* Cyc4, *Babesia bovis* Cyc4, and *Theileria annulate* Cyc4 overlayed with H. sapiens Cyclin A2 and Cyclin E1 based on best predicted fit (SWISS-MODEL). Changes in the structures are indicated with arrows. The percentage of identical amino acid residues in the modeled regions is shown below. (C) Comparative schematics of *H. sapiens* Cyclin A2 and TgCyc4, BbCyc4, and TaCyc4 protein organization.

than inherited this cyclin. The fact that TgCyc4-related factors were found only in several coccidian species and in piroplasmids further corroborated the possibility of an independent rise of Cyc4 factor in apicomplexans.

Structural analyses of apicomplexan TgCyc4-related factors revealed a limited resemblance to mammalian Cyclin A and Cyclin E. Although highly similar at the level of amino acid residues, Cyclin A and Cyclin E regulate different cell cycle phases (Honda et al, 2005). In mammalian cells, Cdk2-Cyclin E complex controls the entry into S-phase, while Cdk1/2-Cyclin-A complex is involved in regulating the $G_2/M$ transition. Interestingly, our modeling of apicomplexan Cyc4 revealed the resemblance of the coccidian and *Babesia* Cyc4 with $G_2$ Cyclin A, while the *Theileria* Cyc4 was preferentially modeled to $G_1/S$ Cyclin E (Fig. 5B). It suggests that, while related, *Theileria* Cyc4 may regulate its cell cycle differently than how Cyc4 regulates cell cycle in other apicomplexans. Further examination of cyclin folds revealed the *Eimeria* ortholog of TgCyc4 had a significant alteration in its cyclin domain (Fig. EV4). Cyclins are a highly divergent group of proteins with the unifying feature of two conservative cyclin domains located in the C-terminus. The cyclin domain typically responsible for binding and activation of Cdk type kinase is composed of six short α-helixes (Fig. 5C) (Honda et al, 2005). Our AlphaFold2 analysis predicted drastic structural changes caused by sizable poly-Q insertions that fused α3, α4, and α5 helices, which would severely affect the *Eimeria* Cyc4 ability to bind or activate a Cdk kinase (Fig. EV4B).

The coccidian Cyc4 factors found in *T. gondii*, *Hammondia hammondia*, *Neospora caninum*, and *Besnoitia besnoiti* also contained an insertion between N-terminal activating and α1 helix (Figs. 5B,C and EV4A). However, our study of TgCrk4–TgCyc4 complex verified that the loop in TgCyc4 had no effect on TgCyc4 ability to selectively bind TgCrk4 (Fig. 1). Altogether, we identified six Apicomplexa genera that express functional TgCrk4-TgCyc4-related complex: *Toxoplasma*, *Hammondia*, *Neospora*, *Besnoitia*, *Babesia,* and *Theileria*. Structural similarity with $G_2$ Cyclin A suggested that the Crk4–Cyc4 complex of *Toxoplasma*, *Hammondia*, *Neospora*, *Besnoitia*, and *Babesia* may control $G_2$ events. On the contrary, similarity of *Theileria* Cyc4 and Cyclin E indicated the involvement of *Theileria* Crk4–Cyc4 complex in regulation of the different cell cycle stages or different cellular processes. Coincidently, unlike most apicomplexans, *Toxoplasma*, *Hammondia*, *Neospora*, *Besnoitia*, and *Babesia* divide in binary fashion such as endodyogeny (*T. gondii*, *H. hammondia*, *N. caninum*) or binary fission (*Babesia* spp.) (Gubbels et al, 2020). Based on our examination of TgCrk4–TgCyc4 function in *T. gondii* and our phylogenetic and structural analysis across Apicomplexa phylum, we propose that TgCrk4–TgCyc4 complex and its functional orthologs had evolved to repress multinuclear division in a subgroup of apicomplexan parasites.

## TgCrk4–TgCyc4 complex interacts with CDT1 C-terminal domain-containing factor

The interactomes of TgCrk4 and TgCyc4 identified the TGME49_247040 protein with unknown function as a primary interactor of the complex (Fig. 1E). The protein folding modeling detected a partial resemblance to the DNA replication licensing factor CDT1 (Fig. EV5A). Mammalian CDT1 is composed of an N-terminal regulatory, central region responsible for binding geminin and C-terminal minimal origin licensing domain (Fig. 6A) (Pozo and Cook, 2016). Besides its interaction with the inhibitor Geminin and a mini-chromosome maintenance (MCM) replicating helicase complex, CDT1 directly binds to DNA polymerase loading factor PCNA1 (PIP box) and $G_2$ cyclin A (Cy motif) (Pozo and Cook, 2016). We found that TGME49_247040 had limited homology to the Geminin-binding region of mammalian CDT1, but preserved the C-terminal winged helix-turn-helix domain that mediates the MCM complex binding (Fig. EV5A) (De Marco et al, 2009). Furthermore, like eukaryotic CDT1, TGME49_247040 factor had several recognizable cyclin-binding motifs (Cy). While phylogenetic analyses confirmed a common inheritance of apicomplexan TGME49_247040-related proteins and CDT1 orthologs from mammalians, land plants, fungi, and dinoflagellates, the examination of the structure revealed several discrepancies (Fig. 6B). We could not detect CRL4$^{Cdt2}$-dependent PEST degron critical for a timely CDT1 removal in S and $G_2$ phases or conventional PCNA1 binding PIP motif (Jin et al, 2006; Lee et al, 2010). Instead, the TGME49_247040 protein had three putative D-box degrons recognized by APC/C machinery and two nuclear localization signals (NLS), suggesting altered regulation or function (Fig. 6A).

To confirm the interaction of TGME49_247040 with the TgCrk4-TgCyc4 complex, we fused endogenous TGME49_247040 with a 3xmyc epitope and isolated TGME49_247040 complexes. In good agreement with the presence of Cy motifs, we detected a weak but specific interaction with TgCyc4 (Fig. 6C; Dataset EV3). No other cyclins or cell cycle Crks were detected in the TGME49_247040 proteome, suggesting that TGME49_247040 binds TgCrk4-TgCyc4 complex via the TgCyc4 subunit. Studies in metazoans demonstrated a predominant CDT1 interaction with Geminin (Li and Blow, 2005). Corroborating poor conservation of the Geminin-binding sites, we could not identify a candidate for *Toxoplasma* Geminin (Fig. EV5B). The absence of a Geminin-like factor in the TGME49_247040 proteome suggests that, like budding yeast, *T. gondii* may not encode Geminin (McGarry and Kirschner, 1998). We also did not detect the TGME49_247040 interaction with pre-RC complex components the ORC subunits critical for licensing of DNA replication in $G_1$ phase (Yuan et al, 2017). The structural alterations together with the lack of conservative interactors further indicated that TGME49_247040 may have functions different than conventional DNA replication licensing factor CDT1.

## TgiRD1 (TGME49_247040) controls DNA and centrosome reduplication in tachyzoites

Studies in metazoans and fungi established the CDT1 role in licensing of DNA replication (late mitosis and $G_1$), controlling chromosome rereplication (S-phase and $G_2$), and segregation (mitosis) (Gopalakrishnan et al, 2001; Pozo and Cook, 2016; Varma et al, 2012; Zhou et al, 2020). To examine the TGME49_247040 function in tachyzoites, we built and analyzed TGME49_247040 AID conditional expression model. Immunofluorescence microscopy analyses confirmed the nuclear localization of TGME49_247040$^{AID-HA}$, which was consistent with predicted NLS motifs and TGME49_247040 interactions with α- and β-Importins (Fig. 6C,F). Western blot analysis demonstrated the fast auxin-induced degradation of TGME49_247040$^{AID-HA}$, and a plaque assay verified that TGME49_247040 was required for

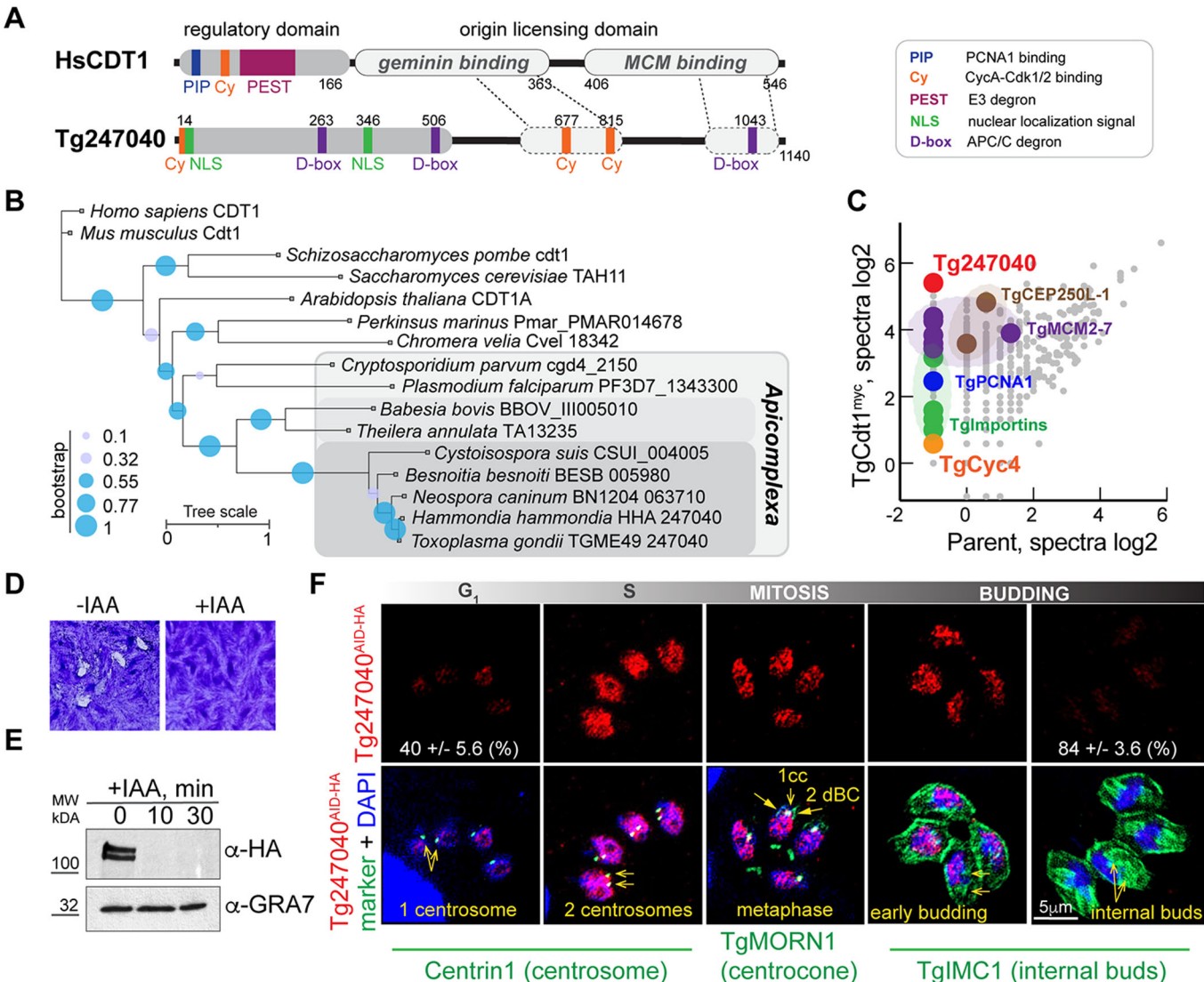

**Figure 6. *T. gondii* expresses factor related to DNA replication licensing factor CDT1.**

(A) Schematics of *H. sapiens* CDT1 and TGME49_247040 protein organization. The signature domains and TGME49_247040 regions are shown based on the ELM prediction. Dotted lines indicate two regions of similarity detected by SWISS-MODEL software. (B) Phylogenetic tree of apicomplexan CDT1-related proteins. (C) The log2 values of the protein spectra detected by mass-spectrometry analysis of the TGME49_247040$^{myc}$ complexes are plotted on the graph. The different color dots represent categories of the selected TGME49_247040$^{myc}$ interactors. The experiment has been performed in two biological replicates. (D) Images of the host cell monolayers infected with TGME49_247040$^{AID-HA}$ tachyzoites and grown with or without 500 μM IAA for 7 days. The experiment has been performed in three biological replicates. (E) Western blot analysis confirms downregulation of TGME49_247040$^{AID-HA}$ after 10 and 30-min treatment with 500 μM IAA. Western blots were probed with α-HA and α-GRA7 to confirm equal loading of the total lysates. (F) Immunofluorescent microscopy analysis of TGME49_247040$^{AID-HA}$ cell cycle expression. The TGME49_247040$^{AID-HA}$ (α-HA/α-rat IgG Fluor 568) co-staining with Centrin1 (α-Centrin1/α-mouse IgG Fluor 488) was used to distinguish G$_1$ (1 centrosome) and S-phase (2 centrosomes). The mitotic metaphase was detected by co-staining with TgMORN1(α-MORN1/α-rabbit IgG Flour 488). The budding parasites were identified by co-staining with alveolar protein TgIMC1(α-IMC1/α-rabbit IgG Fluor 488). The blue staining represents nucleus (DAPI). Cell cycle phases were determined based on the number of the reference structures and morphology of the nucleus as indicated. The experiment has been performed in three biological replicates. A minimum hundred random vacuoles per experiment were evaluated. The numbers in the Centrin panel represent the percentage of the TGME49_247040-expressing parasites that contain a single centrosome and numbers in the TgIMC1 panel represent budding TGME49_247040-negative parasites. Source data are available online for this figure.

tachyzoite growth (Fig. 6D,E). Detailed examination of TGME49_247040 function in tachyzoites revealed significant differences from the conventional eukaryotic CDT1 factor. As a critical DNA replication licensing factor, metazoan and fungal CDT1 is active on chromosomes in late mitosis and G$_1$ phase (Pozo and Cook, 2016). To avoid unlawful rereplication of the chromosomes, CDT1 dissociates from replicating chromosomes in S-phase, which is achieved by proteolysis of CDT1 in metazoans, or by shuttling CDT1 from the nucleus in budding yeast (Pozo and Cook, 2016). We found that, contrary to conventional CDT1, TGME49_247040 was abundantly expressed in the nucleus of S-phase tachyzoites and was absent in late mitosis and early G$_1$ phase (Fig. 6F).

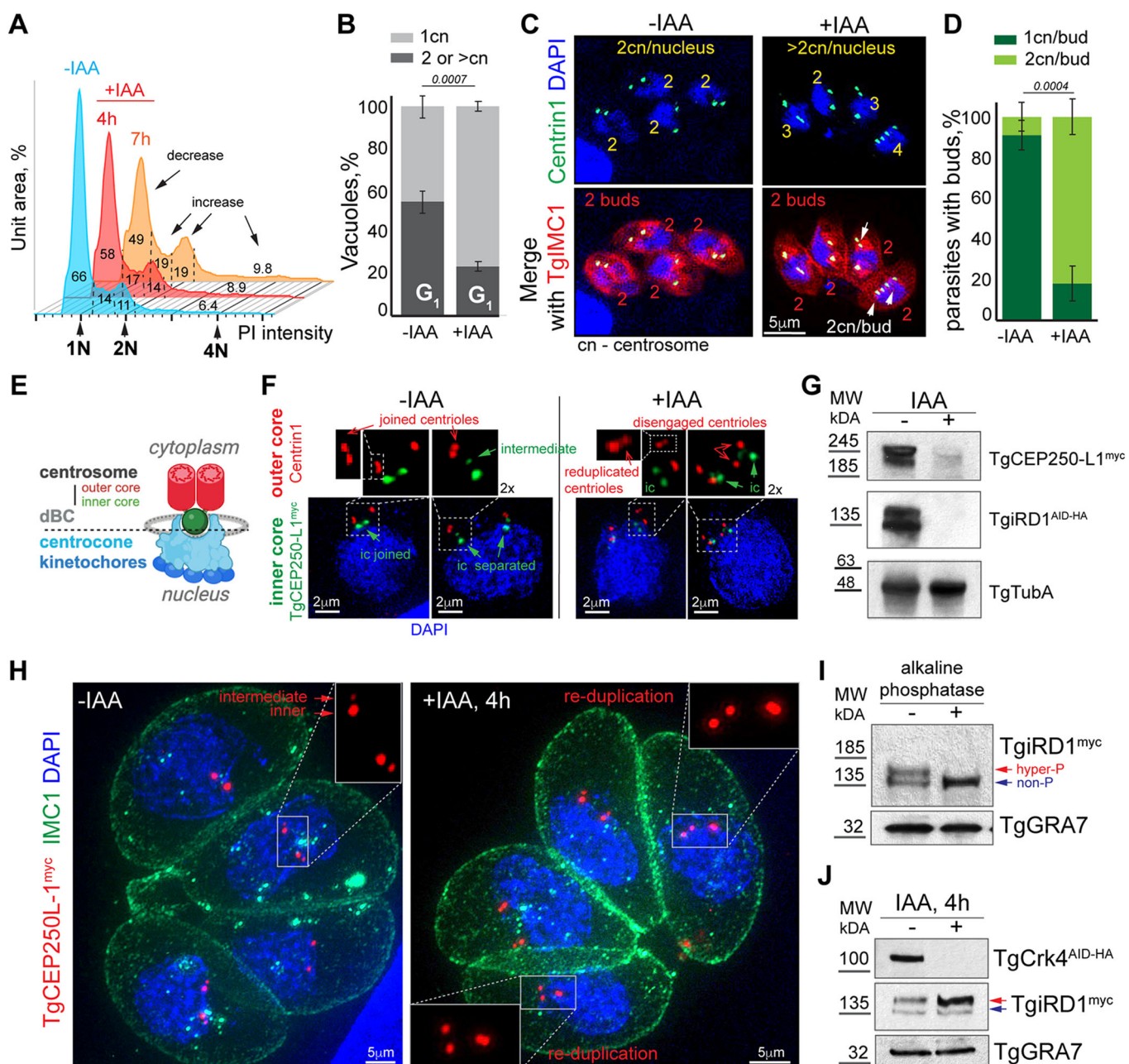

Although the TGME49_247040 interactome showed the enrichment of conventional CDT1 interactors such as a replisome complex (MCM2-7) and DNA polymerase loading factor TgPCNA1, the unpredicted spatiotemporal expression raised doubt over if TGME49_247040 functions as a DNA replication licensing factor in *T. gondii* (Fig. 6C; Dataset EV3) (Zhai et al, 2017). To address this question, we examined the DNA content in tachyzoites either expressing or depleted of TGME49_247040 for the duration of one division cycle (8 h). Our FACScan analysis of TGME49_247040 AID tachyzoites did not detect the anticipated repression of DNA replication in TGME49_247040-deficient tachyzoites (Fig. 7A). On the contrary, eliminating TGME49_247040 led to a decrease of the $G_1$ population (1N

DNA content) and was accompanied by an increase in parasites both undergoing DNA replication (1–2N DNA content) and with re-replicated DNA (>2N DNA content). The parental strain showed no changes in the DNA content under the similar conditions (Fig. EV2H). Quantitative IFA of TGME49_247040-deficient tachyzoites verified the substantial reduction of parasites in $G_1$ phase and the expansion of replicating (S-phase) and mitotic populations (Fig. 7B). These findings strongly suggested TGME49_247040 has minor to no involvement in licensing DNA replication in $G_1$ tachyzoites.

Further examination revealed that TGME49_247040 retained the conventional CDT1 role in control of chromosome rereplication and segregation, and an unexpected role in regulating

**Figure 7. TgiRD1 is a phosphorylated protein that controls centrosome reduplication and chromosome rereplication.**

(A) FACScan analysis of DNA content of TGME49_247040$^{AID\text{-}HA}$ expressing (0, blue plot) and TGME49_247040 deficient for 4 h (+IAA, red plot) and 8 h (+IAA, orange plot) parasites. The results of one of three independent experiments are shown. Dashed lines show the gates used to segregate parasites containing non-replicated (G$_1$: <1 − 1N), replicating (S: 1 − <2N), replicated (G$_2$ + M: 2N) and over-replicated (>2N) DNA. (B) Quantification of the G$_1$ (single centrosome) and S/G$_2$/M/budding (2 centrosomes) populations. The TGME49_247040$^{AID\text{-}HA}$ expressing and TGME49_247040 deficient (4 h with 500 μM IAA) parasites were co-stained with α-Centrin1, α-TgIMC1, and DAPI. A hundred random vacuoles of the parasites were evaluated in three independent experiments. The mean −/+ SD values are plotted on the graphs. The unpaired two-sided *t* test value 0.0007. (C) Immunofluorescent microscopy analysis of RHΔ*Ku80TIR1* tachyzoites expressing (−IAA) or TgiRD1$^{AID\text{-}HA}$ deficient (+IAA, 4 h). To determine the number of centrosomes per internal bud, parasites were co-stained with α-Centrin1 (α-mouse IgG Fluor 488), α-TgIMC1 (α-rabbit IgG Fluor 568) antibodies and DAPI (blue). Representative images of the budding parasites from three independent experiments are shown. Internal buds with more than 1 centrosome are indicated with white arrows. (D) Quantification of the centrosomes per internal bud in the budding populations of RHΔ*Ku80TIR1* tachyzoites expressing (−IAA) or TgiRD1$^{AID\text{-}HA}$ deficient (+IAA, 4 h) based on IFA analysis in (C). A hundred random vacuoles of the parasites were evaluated in three independent experiments. The mean −/+ SD values are plotted on the graphs. The unpaired two-sided *t* test value 0.0004. (E) Schematics of the *T. gondii* perinuclear structures showing relative position and composition of the bipartite centrosome, centrocone and kinetochores. The drawing depicts one half of the mitotic figure. The dotted line indicates nuclear/cytoplasmic interface. dBC daughter basal complex. (F) The ultra-expansion microscopy analysis of RHΔ*Ku80TIR1* tachyzoites co-expressing TgiRD1$^{AID\text{-}HA}$ and TgCEP250-L1$^{myc}$. To visualize the outer core of the centrosome parasites were stained with α-Centrin1 (α-mouse IgG Fluor 568). To detect TgCEP250-L1 in the inner cores of the bipartite centrosome, tachyzoites were stained with α-myc (α-rabbit IgG Flour 488) antibodies. The non-treated tachyzoites (−IAA) retain tightly linked centrioles in mitosis, while the TgiRD1 deficiency (+IAA, 4 h) led to centrioles disengagement and reduplication. The experiment has been performed in three biological replicates. (G) Western blot analysis of the TgCEP250-L1$^{myc}$ expression in the TgiRD1 expressing (−IAA) and deficient (+IAA, 4 h) parasites. Western blots were probed with α-myc to identify TgCEP250-L1, α-HA to verify TgiRD1 downregulation and α-TubA to confirm equal loading of the total lysates. (H) The ultra-expansion microscopy images of RHΔ*Ku80TIR1* tachyzoites co-expressing TgiRD1$^{AID\text{-}HA}$ and TgCEP250-L1$^{myc}$. The inner core of the centrosome was detected with α-myc (α-rabbit IgG Flour 568) antibodies, the parasite surface with α-IMC1 (α-mouse IgG Fluor 488) and nucleus with DAPI stain. The inner core changes caused by TgiRD1 depletion (+IAA, 4 h) are indicated. The experiment has been performed in three biological replicates. (I) Western blot analysis of RHΔ*Ku80TIR1* TgiRD1$^{myc}$ protein extracts non-treated and treated with FAST-AP alkaline phosphatase. Western blots were probed with α-myc to identify TgiRD1 and α-GRA7 to confirm equal loading of the total lysates. Non-phosphorylated (blue arrow) and hyperphosphorylated (red arrow) TgiRD1 are indicated. (J) Western blot analysis of TgiRD1$^{myc}$ in RHΔ*Ku80TIR1* parasites expressing (−IAA) or deficient for TgCrk4$^{AID\text{-}HA}$ (+IAA, 4 h). Western blots were probed with α-HA antibodies to confirm TgCrk4 downregulation, α-myc antibodies to show accumulation of the hyperphosphorylated TgiRD1 in TgCrk4-deficient parasites (+IAA), and α-GRA7 to confirm equal loading of the total lysates. Non-phosphorylated (blue arrow) and hyperphosphorylated (red arrow) TgiRD1 are indicated. Source data are available online for this figure.

centrosome reduplication. Thus, we decided to name this factor TgiRD1 (inhibitor of Re-Duplication 1). Echoing the DNA rereplication defect, over 80% of budding TgiRD1-deficient parasites contained doubled number of the Centrin1-positive structures per internal bud (Fig. 7C,D). The ultra-expansion microscopy analysis identified these structures as disengaged centrioles, suggesting premature disengagement. In conventional eukaryotes, two centrioles, constituting the centrosome core, remain joined throughout most of the cell cycle and only briefly disengage in the G$_1$ phase, which allows templating and growth of the daughter centriole (centriole duplication) (Wang et al, 2014). Similarly, the *T. gondii* centrioles located in the outer core of the bipartite centrosome remain tightly linked during S-phase and mitosis (Fig. 7E,F, −IAA panel, red). However, centrioles in the TgiRD1-deficient tachyzoites at the similar cell cycle stage had drifted a distance from each other (Fig. 7F, +IAA panel, red). In some cases, the separation induced early duplication of the centrioles in mitotic parasites (Fig. 7F, +IAA panel, reduplicated centrioles). Previous studies showed that the outer centrosomal core promotes the assembly of internal buds (Suvorova et al, 2015). In good agreement with the prediction, tachyzoites lacking TgiRD1 formed multiple internal buds, suggesting that individual centrioles are equally efficient in supporting daughter scaffold assembly (Fig. EV6A).

To find out how the TgiRD1 deficiency affected the inner core of the bipartite *T. gondii* centrosome, we endogenously tagged the compartment marker TgCEP250-L1$^{myc}$ in the TgiRD1 AID line (Fig. 7E,F, green channel). Examination of the TgiRD1 depleted parasites revealed dramatic effect on the inner core of the centrosome. While normally dividing the S-phase and mitotic tachyzoites sequentially duplicated and segregated inner cores, the TgiRD1 deficient tachyzoites increased the number of the TgCEP250-L1 positive structures (Figs. 7F, green, H, red and EV6B,

red). To confirm the inner core amplification, we compared TgCEP250-L1 levels in parasites expressing or deficient in TgiRD1 (Figs. 7G and EV6E). Contrary to our predictions, we detected the substantial reduction of the full-length TgCEP250-L1 in the IAA-treated populations. Interestingly, the reduction of the full-length protein was accompanied by accumulation of the TgCEP250-L1 degradation products, suggesting that amplified TgCEP250-L1-positive structures in TgiRD1 deficient tachyzoites contained truncated TgCEP250-L1 (Fig. EV6E, intensity plots). We also examined whether TgiRD1 knockdown affected nuclear compartments linked to the bipartite centrosome (Fig. EV6C). To do so, we engineered a TgiRD1 AID line expressed inner kinetochore marker TgCenP-C$^{myc}$ and found that tachyzoites lacking TgiRD1 displayed normal dynamics of kinetochores segregation coordinated with centrocone expansion and resolution (TgMORN1 marker) (Fig. EV6D).

Altogether, our analysis of mitotic structures in the TgiRD1 deficient parasites revealed a pronounced effect of the TgiRD1 expression on the outer and inner cores of bipartite centrosome. We determined that TgiRD1 was required for tight connection of the centrioles pair after initial duplication in the G$_1$ phase. The observed TgiRD1-dependent proteolysis of TgCEP250-L1 implied that TgiRD1 also plays role in stabilization of the inner core of the centrosome. Together with the DNA replication defects, our results suggested that, together with TgCrk4-TgCyc4, TgiRD1 controls the "copy once per cell cycle" rule of the chromosome and centrosome duplication in *T. gondii*.

## TgiRD1 is regulated by cell cycle-dependent phosphorylation

Evaluation of TgiRD1 expression by western blot analysis revealed that, independent of the fused epitope, TgiRD1 migrated as two

prominent bands (Figs. 6E and 7I). The apparent molecular weight of the faster moving species corresponded to the predicted TgiRD1 molecular weight of 135 kDa, suggesting that the slower-moving band represented TgiRD1 containing post-translational modifications. Since phosphorylation affects protein mobility and conventional CDT1 is known to be regulated by different kinases, we predicted that the slower-moving TgiRD1 species was a phosphorylated form of the protein (Liu et al, 2004; Zhou et al, 2020). To test our prediction, we treated the total protein extracts with alkaline phosphatase. Confirming our predication, phosphatase treatment did not affect the lower molecular weight band, but eliminated the upper TgiRD1 band, which validated the existence of a hyperphosphorylated form of TgiRD1 (Fig. 7I).

Our proteomics studies detected TgiRD1 as a dominant interactor of the $G_2$ complex TgCrk4-TgCyc4 (Fig. 1E). To find out if TgCrk4 had any effect on the TgiRD1 expression or localization, we introduced TgiRD1$^{myc}$ into TgCrk4 AID transgenic parasites. A TgCrk4-dependent $G_2$ block did not significantly affect TgiRD1 expression or localization (Fig. EV6F). Metazoans use a combination of hyper-phosphorylation and Geminin inhibition to strategically block CDT1-dependent DNA replication licensing during $G_2$ phase (Zhou et al, 2020). To find out if TgiRD1 phosphorylation state changes in parasites arrested in $G_2$, we analyzed the state of TgiRD1 in both asynchronously dividing and $G_2$-arrested parasites (Fig. 7J). $G_2$ arrest induced by TgCrk4 depletion led to accumulation of hyperphosphorylated TgiRD1, confirming cell cycle-dependent regulation of TgiRD1 by posttranslational modification. Since hyperphosphorylated TgiRD1 accumulated in TgCrk4-deficient parasites, it is likely that kinases other than TgCrk4 contribute to TgiRD1 hyper-phosphorylation.

# Discussion

In this study, we made several critical discoveries regarding the apicomplexan cell cycle. First, we identified the previously unrecognized $G_2$ period which operates during apicomplexan endodyogeny. Second, we showed that apicomplexan $G_2$ phase is regulated by the Crk4–Cyc4 complex that exists only in parasites dividing in a binary fashion. Third, we determined that the TgCrk4–TgCyc4 complex interacts with TgiRD1 factor orthologous to DNA replication licensing factor CDT1. We also presented evidence that TgiRD1 plays no role in licensing of DNA replication in $G_1$ phase, and primarily controls centrosome and DNA reduplication in $G_2$ phase. Our findings support the cooperative action between TgCrk4–TgCyc4 complex and TgiRD1 to repress multinuclear division, which is likely the default cell cycle mechanism in Apicomplexa phylum. Thus, contrary to the common view of *T. gondii* endodyogeny as the simplest mode of apicomplexan division, our findings strongly argue that endodyogeny has more complex regulation of the cell cycle and had instead evolved additional mechanisms to suppress DNA rereplication cycles.

Although rare, the multinuclear division is not a novelty in eukaryotes. The best-known example of multinuclear division in higher eukaryotes is in the embryonic development of *Drosophila melanogaster* (Yuan et al, 2016). The first 13 cycles of DNA amplification in the fruit fly zygote consists of S-phase (chromosome replication) and mitosis (chromosome segregation), repeating

to produce a single cell with 500 nuclei, which is called a blastoderm. The binary division is then introduced with the $G_2$ phase, which extends the cell cycle and allows the time for the first cytokinetic event to occur and separate these nuclei by cell membranes. Thus, one of the critical functions of $G_2$ is to repress multinuclear division by enforcing the "once per cell cycle duplication" rule that minimizes the chances of losing or altering genetic information due to chromosome mis-segregation. *Drosophila* development is somewhat reminiscent of cell division in apicomplexan parasites. Given the fact that only a fraction of apicomplexans divides by a binary mode, and that the vast majority undergo multinuclear replication, it is tempting to suggest that multinuclear division is the ancestral mechanism of apicomplexan cell division, with binary division evolving later to accommodate the specific needs of such apicomplexans genera as *Toxoplasma*, *Hammondia*, *Neospora*, *Besnoitia*, and *Babesia* (Gubbels et al, 2020). In support of this proposition, reflecting the additional levels of cell cycle regulation, *T. gondii* maintains an extended repertoire of cell cycle machinery to predominantly divide into two (Alvarez and Suvorova, 2017; White and Suvorova, 2018). What mechanism would apicomplexan parasites evolve to convert from multinuclear to binary division? The zygotic fruit fly example points toward the possibility that some apicomplexans evolved a mimic $G_2$ period by installing mechanisms to repress relicensing of DNA replication and centrosome duplication and to advance cytokinesis (budding). This could be achieved by connecting replicated chromosomes to the cytoskeleton of future daughter buds using a modular centrosome such as the bipartite *T. gondii* centrosome (Suvorova et al, 2015). The inner centrosomal core is functionally linked to chromosome segregation and the outer core is involved in budding, and, during endodyogeny, they are permanently associated with one another (Chen and Gubbels, 2019; Suvorova et al, 2015; Tomasina et al, 2022). It has been shown that separating centrosomal cores leads to mitotic death in tachyzoites (Chen and Gubbels, 2019; Suvorova et al, 2015).

Characterizing the processes regulated by TgCrk4 led to the construction of the $G_2$ network in *T. gondii*, which includes its immediate effectors, direct interactors, and cellular pathways that uniquely respond to the $G_2$/M transition (Fig. 8A). We identified the families of conventional $G_2$ regulators, such as components of the origin recognition complex (TgORC4) and the anaphase-promoting complex APC/C (Cdc20). However, the immediate effectors differed from known $G_2$ regulators, and often included parasite-specific groups of proteins. For example, gene expression seems to be directly regulated by the phosphorylation of a parasite-specific AP2X-9 DNA binding factor. Control of centrosome reduplication involves the direct phosphorylation of the centrosomal protein TgCEP530, Ca$^{++}$-dependent kinase TgCDPK7, as well as interaction with a highly deviated CDT1 ortholog TgiRD1 (Courjol and Gissot, 2018; Morlon-Guyot et al, 2014). Although their direct effectors are not yet known, progression through the $G_2$/M checkpoint has a significant effect on the IMC components of daughter buds (TgIMC16, TgIMC30, TgIMC33, TgIMC34, TgIMC36), Apicomplexa-specific components of kinetochores (TgAKiT1, TgAKiT6), and the bipartite centrosome (TgCEP250-L1, TgCEP250, TgCEP530) (Back et al, 2023; Brusini et al, 2022; Butler et al, 2014; Chen and Gubbels, 2019; Courjol and Gissot, 2018; Dos Santos Pacheco et al, 2021; Suvorova et al, 2015; Tomasina et al, 2022). Importantly, this novel $G_2$ network

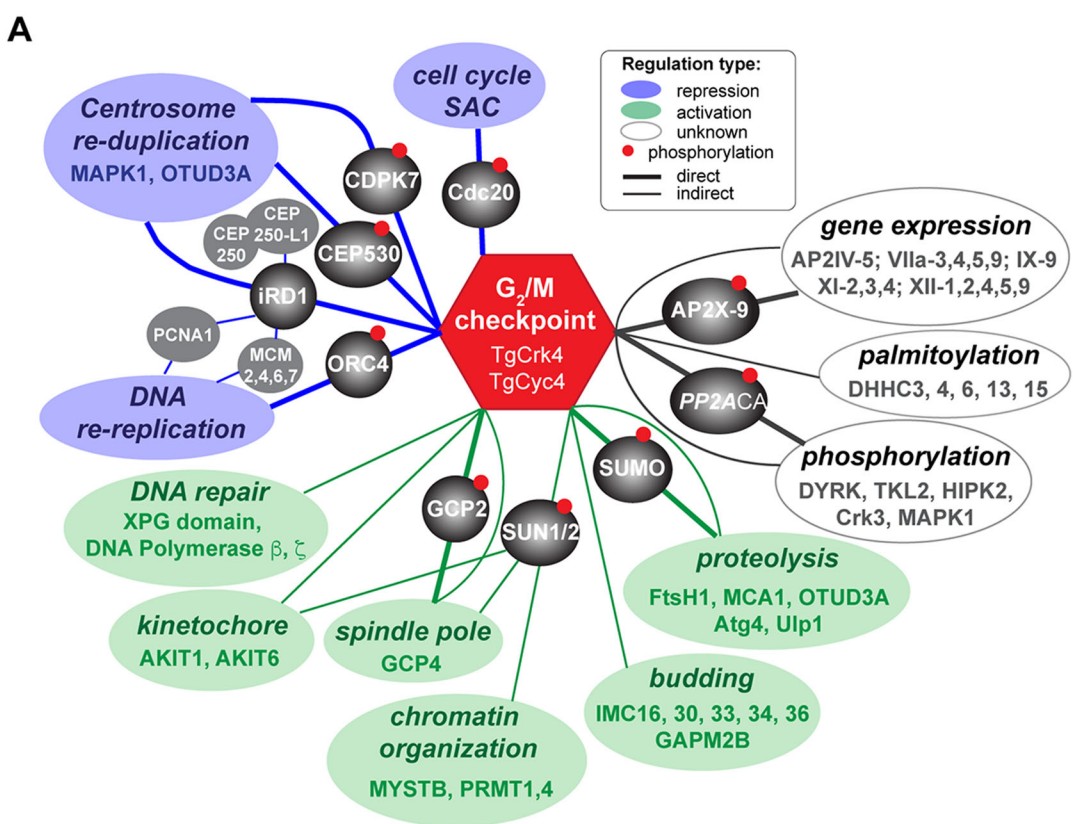

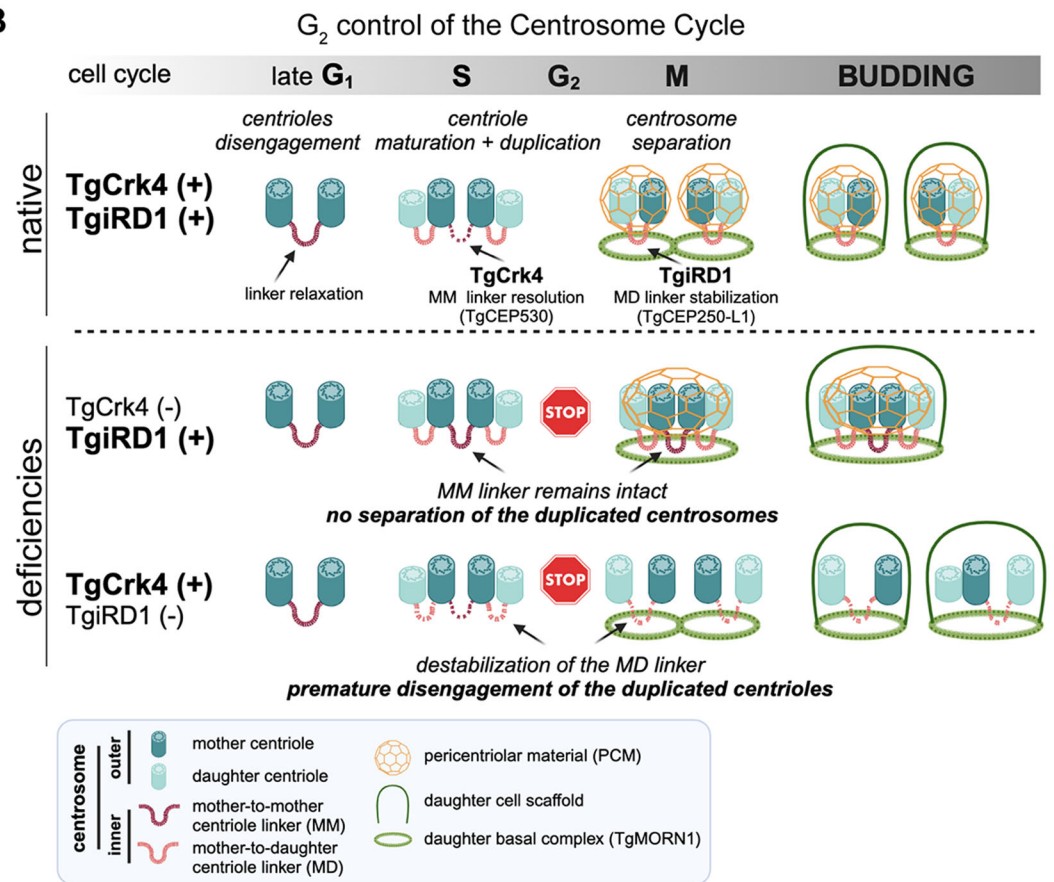

◄

**Figure 8.  The mechanisms that regulate G$_2$ phase in *T. gondii* endodyogeny.**

(A) The summary of the *Toxoplasma* G$_2$/M network identified in this study. The novel TgCrk4-TgCyc4 complex regulates *T. gondii* G$_2$/M checkpoint, shown as the red STOP sign. The affected cellular processes are indicated by ovals. TgCrk4-dependent phosphorylation of TgCDPK7, TgORC4 and TgCEP530 likely represses centrosome reduplication and chromosome rereplication (blue ovals), which also includes dominant interaction with CDT1-related inhibitor of DNA and centrosome reduplication TgiRD1. Activation of the spindle pole assembly and targeted proteolysis is mediated by phosphorylation of TgGCP2 and TgSUMO protease, respectively (green ovals). The G$_2$/M transition also controls APC/C activation via direct phosphorylation of TgCdc20 (blue oval), while regulators of the kinetochore assembly and concurrent assembly of internal buds are currently unknown (green ovals). The G$_2$ phase has a major effect on global protein phosphorylation and gene expression, likely via direct phosphorylation of TgPP2ACA and TgAP2X-9. (B) Putative mechanism of regulation of the *Toxoplasma* centrosome cycle. The upper panel illustrates the normal centrosome dynamics during tachyzoite cell cycle, which progression is indicated on the top gray bar. The cartoon shows two cores of the bipartite centrosome and the components of the daughter cell scaffold. The outer core of the centrosome contains two parallel centrioles. We propose that the inner core structurally resembles a proteinaceous fiber that connects centrioles in the outer core and is made of coil-coiled domain-containing proteins of the CEP family. Major steps of the centrosome cycle include centriole disengagement in G$_1$, duplication and maturation in S-phase, and centrosome separation at G$_2$/M transition. The changes to the outer and inner core of the bipartite centrosome caused by TgCrk4 and TgiRD1 deficiency are shown on the two bottom panels. The STOP sign indicates a position of the TgCrk4-regulated G$_2$/M checkpoint.

incorporates factors that were previously shown to induce reduplication of centrosomes and nuclear DNA. Among them are the deubiquitinase TgOTUD3A and centrosomal kinase TgMAPK1 (Dhara et al, 2017; Suvorova et al, 2015).

One of the unexpected findings from our studies is the dominant interaction of TgCrk4-TgCyc4 complex with an ortholog of the DNA replication licensing factor CDT1, TgiRD1. Eukaryotic CDT1 is a multifunctional protein that plays distinctive roles in nearly all cell cycle phases (Pozo and Cook, 2016). CDT1 activates pre-replication complexes in G$_1$ phase, is actively removed from the S-phase nucleus, and kept off the chromatin in G$_2$ phase to avoid chromosome rereplication. CDT1 also plays a role in kinetochore assembly by securing the connection of Ndc80 to spindle fibers in mitosis, thereby assisting in chromosome segregation (Varma et al, 2012). Our study found substantial differences between eukaryotic CDT1 and phylogenetically related *T. gondii* iRD1 protein. First, TgiRD1 was detected in the nucleus of S, G$_2$, and mitotic parasites, which is opposite to temporal expression of CDT1 in mammalian cells and yeast. Second, as a dominant partner of the TgCrk4-TgCyc4 complex, TgiRD1 played its primary roles in G$_2$ and mitosis. In this aspect, the TgCrk4-TgCyc4 complex operates in the manner analogous to the mammalian Cdk2-Cyclin-A complex. However, the Cdk2-Cyclin-A complex is dispensable for somatic cell division, while TgCrk4 and TgCyc4 are independently required for tachyzoite survival (Honda et al, 2005). The functional resemblance of TgCrk4-TgCyc4 and Cdk2-Cyclin-A complexes was also supported by the structural similarities between the cyclin partners. Third, we found that depleting TgiRD1 did not block the licensing of DNA replication in G$_1$ phase, suggesting alternative mechanisms of regulation of DNA replication in parasites. The fact that TgiRD1 is absent during primetime of the CDT1 function in conventional eukaryotes (late mitosis and half of G$_1$ phase) further supports the minimal involvement of TgiRD1 in licensing DNA replication in G$_1$ phase. Furthermore, contradicting its expected role as a licensing factor, CDT1-related TgiRD1 was constitutively present in the nucleus while DNA replication was ongoing (S-phase). The changed temporal expression and primary role in DNA and centrosome reduplication suggested that TgiRD1 may function similar to CDT1 inhibitor, Geminin. Coincidentally, E3 ligase APC/C controlled the Geminin removal in late mitosis and TgiRD1 was predicted to have APC/C degrons. Further studies are needed to determine what influenced dramatic differences in function and regulation of the major DNA replication licensing factor CDT1 in apicomplexan parasites.

How does the TgiRD1 factor contribute to the G$_2$ phase progression? Based on our collective findings we proposed a model of the centrosome cycle regulation by TgiRD1 and G$_2$ kinase TgCrk4 during *T. gondii* endodyogeny. The centrosome cycle consists of sequentially executed steps of the centrioles disengagement (G$_1$), centriole duplication and maturation (S-phase), and separation of the duplicated centrosomes each containing a pair of centrioles (G$_2$ and mitosis) (Fig. 8B). The cell cycle-dependent assembly and resolution of the linker controls centrioles connection (Wang et al, 2014). In the conventional cell cycle, a single linker connecting centrioles is assembled in G$_1$ and is resolved in G$_2$ phase to allow centrosome separation. The orthogonal position of the daughter centriole likely inhibits early disengagement from the mother centriole in mitosis. However, *Toxoplasma* centrioles lay parallel and may require additional connectors to hold mother and newly assembled daughter centriole during S, G$_2$ phases and mitosis, thus, preventing an unlawful centrioles reduplication (Francia and Striepen, 2014). According to our model, unlike a conventional centrosome, the *Toxoplasma* centrosome uses two connectors that differentiate a mother–mother and a mother–daughter pair of centrioles (Fig. 8B). One linker, inherited from the previous cycle, connects two mother centrioles in G$_1$, S, and G$_2$ phases. The other linker, formed in S-phase, holds mother and daughter centrioles until the cell division is complete. The inheritance of the centrioles from the previous cycle suggests that the mother–daughter linker matures into a mother–mother linker as the parasite enters the new cell cycle. Judging by the joined centrosome phenotype, TgCrk4 likely regulates resolution of the mother–mother centriole linker, possibly by direct phosphorylation of TgCEP530 located in the uncharacterized intermediate centrosomal core (Courjol and Gissot, 2018). Previous studies implicated mitotic kinase TgNek1/2 in the process of centrosome separation (Chen and Gubbels, 2013). Since we could not detect TgNek1/2 in the G$_2$ network, this kinase likely acts upstream of TgCrk4. Contrary to TgCrk4 knockdown phenotype, TgiRD1 deficient tachyzoites prematurely disengaged centrioles in G$_2$ and mitotic parasites, suggesting that TgiRD1 regulates the mother–daughter centriole linker. The likely mechanism involves stabilization of another coil-coiled domain protein, TgCEP250-L1 (Suvorova et al, 2015; Tomasina et al, 2022). Our model implies that differential protein composition of the mother–mother and mother–daughter centriole linkers facilitates cell cycle-dependent processing of individual connectors. For instance, the presence of TgCEP530 in

the mother–mother and not mother–daughter connector can make the mother–mother linker accessible for the TgCrk4-dependent resolution in the $G_2$ phase. It drives the centrosome separation but preserves the connection of the centrioles within individual centrosomes. On the other hand, the alternative TgCEP250-L1 state in the connectors may explain selective protection of the mother–daughter and not mother–mother linker by TgiRD1 in $G_2$ phase. Supporting our model, it has been recently shown that in mitotic parasites, the C-Nap related protein TgCEP250 plays a role in connecting the outer core (centrioles) and inner core (linker) (Chen and Gubbels, 2019). Similar to TgCEP250-L1, TgCEP250 undergoes proteolytic maturation suggesting co-expression of the factors in a centriole linker. Further studies are required to dissect the mechanisms connecting parallel *T. gondii* centrioles. Comprehensive analysis of the *T. gondii* centrosome cycle will answer outstanding question of the apicomplexan biology such as why only a small group of apicomplexans preserved centrioles, why the parallel orientation is favored, and whether the centriole retention benefits the binary type of cell division.

Does the $G_2$ phase unique for endodyogeny or it also operates within the multinuclear divisions of apicomplexans? Previous studies of apicomplexan schizogony showed that chromosome replication always followed by chromosomes segregation, even when karyokinesis was delayed (Gubbels et al, 2020; Klaus et al, 2022; McDonald and Merrick, 2022; Vaishnava et al, 2005). It suggests that the $G_2$ functions such as repression of DNA and centrosome reduplication are likely active in various types of apicomplexan cell cycles. Indeed, the TgiRD1 orthologs were present in all the annotated apicomplexan genomes. One of the favorable hypotheses is that TgiRD1 supports a core $G_2$ mechanism that controls premature separation of the duplicated microtubule organizing centers (centrosomes or centriolar plaques) in all the apicomplexans, while the Crk4–Cyc4 complex had evolved to link the duplicated genomes to the daughter scaffolds in the binary dividing parasites. The future studies will determine whether the $G_2$ phase operates in the other branches of Apicomplexa phylum, what role the TgiRD1 orthologs play in multinuclear cell division and how apicomplexan parasites segregate the initiation and relicensing of DNA replication.

# Methods

## Parasite cell culture and plaque assay

*T. gondii* RH *ΔKU80Δhxgprt TIR1* and RH *TatiΔKU80* strains were grown in Human Foreskin Fibroblasts (HFF) (ATCC, SCRC-1041) in DMEM media (Millipore Sigma) supplemented with 10% FBS (Brown et al, 2017; Sheiner et al, 2011). Cultures were tested for mycoplasma contamination using Mycoplasma Detection kit (MP Biomedicals). The viability of transgenic strains was measured by plaque assays in which monolayers of HFF cells were infected with 50 parasites per well. Cultures were treated with either a 95% EtOH (control), or 500 μM Indole-3-acetic-acid (IAA or auxin), or 1 μg/mL of anhydrotetracycline (ATc) to trigger protein degradation or transcriptional downregulation, respectively. Plaques developed after 7 days of growth at 37 °C were fixed, stained, and counted. Three biological replicates of each assay were performed.

## Phylogenetic analysis

Protein sequences were downloaded from UniProt (www.uniprot.org) database and analyzed using NGPhylogeny (www.NGPhylogeny.fr.) custom workflow (Lemoine et al, 2019; UniProt, 2021). Proteins were aligned with MUSCLE algorithm, curated with Block Mapping and Gathering with Entropy (BMGE) (Criscuolo and Gribaldo, 2010; Edgar, 2004). Maximum likelihood-based inference with Smart Model Selection was implemented using PhyML+SMS algorithm (Guindon et al, 2010). We applied 100 bootstraps to test optimization of the tree. Phylogenetic trees were constructed using Newick utilities (Junier and Zdobnov, 2010). Amino acid sequences of the analyzed proteins are listed in Dataset EV2.

## Construction of transgenic strains

All transgenic strains created in this study and used primers are listed in Dataset EV1. Targeting constructs were verified by PCR using gene- and epitope tag-specific primers, ensuring that tags were incorporated at the appropriate genomic locus.

### Endogenous C-terminal tagging and conditional expression
To create conditional expression models for TgCrk4 and TgiRD1, we amplified a fragment of the 3'-end of the gene of interest via PCR and cloned it into the pLIC-mAID-3xHA-hxgprt, and/or pLIC-10xHA-CmR, vectors digested with PacI endonuclease using Gibson Assembly method. In total, 50 μg DNA of the resulting constructs were linearized with endonuclease within the gene fragment and transfected using Amaxa Nucleofector II device (Lonza) into RH *TIRΔKU80Δhxgprt* and RH *TatiΔKU80Δhxgprt* parents, respectively, with 100 μl Cytomix buffer supplemented with 20 mM ATP and 50 mM reduced glutathione. The same approach was applied in building myc-epitope tagged TgCrk4, TgiRD1, and TgCEP250-L1. Genomic fragments were amplified then cloned into pLIC-3xMyc-DHFR and/or pLIC-10x-Myc-CmR, linearized, and transfected into respective parental strains. Dual transgenic strains were developed in the same fashion and sequentially electroporated into parental transgenic strains and selected for with alternative drugs. All transgenic parasites were incubated for 24 h in regular growth media prior to drug selection, limited dilutions were used to obtain individual clones and incorporation was tested using PCR and IFA analysis.

### Endogenous N-terminal tagging and conditional expression
To build tet-OFF mutant TgCyc4 a fragment of the 5'-end of the gene was amplified (Dataset EV1 lists primers used) and digested with BglII/NotI for compatible ends and ligated into the promoter replacement vector pTetO7sag4-3xHA-DHFR (Alvarez and Suvorova, 2017). Constructs were linearized with endonuclease within the gene fragment and electroporated into RH *TatiΔKu80* strain (Sheiner et al, 2011).

## Immunofluorescence microscopy analysis

All the immunofluorescence microscopy analyses had been performed in more than three independent experiments. Monolayers of HFF cells were grown on coverslips and infected with parasites under indicated conditions. Cells were fixed with 4%

paraformaldehyde, permeabilized with 0.5% Triton TX-100, blocked with a 3% bovine serum albumin solution, and incubated consecutively with primary and secondary antibodies. Primary antibodies used: rat α-HA (3F10, Roche Applied Sciences), mouse α-Centrin1 (clone 20H5; Millipore Sigma), mouse α-Atrx1 (kindly provided by Dr. Peter Bradley, UCLA, CA), rabbit α-Myc (clone 71D10, Cell Signaling), rabbit α-MORN1 (kindly provided by Dr. Marc-Jan Gubbels, Boston College, MA), mouse and rabbit α-IMC1 (kindly provided by Dr. Gary Ward, University of Vermont, VA), mouse α-Tubulin A (kindly provided by Dr. Jacek Gaertig, University of Georgia, Athens, GA), and mouse α-Tubulin A (clone 12G10; DSHB). Alexa-conjugated secondary antibodies of different emission wavelengths (Thermo Fisher) were used at 1:500 dilution, and nucleic acids were stained with 4',6-diamidino-2-phenylindole (DAPI, Sigma). Stained parasites on the coverslips were mounted using Immu-mount (Fisher), dried overnight at 4 °C, and viewed on a Zeiss Axiovert Observer Microscope equipped with a 100× objective and ApoTome slicer. Collected images were processed in Zen2.2 and Adobe Photoshop 2020 using linear adjustment when needed. All the experiments were performed in triplicate. To minimize the effect of subjective bias, the qIFA replicates were quantified by different investigators.

## Ultra-expansion microscopy

The tachyzoite internal structures were examined by ultra-expansion previously described (Engelberg et al, 2022). Briefly, tachyzoites growing in HFF cells were fixed in with 4% PFA, incubated in 2% formaldehyde, 1.4% acrylamide (AA) in PBS for 5 h at 37 °C. The sample was converted into a gel by incubation in 19% (w/w) sodium acrylate, 10% (w/w) AA and 0.1% (w/w) BIS-AA in PBS supplemented with ammonium persulfate and TEMED (tetramethylethylenediamine) for 1 h at 37 °C, followed by incubation in denaturation buffer (200 mM SDS, 200 mM NaCl, 50 mM Tris, pH 9) at 95 °C for 90 min. Gels were expanded in ddH$_2$O overnight and washed twice in PBS on the next day. Primary and secondary antibody incubation was performed for 3 h at 37 °C. The primary antibodies were used at the following dilution: mouse α-Tubulin A (12G10, DSHB; 1:250), mouse α-myc (9B11, Cell Signaling; 1:500), rabbit α-myc (71D10, Cell Signaling 1:250), rabbit α-IMC1 (kindly provided by Dr. Gary Ward, University of Vermont, VA; 1:250), rabbit α-MORN1 (kindly provided by Dr. Marc-Jan Gubbels, Boston College, MA; 1:100) and mouse α-Centrin1 (clone 20H5; Millipore Sigma; 1:100). Stained gels were washed three times with PBST (1×PBS + 0.1% Tween20) before a second overnight expansion in ddH$_2$O.

## Western blot analysis

Infected HFF monolayers were lysed in Laemmli sample buffer, sonicated, and heated at 95 °C for 10 min. Alternatively, lysates were prepared from purified parasites. After separation on SDS-PAGE gels, proteins were transferred onto nitrocellulose membranes and probed with rat α-HA (3F10, Roche), rabbit α-Myc (Cell Signaling), mouse α-GRA7 (kindly provided by Dr. Peter Bradley, UCLA CA) and mouse α-Tubulin A (12G10; kindly provided by Dr. Jacek Gaertig, University of Georgia, Athens GA). After incubation with species-specific secondary horseradish peroxidase (HRP)-conjugated antibodies (Jackson ImmunoResearch), proteins

were visualized by enhanced chemiluminescence detection (MilliporeSigma Immobilon). For western blot analysis of TgiRD1 phosphorylation parasites were collected, washed in PBS with protease and Halt™ phosphatase inhibitor cocktail, resuspended in PBS and treated with FAST AP according to the manufacturer recommendation (NEB).

## FACS scan analysis

Parasite DNA content was evaluated by flow cytometry using propidium iodide staining of tachyzoites DNA. HFF Monolayers were infected at 1:1 MOI and left to grow overnight at 37 °C. The next day, TgCrk4$^{AID-HA}$ parasites were treated with 500 μM Auxin for 0, 4 h, with 1 h and 2-h recovery periods after auxin was removed, and TgiRD1$^{AID-HA}$ parasites were treated with 0, 4, and 8 h of 500 μM Auxin. Parasites were collected by scraping, passing 5 times through an 18 G needle, then filtering with a 3-μm filter. Parasites were pelleted by centrifugation at 1800 rcf for 15 min. Media was aspirated, and parasites were resuspended in 300 μL of cold PBS with 700 μL of ice cold 100% molecular grade ethanol added while being vortexed. Samples were then stored at −20 °C until day of FACS scan analysis. Day of, parasites were pelleted at 3000 rcf for 20 min, ethanol was aspirated, and pellet was resuspended in 900 mL of 50 μM Tris solution pH 7.5. In all, 100 μL of 2 mg/mL propidium iodide (Fisher) for a final concentration of 0.2 mg/mL, and 20 μL of 10 mg/mL RNase A for a total concentration of 0.2 mg/mL. Samples were left to incubate in the dark at room temperature for 30 min. DNA content was measured based on the intensity of the emission from PI-stained DNA using the PE-Texas Red-A laser. Parameters for the specific size of *Toxoplasma gondii* was used to determine single cells from debris and non-single cells, and percentages of each cell cycle phase were calculated based on the defined gates for each population.

## Proteomics analysis

### The immunoprecipitation sample preparation
To analyze TgCrk4, TgCyc4, and TgiRD1 complexes, the samples were prepared from 10$^9$ parental (RHΔKu80Δhxgprt TIR1 or RH Tati ΔKu80) and tachyzoites expressing endogenously tagged TgCrk4$^{AID-HA}$, $^{HA}$TgCyc4 or TgiRD1$^{myc}$ as previously described (Hawkins et al, 2022). Parasites were collected by filtration and centrifugation. Extracted proteins were incubated with α-HA or α-myc magnetic beads (MblBio). The efficiency of immune precipitation was verified by western blot analysis. Protein samples were then processed for mass-spectrometry-based proteomic analysis as previously described (HaileMariam et al, 2018; Zougman et al, 2014).

### The global proteome analysis sample preparation and mass spectrometry
Samples for TgCrk4 global proteome were prepared and processed as previously described (Hawkins et al, 2022). Briefly, tachyzoites were collected and solubilized with 5% SDS (w:v) in 50 mM TEAB. Equal amounts of protein (200μg) was fractionated and were processed for LC-MS/MS using s-traps (Protifi). Phosphopeptides of each fraction were enriched on the TiO$_2$ nanopolymer beads (Tymora Analytical), eluted, dried and resuspended in H$_2$O/1% acetonitrile/0.1% formic acid for LC-MS/MS analysis. Peptides

were analyzed on a hybrid quadrupole-Orbitrap instrument (Q Exactive Plus, Thermo Fisher Scientific). Full MS survey scans were acquired at 70,000 resolution. The top ten most abundant ions were selected for MS/MS analysis.

### Raw data processing

Files were processed in MaxQuant (v 1.6.14.0, www.maxquant.org) and searched against the current Uniprot *Toxoplasma gondii Me49* protein sequences database. Search parameters included constant modification of cysteine by carbamidomethylation and the variable modifications, methionine oxidation, protein N-term acetylation, and phosphorylation of serine, threonine, and tyrosine. Proteins were identified using the filtering criteria of 1% protein and peptide false discovery rate. Protein intensity values were normalized using the MaxQuant LFQ function (Cox et al, 2014).

Label-free quantitation analysis of the global proteome and phosphoproteome was performed using Perseus (v 1.6.14.0), software developed for the analysis of omics data (Tyanova et al, 2016). LFQ Intensity values were Log2-transformed, and then filtered to include proteins containing at least 60% valid values (reported LFQ intensities) in at least one experimental group. Then, the missing values in the filtered dataset were replaced using the imputation function in Perseus with default parameters (Tyanova et al, 2016). Statistical analyses were carried out using the filtered and imputed protein groups file. Statistically significant changes in protein abundance are determined using Welch's *t* test *P* values and z-scores.

## Statistical analysis

### Significance analysis of interactome (SAINT)

The computational platform SAINT (Significance Analysis of Interactome) had been applied to examine protein-protein interactions detected by mass-spectrometry analysis of TgCrk4, TgCyc4, and TgiRD1 complexes (Choi et al, 2011). The platform utilizes spectral count data, applies a probabilistic model to gauge the likelihood of authentic interactions between proteins. Specifically, we had used the SAINT express-spc command in version 3.6.3 with default parameters. The hits with the cut-off range 0.5–1 were considered significant.

### Global protein expression and phosphorylation changes

To determine a relative abundance of the phosphorylated peptides, each phosphorylation intensity value was normalized to intensity of corresponding protein expression (global proteome). We then introduced an improved Gaussian model to identify proteins or phosphorylated peptides that were significantly altered during checkpoint block induced by TgCrk4 degradation (4 h with auxin) (Jorge et al, 2009; Li et al, 2019; Wang et al, 2011; Zhang et al, 2010). In brief, a Gaussian distribution was implemented to model the log2 fold change (log2FC) of normalized protein/peptide intensities. Using a sliding window from low to high of average protein/peptide intensity, we modeled the variance changing of log2FC and selected proteins/peptides within a specific range of intensity. For each replicate, the maximum likelihood estimation (MLE) was used to estimate the parameters in Gaussian distribution from each sliding window after removing entities with log2FC higher than the top 95% or lower than the bottom 5%. The *P* value was calculated as the probability of the fitted Gaussian distribution

higher or lower than the observed log2FC value when log2FC was higher or lower than the fitted expectation. The false discovery rate (FDR) was used to adjust *P* value. We selected significantly changed entities based on the criteria that log2FC > 1 and adjusted *P* value < 0.1, or log2FC < −1 and adjusted *P* value < 0.1 consistently in all the replicates.

### Heatmap construction

To capture phosphorylation pattern changes due to TgCrk4 deficiency, we compared pairwise significantly changed phosphorylated peptides detected in untreated, auxin treated for 30 min, or 4 h parasites. We started with building *K*-means classification for the 0/30 min pair. Akaike information criterion (AIC) and BIC were used to search optimized category numbers of *K*-means classification. The normalized intensity of each selected peptide was further divided by the maximum intensity of this peptide from checkpoint block time to 4 h, which is represented as "Proportion-of-max" in heatmap. The enriched GO terms for each group of genes are identified by using a hypergeometric test compared with whole *T. gondii* genome with FDR adjusted *P* value lower than 0.1.

### Phosphorylation motif search

The featured motifs of phosphorylated peptides were analyzed by using motif-x algorithm Soft MoMo v5.1.1 (http://meme-suite.org/tools/momo) (Bailey et al, 2009). For analysis, we selected 13-mer phosphorylated peptides with phosphor-modified amino acid residue positioned between 6 residues upstream and downstream of the phosphorylation site. In cases where flanking restudies were missing in the MS identified peptide, the neighboring sequence was extracted from the predicted protein sequence in *T. gondii* genome (ToxoDB). The background or control datasets were selected according to the type of amino acids phosphorylated in query data. The motif sequence was considered when the minimum number of occurrences was over 25 and *P* value < $1e^{-9}$.

### T test was performed in Microsoft Excel

The results are included in the corresponding worksheets of the Dataset EV4 and Source Data files.

## Structural analysis

Using UniProt database, amino acid sequences were downloaded and used to build protein homology models in the SWISS-MODEL suite (Waterhouse et al, 2018). The resulting models were further analyzed in PyMol (www.pymol.org). Available images of the folded proteins were downloaded from AlphaFold2. Alignment of the protein sequences downloaded from UniProt database was performed using MUSCLE algorithm and exported to Jalview (Waterhouse et al, 2009).

# Data availability

The mass-spectrometry proteomics data have been deposited to the ProteomeXchange Consortium via the PRIDE (Perez-Riverol et al, 2022) partner repository (https://www.ebi.ac.uk/pride/) with the dataset identifiers: PXD050478 (TgCrk4 global proteome and phosphoproteome), PXD044592 (TgCrk4 interactome), PXD043994 (TgCyc4 interactome), PXD043924 (TgiRD1

interactome). Processed proteomics data are available in Supplementary Datasets S3, S5, S6, S7, and S8.

## Peer review information

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

## Acknowledgements

The authors want to thank USF flow cytometry and USF-CAS Proteomics cores, and USF Omics Hub for assistance with the DNA FACScan analysis, proteomics efforts and computational support. Dr. Daria Naumova for her valuable suggestions and manuscript editing. The authors thank Dr. Klemens Engelberg (Boston College) and Dr. Chunlin Yang (Indiana University) for sharing protocols and advice on U-ExM technology. The authors thank members of the VEuPathDB team for helpful resources in preparing this manuscript. This project was supported by a grant from the National Institutes of Health to ES (1R21AI178797).

## Author contributions

**Lauren M Hawkins**: Resources; Data curation; Validation; Investigation; Methodology; Writing—review and editing. **Chengqi Wang**: Resources; Data curation; Software; Formal analysis; Validation; Methodology; Writing—review and editing. **Dale Chaput**: Resources; Data curation; Software; Formal analysis; Investigation; Methodology. **Mrinalini Batra**: Data curation; Formal analysis; Validation; Investigation; Methodology; Writing—review and editing. **Clem Marsilia**: Resources; Data curation; Software; Formal analysis; Validation; Investigation; Methodology; Writing—review and editing. **Danya Awshah**: Formal analysis; Validation; Investigation; Methodology. **Elena S Suvorova**: Conceptualization; Resources; Data curation; Software; Formal analysis; Supervision; Funding acquisition; Validation; Investigation; Visualization; Methodology; Writing—original draft; Project administration; Writing—review and editing.

## Disclosure and competing interests statement

The authors declare no competing interests.

# Expanded View Figures

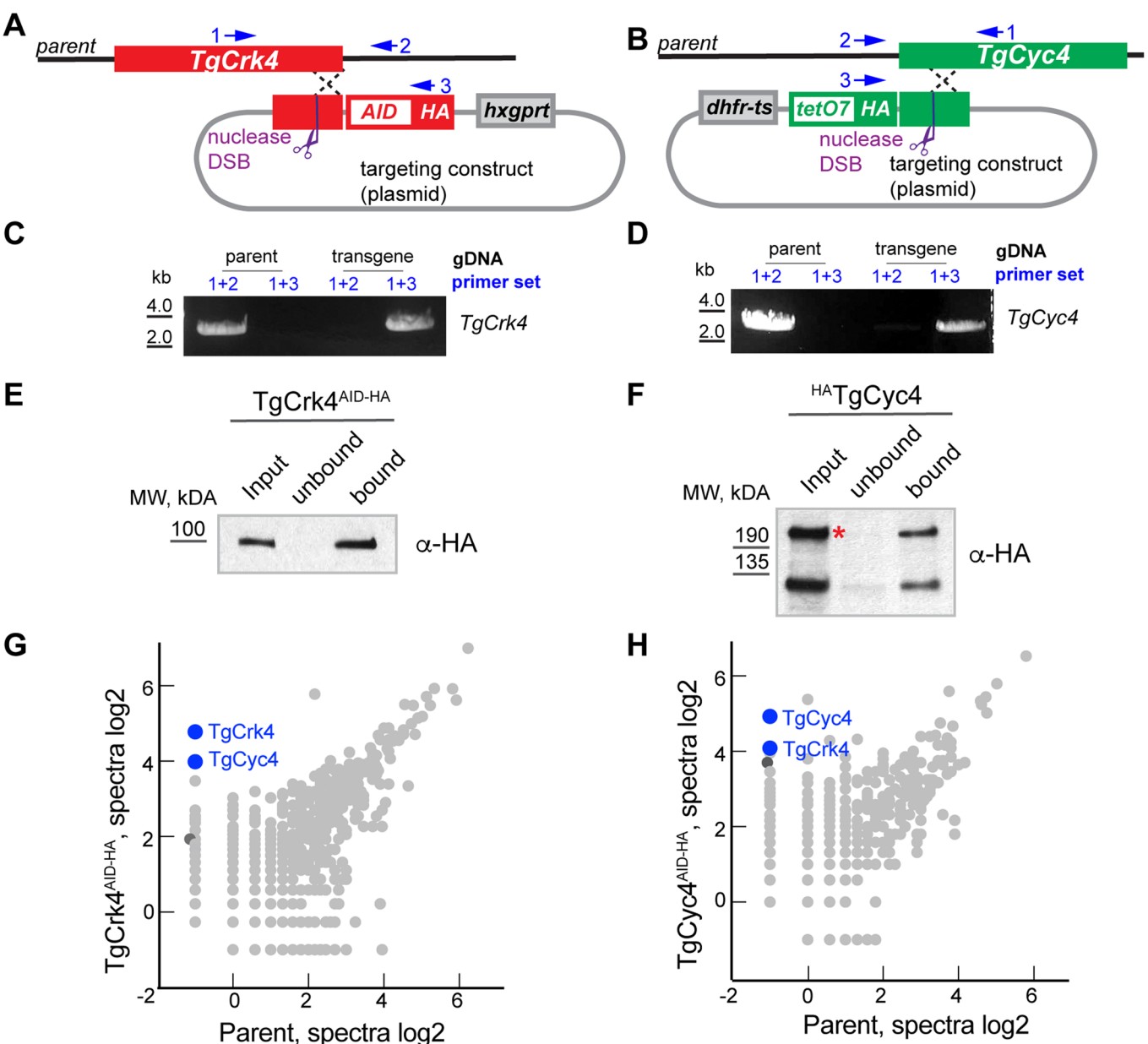

**Figure EV1.  Generation and analysis of conditional expression models for TgCrk4 and TgCyc4.**

(**A**) Schematics for constructing TgCrk4 AID-modified gene. Targeting plasmid included 3′ fragment of *TgCrk4* genomic locus fused with encoded sequence for mini-version of AID (mAID), 3xHA (HA) epitopes and the drug-selection marker *hxgprt* gene (gray box). The plasmid linearization with a unique endonuclease induced recombination at the *TgCrk4* locus. Schematics also indicates the relative position of the primers used to conform the TgCrk4 knock-in (**C**). (**B**) Schematics for constructing TgCyc4 Tet-OFF-modified gene. The 5′ fragment of *TgCyc4* genomic locus was amplified and cloned into the targeted plasmid to create N-terminal fusion with 3xHA (HA) epitopes (gray box). The plasmid linearization with a unique endonuclease induced recombination at the *TgCyc4* locus. Schematics also indicates the relative position of the primers used to conform the TgCyc4 knock-in (**D**). (**C, D**) PCR analysis of the parental and transgenic lines expressing TgCrk4^AID-HA^ or ^HA^TgCyc4. The combination of the primers used to detect either native or recombined locus are shown. (**E, F**) Western blot analysis of TgCrk4 (**E**) and TgCyc4 (**F**) immunoprecipitation. The protein complexes were immunoisolated from the soluble fraction [In] (input) of parasites co-expressing endogenous TgCrk4^AID-HA^ or ^HA^TgCyc4. Beads with precipitated complexes (**B**) and depleted soluble fraction [Un] (unbound) were probed with α-HA (α-rat IgG-HRP) antibodies to confirm efficient pulldown of the target proteins. Red asterisk marks the full-length TgCyc4 protein. (**G, H**) The results of the SAINT analysis of TgCrk4 (**G**) and TgCyc4 (**H**) proteomes. The log2 values of the protein spectra detected by mass-spectrometry analysis of the parent parasites and parasites expressing TgCrk4^AID-HA^ or ^HA^TgCyc4 are plotted on the graph. The TgCrk4-TgCyc4 complex is indicated with blue color. TGME49_247040 protein is shown as a dark gray circle.

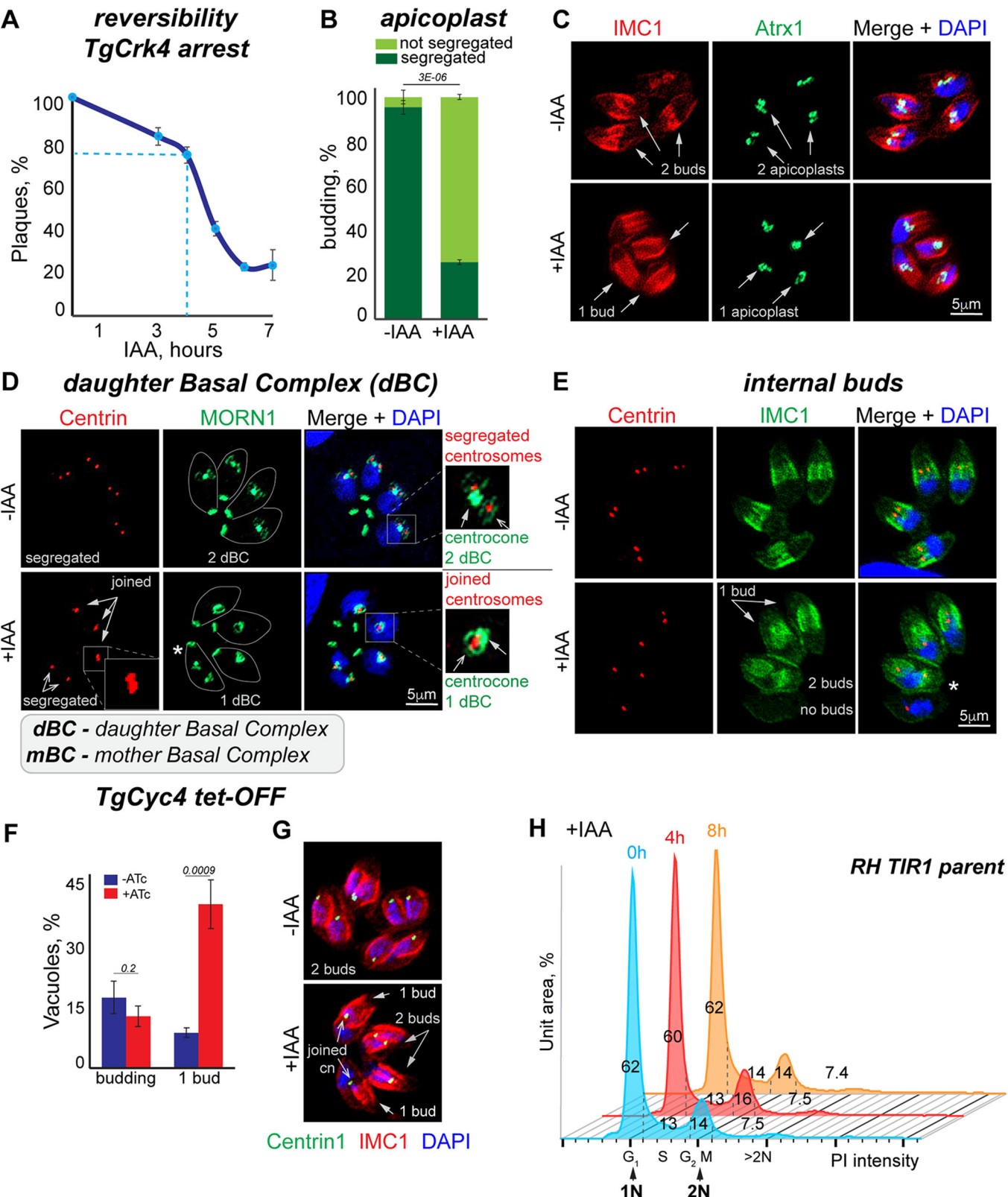

**A** *reversibility TgCrk4 arrest*

**B** *apicoplast*

**C** IMC1 | Atrx1 | Merge + DAPI

**D** *daughter Basal Complex (dBC)*

Centrin | MORN1 | Merge + DAPI

dBC - daughter Basal Complex
mBC - mother Basal Complex

**E** *internal buds*

Centrin | IMC1 | Merge + DAPI

*TgCyc4 tet-OFF*

**F**

**G**

Centrin1 IMC1 DAPI

**H** +IAA

RH TIR1 parent

**Figure EV2.   Characterization of TgCrk4 knockdown phenotype.**

(A) The reversibility of the TgCrk4-induced block was determined by plaque assay. Freshly invaded RH TgCrk4[AID-HA] parasites were incubated with 500μM IAA for indicated times before the medium was replaced with normal growth medium without IAA to allow for plaque development. The plotted mean $-/+$ SD values represent the average plaque numbers from three independent measurements. (B) Quantification of the apicoplast segregation defect caused by RH TgCrk4[AID-HA] deficiency. The number of the vacuoles containing budding parasites that segregated or did not segregate apicoplast were quantified in non-treated and treated with 500 μM IAA for 8 h parasites using TgAtrx1 staining as shown on (C). The longer IAA treatment allowed development of the bigger buds to aid quantifications. A hundred vacuoles of the budding parasites were examined in three independent experiments. The mean $-/+$ SD values are plotted on the graph. The unpaired two-sided $t$ test value is shown. (C–E) Immunofluorescent microscopy analysis of RH$\Delta$Ku80TIR1 TgCrk4[AID-HA] tachyzoites incubated without ($-$IAA) or with 500 μM IAA for 4 h ($+$IAA). (C) Parasites were co-stained with α-TgAtrx1 (α-mouse IgG Fluor 488), α-TgIMC1 (α-rabbit IgG Fluor 488) antibodies and DAPI to visualize apicoplast, internal buds and nuclei. (D) Parasites were co-stained with α-TgMORN1 (α-rabbit IgG Fluor 488), α-Centrin1 (α-mouse IgG Fluor 488) antibodies and DAPI to determine the number of centrosomes and daughter basal complexes (dBC). (E) Parasites were co-stained with α-TgIMC1 (α-rabbit IgG Fluor 488), α-Centrin1 (α-mouse IgG Fluor 488) antibodies and DAPI to visualize internal buds and determine the number of centrosomes per parasite. White asterisk indicates normally dividing tachyzoite. (F) Quantification of the defects caused by TgCyc4 downregulation. The TgCyc4 Tet-OFF parasites were grown without or with ATc for 16 h. The mean $-/+$ SD number of the vacuoles containing budding parasites and parasites forming a single bud in three independent experiments is plotted on the graph. The unpaired two-sided $t$ test values are shown. (G) The immunofluorescence microscopy analysis of the TgCyc4 Tet-OFF parasites grown without or with ATc for 16 h. The parasites were co-stained with α-Centrin1 (α-mouse IgG Fluor 488), α-TgIMC1 (α-rabbit IgG Fluor 488) antibodies and DAPI to visualize centrosomes, internal buds and nuclei. The quantifications of three independent experiments are shown on (F). (H) FACScan analysis of DNA content of the parental RH$\Delta$Ku80TIR1 strain non-treated (0 h, IAA, blue plot) or treated with auxin for 4 h (red plot) and 8 h (orange plot). The results of one of three independent experiments are shown. Dashed lines show the gates used to segregate parasites containing non-replicated ($G_1$: $<1-1N$), replicating (S: $1-<2N$), replicated ($G_2 + M$: $2N$) and over-replicated ($>2N$) DNA.

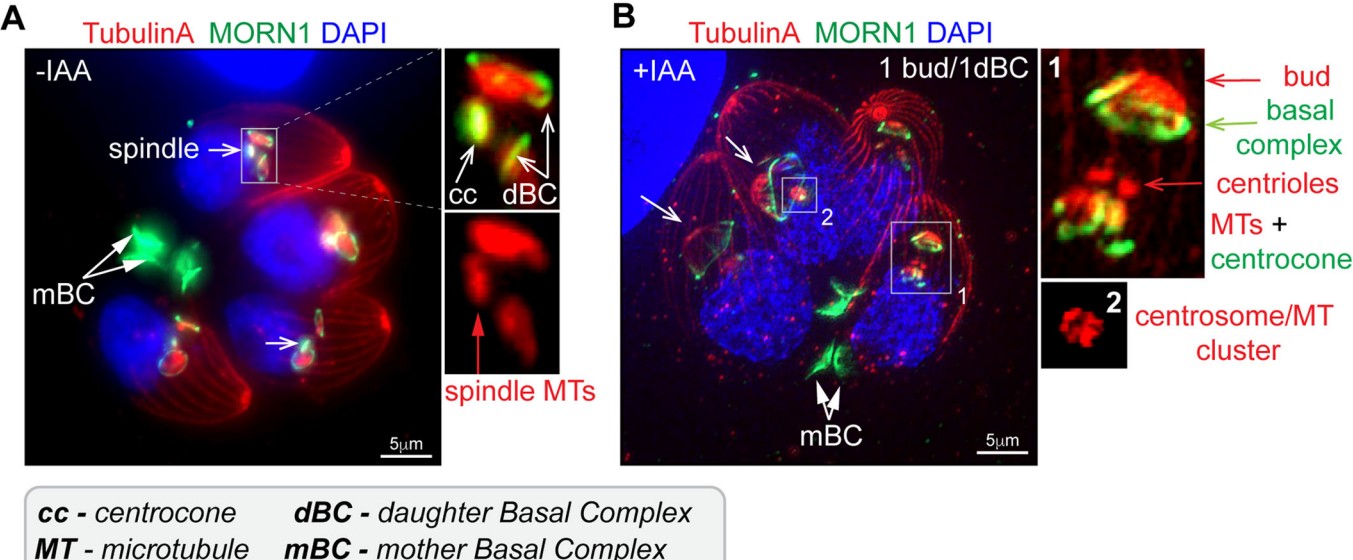

| cc - centrocone | dBC - daughter Basal Complex |
|---|---|
| MT - microtubule | mBC - mother Basal Complex |

**Figure EV3.  Microscopy analysis of TgCrk4-deficient tachyzoites.**

(A, B) The ultra-expansion microscopy analysis of RHΔ*Ku80TIR1* TgCrk4[AID-HA] expressing (A) and deficient (B) tachyzoites. (A) Co-staining of Tubulin A (α-TubulinA/α-mouse IgG Fluor 568) and TgMORN1 (α-MORN1/α-rabbit IgG Flour 488) shows spindle microtubules located in the extended centrocone and two daughter basal complexes (dBC) encircling subpellicular microtubules of the daughter bud. Nucleus stained with DAPI (blue). (B) Co-staining of the TgCrk4-deficient parasites (4 h, 500 µM IAA) shows assembly of a single daughter bud.

**A**

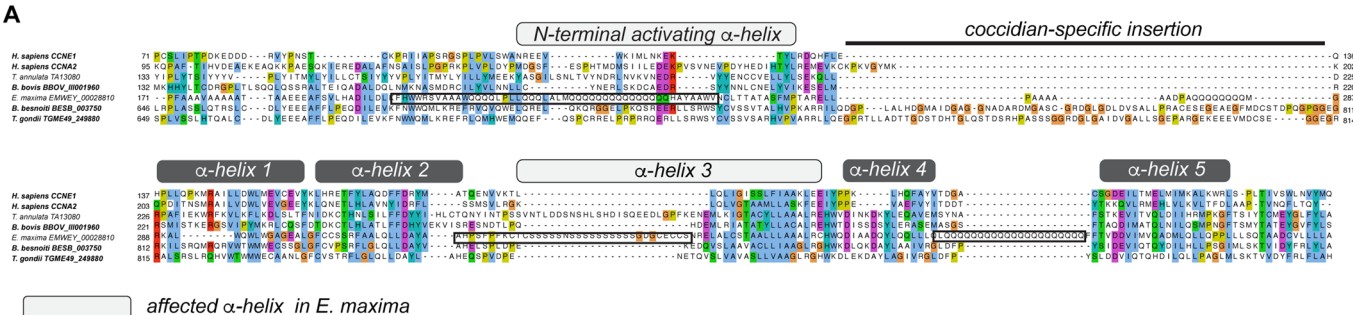

**B**

### N-terminal cyclin domain

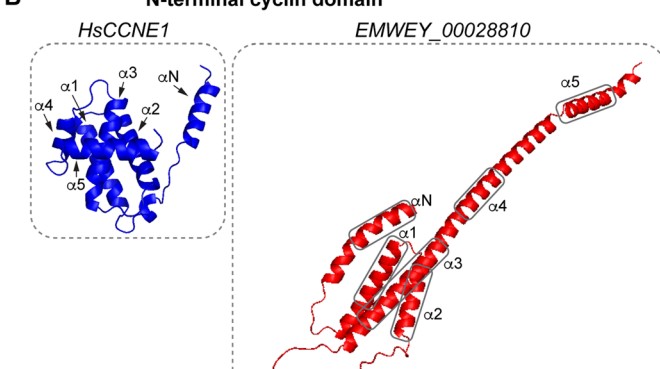

**Figure EV4.  Structural analysis of TgCyc4-related proteins.**

(**A**) Alignment of the N-terminal cyclin domains of apicomplexan Cyc4 proteins with *H. sapiens* Cyclin A2 and Cyclin E1 (MUSCLE). (**B**) Predicted folding of the cyclin domains of *H. sapiens* Cyclin E1 and Eimeria Cyc4.

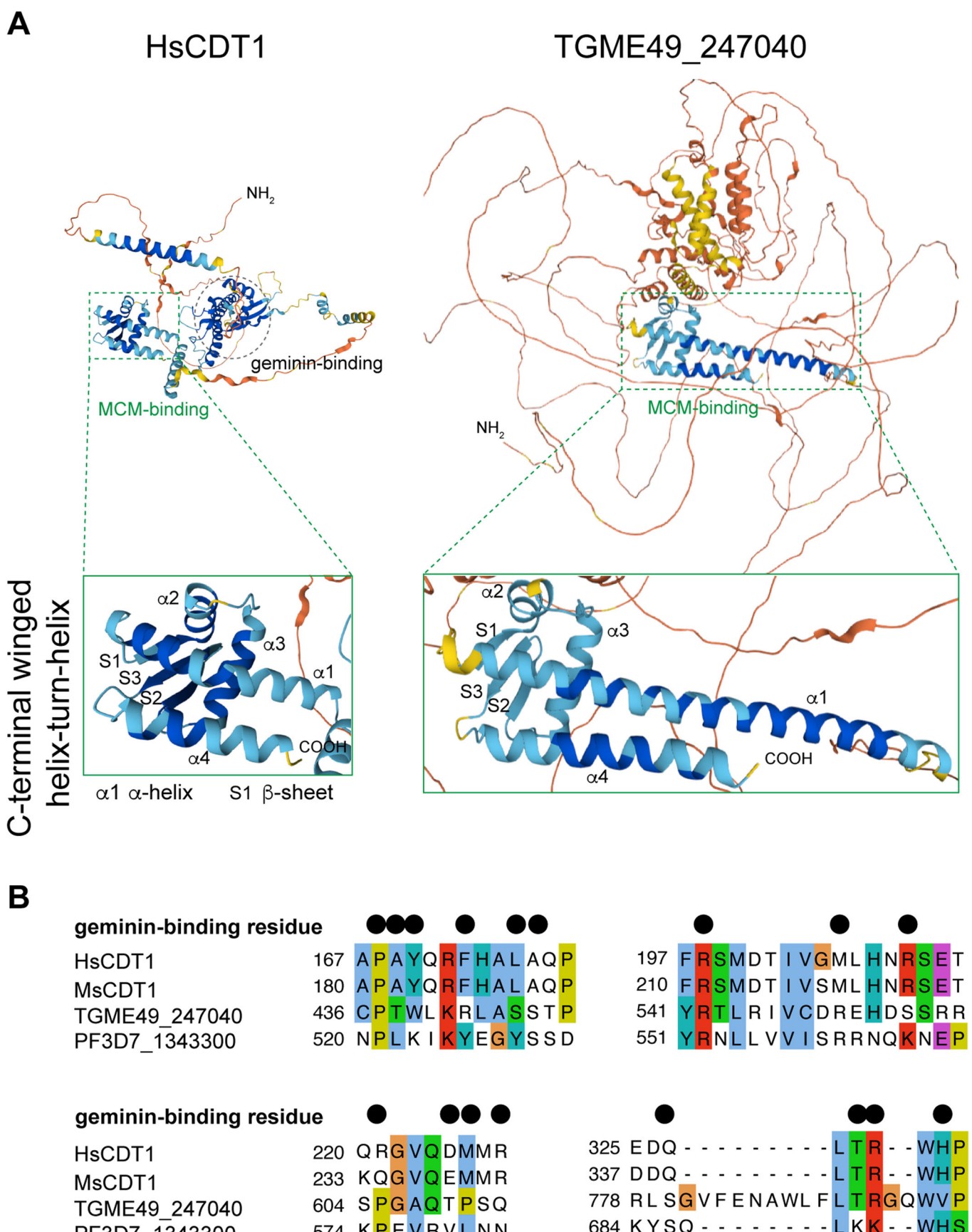

◀ **Figure EV5.  Structural analysis of *Toxoplasma* CDT1-related protein.**

(A) Folding prediction of *H. sapiens* CDT1 and *T. gondii* TGME49_247040 factor (AlphaFold2). The enlarged regions show C-terminal winged helix-turn-helix structure responsible for interaction with MCM complex. (B) Alignment of the central domain of *H. sapiens* CDT1 and *M. musculus* CDT1 with corresponding regions of *T. gondii* TGME49_ 247040 and *P. falciparum* PF3D7_1343300. The black dot indicates the residues involved in CDT1 interaction with inhibitor Geminin. Note a poor conservation of the Geminin-binding sites in apicomplexans.

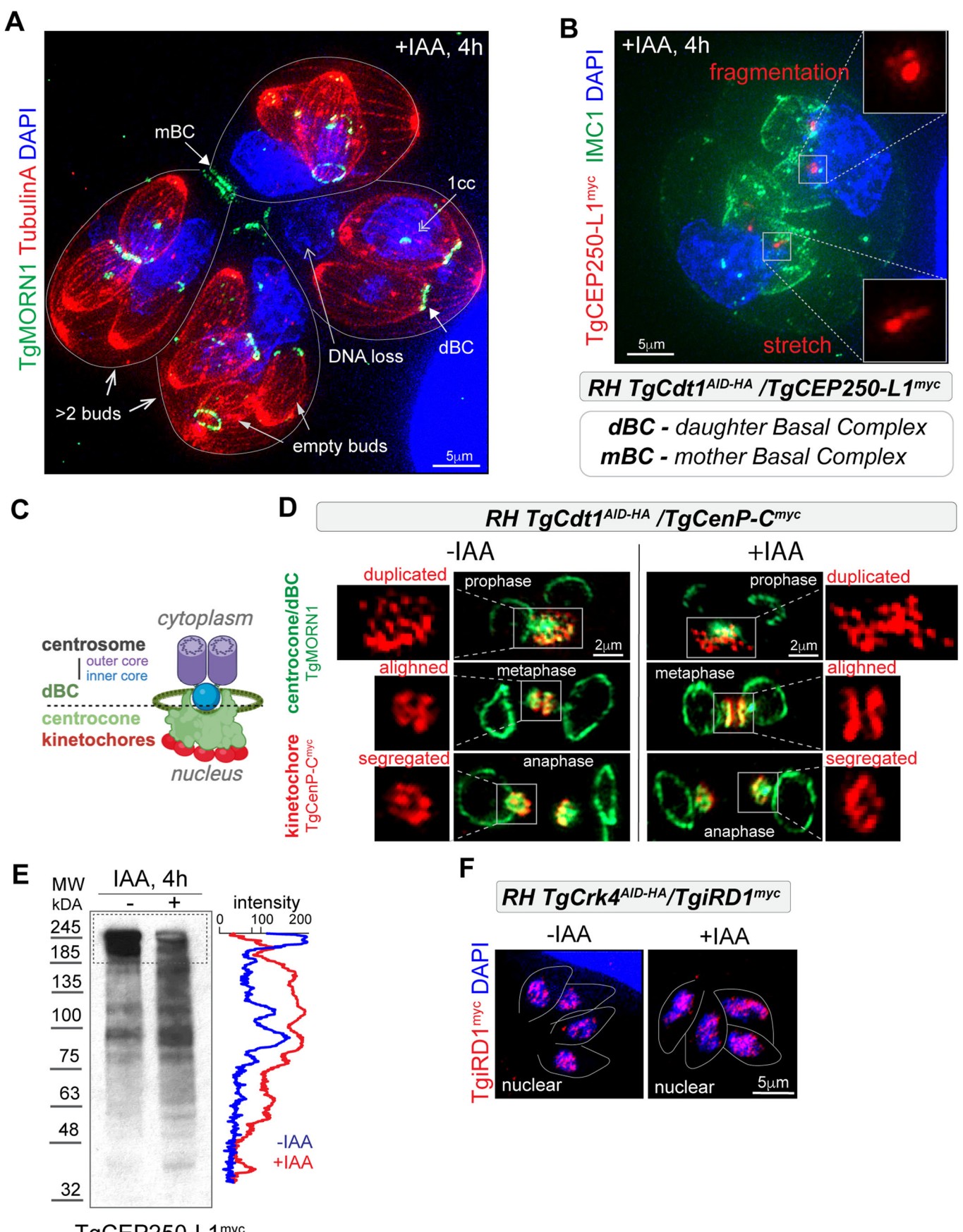

◀ **Figure EV6.  The phenotypic analysis of the TgiRD1 deficient tachyzoites.**

(A) Ultra-expansion microscopy analysis of RHΔ*Ku80TIR1* TgiRD1^AID-HA deficient tachyzoites. Staining with Tubulin A (α-TubulinA/α-mouse IgG Fluor 568) visualizes the mother and the daughters' subpellicular microtubules (buds). The TgMORN1 (α-MORN1/α-rabbit IgG Flour 488) staining shows changes to centrocone and segregates the mother (mBC) and the daughter basal complexes (dBC). DNA mis-segregation defect is depicted with nuclear DAPI stain (blue). (B) The ultra-expansion microscopy images of RHΔ*Ku80TIR1* TgiRD1^AID-HA deficient tachyzoites expressing TgCEP250-L1^myc. The inner core of the centrosome was detected with α-myc (α-rabbit IgG Flour 568) antibodies, the parasite surface with α-IMC1 (α-mouse IgG Fluor 488) and nucleus with DAPI stain. The inner core changes caused by TgiRD1 depletion (+IAA, 4 h) are highlighted in the insets. (C) Schematics of the *T. gondii* perinuclear structures including the bipartite centrosome, centrocone and kinetochores. The drawing depicts one half of the mitotic figure. The dotted line separates nucleoplasm and cytoplasm. dBC daughter basal complex. (D) The ultra-expansion microscopy images of RHΔ*Ku80TIR1* TgiRD1^AID-HA tachyzoites expressing TgCenP-C^myc after 4 h incubation without or with IAA. Three stages of mitosis are shown. To determine the relative position of kinetochores and centrocone, the samples were co-stained with α-myc (α-mouse IgG Flour 568) antibodies and α-TgMORN1 (α-rabbit IgG Fluor 488). (E) Western blot analysis of the TgCEP250-L1^myc expression in the TgiRD1 expressing (−IAA) and deficient (+IAA, 4 h) parasites probed with α-myc. An overexposed image of TgCEP250-L1^myc (Fig. 7G) shows accumulation of the degradation products. The densitometry analysis of the image is shown on the right (ImageJ). (F) IFA analysis of TgiRD1^myc localization in RHΔ*Ku80TIR1* parasites expressing (-IAA) or deficient for TgCrk4^AID-HA ( + IAA, 4 h). Nuclear localization of TgiRD1^myc was detected with α-myc (α-rabbit IgG Flour 568) antibodies and DAPI stain (blue).

