## [Peer Review File · The EMBO Journal]

The Crk4-Cyc4 complex regulates G2/M transition in *Toxoplasma gondii*.

Lauren Hawkins, Chengqi Wang, Dale Chaput, Mrinalini Batra, Clem Marsilia, Danya Awshah, and Elena Suvorova

Corresponding author(s): Elena Suvorova (suvorova@usf.edu)

Review Timeline:

Transfer from Review Commons:	10th Jan 24
Editorial Decision:	28th Feb 24
Revision Received:	12th Mar 24
Accepted:	26th Mar 24

Review
COMMONS

Editor: Hartmut Vodermaier

Transaction Report: This manuscript was transferred to The EMBO JOURNAL following peer review at Review Commons.

Review #1

1. Evidence, reproducibility and clarity:

Evidence, reproducibility and clarity (Required)

Summary:

Hawkins et al. employ a reverse genetic approach to analyze the molecular function of the *Toxoplasma gondii* kinase Crk4 and the *Toxoplasma gondii* cyclin 4. The authors combine inducible depletion with imaging, (phospho-)proteomics, molecular modeling, and protein-protein interaction studies.

Major comments:

- The major conclusion of the manuscript is that TgCrk4/TgCyc4 regulate entry into mitosis and that the primary role of TgCrk4 is to suppress DNA re-replication and chromosome re-duplication (lines 105-106). The authors also provide evidence that TgCrk4 interacts with TgCdt1, a DNA licensing factor ("TgCdt1" is missing in line 107). By sequence homology, the authors found homologues of TgCrk4 only in apicomplexan parasites with binary division and concluded that the dominant division mode, presumably schizogony, is repressed in these organisms in favor of binary division.

Indeed, internal budding and daughter cell formation is defective in the inducible depletion mutants of TgCrk4 and most experiments focus on this developmental stage. However, the analysis of preceding events, such as DNA replication is rather brief. If G2 is indeed regulated by TgCrk4/TgCyc4, one would assume that the parasites are post-S phase and the nucleus contains two copies of the genome, as indicated in Fig. 2C. The data shown in Fig. 3H and 7A, however, show that the TgCrk4 and TgCTD1 depletion induces a developmental arrest pre-S phase. This contradicts the main conclusions of the manuscript.

Indeed, many data of this manuscript could support an alternative conclusion, i.e., that TgCrk4 regulates entry into S-phase (similar to *Plasmodium falciparum* Crk4: PMID: 28211852). This alternative conclusion is supported by the data showing that TgCyc4 is in the nucleus during S-phase (Fig. 1H) and that TgCrk4 interacts with TgCdt1, which has a well-known role in origin of replication licensing and loading of the MCM complex. MCM subunits were less phosphorylated in absence of TgCrk4, which could also suggest a role for TgCrk4 in S phase.

Together, it seems more parsimonious to interpret the data as a DNA replication phenotype rather than a phenotype in G2.

The currently provided data on the DNA content are, however, clearly insufficient to draw firm conclusions. The gating strategy (dotted lines in Figs. 3H, 7A) is unclear. Why are populations, e.g., not separated at the lowest part of the depression in the histogram, but shifted towards lower DNA content? This seems to overestimate the percentage of cells that have a higher DNA content and the statement in lines 269-271, i.e., that TgCrk4 deficient parasites break the "once and only once" rule, is not supported by data.

It is also unclear how many biological replicates are represented by these data (Figs. 3H, 7A), a critical wild type control at $t = 4$ h is missing, as well as a statistical analysis.

Alternatively, the authors could use microscopy to quantify the DNA content of individual nuclei, which would yield a direct read out on whether a nucleus is in pre-S phase, S-phase or post-S phase. Defining the onset of S-phase indirectly by the number of centrosomes per cell seems imprecise, given the small size of the structure and the resolution of the microscope.

Without solving these issues, the major conclusions and several minor statements throughout the manuscript are in question.

- Lines 187-189: The mentioned checkpoint is unclear and so is the "specific cell cycle population". Fig. 2B analyses budding, but as the final step in the cell cycle, the knock down parasites may have arrested at various other stages of the cell cycle. In addition, it is unclear on which primary data Fig. 2B is based. It appears these may be at least partially shown in Fig. 3. If so, please reorganize as this is highly misleading.

- Line 246-254: It is unclear how many biological replicates were performed and how many cells were analyzed to conclude that TgCrk4 deficient parasites cannot form a bipolar spindle (Fig. 2H, S3B). This, together with the possibility that the developmental arrest occurs pre-S phase (Fig. 3H), does not support the statement, that the G2/M transition is regulated by the novel TgCrk4-TgCyc4 complex.

****Minor comments:****

- Throughout the manuscript, please reorganize and present the figures in order of appearance in the text. Also, Fig. 1G summarizes data that are only presented in Fig. 1H. Please reorder. Similarly, Fig. 2C appears to summarize data that are only presented later.

- Why was only the "G1" timepoint quantified in Fig. 1H? Do the other images shown in F and H represent the majority of cells analyzed?

- Several micrographs lack scale bars (Fig. 1B, D; 2E, F, H, I; 6D; 7F, H and S2G, S3A, B; S5A, B, D).

- Lines 83-85 and 93-95: Recently several publications investigated the cell cycle of the apicomplexan parasite Plasmodium and data are accumulating, showing that there may be a gap between the last S phase and segmentation (e.g., PMID: 35731838;

PMID: 35353560), which may be interpreted as a G2 phase. Thus, these statements could be revised to reflect the current literature.

- Fig. 4 shows the effect on protein abundance and phosphorylation upon TgCrk4 depletion. Fig. 4B seems somewhat redundant as a more detailed analysis with two timepoints is shown in the rest of the figure.

- Lines 146-148: This statement is confusing in light of the expression data in Fig.1 F and H. If they stabilize each other, how is TgCrk4 stabilized in G1, when TgCyc4 is absent?

- Fig. 2D, and G: Please provide representative images of what has been quantified, as E/F and H/I are apparently UxEM images.

- Line 236-243: This statement seems to be based on a single IFA shown in Fig. 2K. If so, the manuscript would benefit from clearly stating that this is a singular observation.

- Lines 301-304: In the cited publication, the TgOTUD3A knockout could not be complemented, which raises the possibility that other factors are involved. Thus, this statement would benefit from revision.

- Lines 421-422: PfCdt1 was annotated in PlasmoDB some time ago and this statement needs to be revised.

- Lines 448-450 and Fig. 6F: Are these data from a single biological replicate and how many cells were analyzed for the different time points? Given the insufficient data on the DNA content, the paper would benefit from more conservative conclusions on the role of TgCdt1.

2. Significance:

Significance (Required)

- This manuscript investigates the role of TgCrk3, TgCyc4 and TgCdt1s and provides a large amount of data.

- These data will contribute to our understanding of the unusual division modes of Apicomplexa, a field of research that recently gained momentum.

- These data will be interesting to the community of cell and molecular biologist, which work on the fundamental biology of eukaryotic microorganisms.

- My field of expertise is the cell biology of Apicomplexa.

3. How much time do you estimate the authors will need to complete the suggested revisions:

Estimated time to Complete Revisions (Required)

(Decision Recommendation)

Between 3 and 6 months

No

Review #2

1. Evidence, reproducibility and clarity:

Evidence, reproducibility and clarity (Required)

****Summary:****

In this Manuscript, Hawkins et. al. describe advances in the apicomplexan parasite cell cycle, which is reminiscent but distinct from mammalian cell cycle regulation. These differences include a presumed lack of G2 phase and the ability to replicate in either a multinuclear (schizogony) or binary (endodyogeny) manner. Using *Toxoplasma gondii* (TG) as a model, the authors seek to expand the current understanding of how these highly variable parasitic cell cycles are regulated by describing a previously unreported G2 phase. Building on the authors earlier work, this manuscript defines the function of TgCrk4 and identifies a novel binding partner, TgCyc4. Crk4 and Cyc4 control a G2/M checkpoint by regulating centrosome duplication and separation.

The authors also identify 247040, a protein with previously no known function, as a binding partner and substrate of TgCrk4/TgCyc4 and several replication fork proteins such as MCM and PCNA. Results indicate that the protein negatively regulates replication and centrosome duplication. The authors propose to rename this protein

TgCDT1 despite "low sequence similarity" and having a completely opposite function to eukaryotic CDT1. Using Swiss-Prot modeling the authors claim 247040 bears a "partial resemblance" to mammalian CDT1. Indeed, both of these proteins show high intrinsic disorder and have 2 folded domains. While 247040, like hCDT1, does contain cyclin interacting motifs (Cy), a collection degrons (not all shared with other CDT1 orthologs), and an NLS, the list of nuclear cell cycle proteins that also contain Cy and degron motifs would be very long. Further, 247040 is regulated in an opposite manner to all other CDT1 orthologs because it is absent in TG G1 and present in TG S phase; eukaryotic CDT1 is either degraded or relocalized to the cytoplasm in S phase, and evidence for degradation via APC/C is minimal. Crucially, loss of 247040 resulted in inappropriate replication ("re-replication"), whereas all other eukaryotic CDT1 orthologs are essential for replication. Re-replication in eukaryotic cells can be caused by excess or hyper-active CDT1, not by loss of CDT1 activity as shown here for 247040. Clearly 247040 is a negative regulator of DNA replication, and as such, is not a candidate for the TgCDT1 ortholog. If anything, it is functionally analogous to metazoan geminin, the negative regulator of metazoan CDT1; of note, geminin also has centrosome-related phenotypes. We cannot support naming 247040 TgCDT1 because it will cause confusion in the field.

Aside from this major issue, the study is well-executed, rigorous, quantitative, and thorough; it has many strengths from the unbiased interaction screens. The authors' sequence analysis also suggests broader possibilities for cyclin structures than had previously been appreciated. We appreciate the legend in Figure 2 to the organism-specific terminology.

****Major comments:****

The spatiotemporal dynamics of 247040, its role in repressing TG DNA replication, lack of PIP motif and winged helix domain indicate that some other nomenclature, other than TgCdt1 will be a better name for this protein of previous unknown function.

****Minor comments:****

1. For clarity, please include the number of replicates in the figure legends where appropriate.
2. For microscopy/imaging, how were representative cells/images chosen?
3. In addition to the ELM analysis, the authors could also employ fold recognition software (such as Promal) to analyze 247040 structural models to show similarity to known protein structures.
4. Line 107: missing words "TgCdt1"

5. Line 141: the interpretation that the C terminus is "unstable" is misleading if it is simply that the protein cannot tolerate a fusion to the C-terminus.
 6. Line 221: word choice "reminisced"
 7. Line 348 refers to Orc4 expression in Figure 4A, but the data point is not labelled.
 8. Lines 407-8 and 510-11: Reference Fig 1E
- Line 408: please define what is meant by "dominant interactor"

2. Significance:

Significance (Required)

This manuscript makes great strides in defining apicomplexan cell cycle control and genome replication. These strides include defining a previously unrecognized G2/M checkpoint controlled by TgCrk4 and the novel TgCyc4. Further, the authors identify a binding partner and substrate of the novel Crk4/Cyc4 kinase complex, 247040 that acts as a repressor of replication.

3. How much time do you estimate the authors will need to complete the suggested revisions:

Estimated time to Complete Revisions (Required)

(Decision Recommendation)

Less than 1 month

No

Review #3

1. Evidence, reproducibility and clarity:

Evidence, reproducibility and clarity (Required)

Summary

The present study Hawkins et al have described the important role of Cyclin-CDK complex in an apicomplexan parasite *Toxoplasma*(Tg) which exhibit binary mode of cell division like many other eukaryotes. In the apicomplexan field it is generally shown that G2 phase of cell cycle is either absent or has very little role. The authors here demonstrate that the combination of Tg CRK4 and Tg Cyclin4 works during the G2 phase of cell cycle such as chromosome rereplication and centrosome reduplication. In order to show the function of Cyclin-CRK function they used Auxin degradation system to down regulate or deplete the protein and study parasite growth during cell cycle as well as they used tagged parasite to identify the protein complex with these two molecules. In the study they showed that these two molecules Cyc4 and cRK4 formed the complex in protein pulldown method and show identical function in the cell cycle. In addition to these two proteins they also found another interacting partner Cdt1 that was further analysed to be involved in controlling Chromosome rereplication and centrosome. So overall the study is nicely performed and three molecules of Cyclin4-CRK4-Cdt1 and their role is illustrated in the binary mode of cell division in *Toxoplasma*.

Comments

1. Though no new experiments need to be performed but it will be good if some details are given as to which stage of tachyzoite cycle the protein complex were performed and if there is difference in the various phases of cell cycle especially the S phase and the M phase. Are these period changed. Since G2 is suppose to be absent in many apicomplexan do the authors suggest that G2 phase is only coupled to binary mode of cell division. Please discuss how it is then linked to the other part of cell cycle.
2. Ganter et al have studied CRK4 in *Plasmodium* previously and they do find in their phosphoproteome study the similar association with the DNA replication machinery with CRK4 but no cyclin was identified in their study. In the cyclin study by Roques et al it has been shown that no cell cycle cyclins are found in Apicomplexan so can the author discuss more how these complex can be different in two apicomplexan species. They describe that Crk4 is novel cell cycle kinase though this has been studied earlier. Authors have almost not discussed these previous finding with respect

to their in this study.

3. The manuscript is too dense, in terms of both figures and text. At times loses the focus and hence can be organised with most important finding in the figure and text. Especially Fig2, Fig4 and Fig7. Fig5 does not give too much in terms of the real finding and in fact take away from the focus. Some parts of these figures can be simplified or moved to supplementary. Some of the figures in Fig2 and 7 are missing the scale bars.

4. May be bit more discussion of ORC in relation to their Cyclin-CRK complex as they did find upregulation of the ORC in their genome profiling. So may be instead of CDT1 these are more important in the licensing of DNA replication.

5 The model in Fig8B does not take Cyc4 into consideration and I feel is bit oversimplified as there are many factors that may be responsible for centrosome non separation. The S and G2 are not separated in the Cell cycle as given in this Fig.

6. It is not clear from the data with CDT1 if this linking the inner and outer centrosome or its down regulation breaks the bipartite centrosome. May be some reflection it will be useful.

****Minor comments****

I what is SAINT analysis as it is not described in methods.

2. How was budding quantified

3. Western blot can have predicted size

4. what does red star mean in Blot 1C

5. What does the number in Fig1H means please explain in the legends and same for Fig6F. In fig 1, removing the inhibition for 5 hours led to very less budding, but in fig 3, removing inhibition showed increased budding (50% in 2 hours). Please explain

6. Fig2 has no scale bars -please add- this figure is too dense. May be fig2A, B,C can be in supplementary, legend in the figure can be in the figure legend.

7. Also this figure2 H and I in not quoted in line 231. Also this figure2 has no panel J but goes directly from I to K

8. Fig3 the FigG can be more relevant in the Figure 8 while describing about the Crk4 and Cyc4 and CDT1 in binary mode of cell division. Also please define what stars mean either in legend or methods section in terms of significance.

Line 107 the sentence is incomplete

Line 217 may be the figure could be referred as then it is not clear about the description.

****Referees cross-commenting****

The study is quite rigorous and with analyses of CRK4-CYC4 and CDT. However it will be better if authors please revisit their conclusions on G2 phase of cell cycle in Toxoplasma based on their findings. The study will have important bearing on the community studying apicomplexan parasites and DNA replication as well as who work on eukaryotic cell cycle.

2. Significance:

Significance (Required)

In the manuscript by Hawkins et al have illustrated that in the apicomplexan parasite that have binary mode of cell division present a Cyclin-Crk complex with detailed analysis of Tg Crk4-Cyc4 that are novel in these group of parasite infect humans and animal alike like malaria parasite and ones affecting cattle and chicken. So these findings are novel as very little is known about this interaction. The significant finding is to show how the G2 phase of cell cycle may be regulated in these parasites and how DNA licensing factor Cdt1 is highly divergent but part of this CRK-Cyclin complex.

So though it discusses more on the Toxoplasma but it may be of interest to the scientist working on eukaryotes with divergent mode of cell cycle.

General Assessment - The findings are novel but the manuscript is too dense and at times loses the focus. May be both text and Figures could be made less dense so that important findings are revealed in a better way.

Advance - It does give important insight into the cell cycle in apicomplexan parasite and how even though there are no cell cycle cyclin in Apicomplexa. The findings here suggest how different complexes can substitute for the function. It does extend the knowledge in the field of Cell division in divergent parasites both in terms of mechanistic, functional and technical way.

3. How much time do you estimate the authors will need to complete the suggested revisions:

Estimated time to Complete Revisions (Required)

(Decision Recommendation)

Between 1 and 3 months

Yes

Full Revision

Manuscript number: RC-2023-02245

Corresponding author(s): Elena S. Suvorova

1. General Statements [optional]

We would like to thank our reviewers for their constructive criticism and for their appreciation and enthusiasm for our study. Some reviewers expressed opposing views, particularly when it came to the function and identity of the Cdt1-related protein in Toxoplasma gondii. To avoid redundancy in our response, we would like to make a brief statement. Toxoplasma gondii and other apicomplexan parasites utilize unique and highly unusual modes of cell division; numerous studies suggest that multiple phases can run concurrently in apicomplexan cell cycles. The best-known examples include the asynchronous S/M cycles in schizogony and concurrent mitosis and budding in Toxoplasma endodyogeny. These overlapping phases are not a feature exclusive to apicomplexans, since in budding yeast, cytokinesis initiates in G₁ phase by marking the location of budding on the surface of the mother. Based on years of previous research and from our experience, we adjusted our approach by focusing on the processes that are associated with each cell cycle phase rather than on their temporal order. While the model of a conventional cell cycle guides our studies, we “follow the breadcrumbs” that we discover and the published studies to create a more accurate model of apicomplexan cell cycle instead of relying on the traditional cell cycle map employed by distantly related eukaryotes. Below are point-to-point responses to reviewers’ comments.

Reviewer #1 (Evidence, reproducibility and clarity (Required)):

Summary:

Hawkins et al. employ a reverse genetic approach to analyze the molecular function of the Toxoplasma gondii kinase Crk4 and the Toxoplasma gondii cyclin 4. The authors combine inducible depletion with imaging, (phospho-)proteomics, molecular modeling, and protein-protein interaction studies.

Major comments:

- The major conclusion of the manuscript is that TgCrk4/TgCyc4 regulate entry into mitosis and that the primary role of TgCrk4 is to suppress DNA re-replication and chromosome re-duplication (lines 105-106). The authors also provide evidence that TgCrk4 interacts with TgCdt1, a DNA licensing factor ("TgCdt1" is missing in line 107). *(had been corrected)* By sequence homology, the authors found homologues of TgCrk4 only in apicomplexan parasites with binary division and concluded that the dominant division mode, presumably schizogony, is repressed in these organisms in favor of binary division.

Indeed, internal budding and daughter cell formation is defective in the inducible depletion mutants of TgCrk4 and most experiments focus on this developmental stage. However, the

analysis of preceding events, such as DNA replication is rather brief.

If G₂ is indeed regulated by TgCrk4/TgCyc4, one would assume that the parasites are post-S phase and the nucleus contains two copies of the genome, as indicated in Fig. 2C. The data shown in Fig. 3H and 7A, however, show that the TgCrk4 and TgCTD1 depletion induces a developmental arrest pre-S phase. This contradicts the main conclusions of the manuscript.

We agree that the G₂ location is odd for a conventional cell cycle model. Given the high possibility that cell cycle phases can overlap in apicomplexans, we determined the relative position of G₂ phase in Toxoplasma endodyogeny by instead focusing solely on the processes that are attributed to a specific cell cycle phase (such as DNA replication for S phase, DNA re-replication for G₂ phase, DNA segregation for mitosis). Our approach shows that Toxoplasma G₂/M checkpoint operates upstream of SAC, which led to enrichment of parasites with replicated DNA (Fig. 3H and Fig. 7A), which places G₂ at the end of S-phase. Our focus in the present study is on the G₂ functions, the control of centrosome and chromosome reduplication, but we appreciate the suggestion to examine DNA replication in Toxoplasma, which could be investigated in future studies.

Indeed, many data of this manuscript could support an alternative conclusion, i.e., that TgCrk4 regulates entry into S-phase (similar to Plasmodium falciparum Crk4: PMID: 28211852). This alternative conclusion is supported by the data showing that TgCyc4 is in the nucleus during S-phase (Fig. 1H) and that TgCrk4 interacts with TgCdt1, which has a well-known role in origin of replication licensing and loading of the MCM complex. MCM subunits were less phosphorylated in absence of TgCrk4, which could also suggest a role for TgCrk4 in S phase. Together, it seems more parsimonious to interpret the data as a DNA replication phenotype rather than a phenotype in G₂.

We understand some confusion from prior data, but PfCrk4 is not orthologous to TgCrk4 (Alvarez & Suvorova, 2017); The true TgCrk4 ortholog had not been found in Plasmodium genomes. Our understanding is that nuclear accumulation of TgCyc4 in S-phase activates TgCrk4, which leads to repression of the DNA reduplication. One of the possible mechanisms involves interfering with loading of the MCM complex on chromatin mediated by hyper-phosphorylated TgiRD1 (former TgCdt1), which has been reported in other eukaryotes. We also believe that increased MCM phosphorylation indicates entry into or active S-phase, while the reduced phosphorylation that was detected in Crk4-depleted cells supports a block at the end of S-phase (G₂).

The currently provided data on the DNA content are, however, clearly insufficient to draw firm conclusions. The gating strategy (dotted lines in Figs. 3H, 7A) is unclear. Why are populations, e.g., not separated at the lowest part of the depression in the histogram, but shifted towards lower DNA content? This seems to overestimate the percentage of cells that have a higher DNA content and the statement in lines 269-271, i.e., that TgCrk4 deficient parasites break the "once and only once" rule, is not supported by data.

We corrected the gating of the FACScan plots to separate G₁, S, G₂+M, and parasites with over-duplicated DNA. Please note that, in general, the cell cycle gating of FACScan data is relative and somewhat subjective when it comes to the gaussian curve. Independent of the chosen

gates, our data show that removal of either TgCrk4 or TgiRD1 led to substantial decrease of the G₁ population (reduction of 1N peak) accompanied by increase of parasites in the process of replication, completed replication (increase of 1.8 N peak), as well as undergoing DNA re-replication, which supports our claim in lines 269-271. In the case of TgiRD1, the number of parasites with re-duplicated DNA nearly doubled upon 8h of factor deficiency.

It is also unclear how many biological replicates are represented by these data (Figs. 3H, 7A), a critical wild type control at t = 4 h is missing, as well as a statistical analysis.

Alternatively, the authors could use microscopy to quantify the DNA content of individual nuclei, which would yield a direct read out on whether a nucleus is in pre-S phase, S-phase or post-S phase. Defining the onset of S-phase indirectly by the number of centrosomes per cell seems imprecise, given the small size of the structure and the resolution of the microscope. Without solving these issues, the major conclusions and several minor statements throughout the manuscript are in question.

Thank you for your point, we performed a minimum of three independent experiments to evaluate the DNA content of TgCrk4- or TgiRD1- (former TgCdt1) depleted tachyzoites and have now indicated this in the figure legends. The 0h time point is a "wild type" control, since the parasites that expressed factors were incubated without auxin (mock treated) for 4h. The DNA content of Toxoplasma has been thoroughly studied and we are thus confident our 0h data is a good representation of asynchronous healthy populations. Although the parental strain had been examined, due to the data density mentioned in the reviews, we included only relative results (control and two experimental points) for clarity. Our concern with using microscopy to analyze DNA content is that it can be highly subjective, hinging on the quality of staining and imaging, while flow cytometry produces more unbiased datasets. We have considered the concern that the start of centrosome duplication can be difficult to identify, but the centrin-positive centrosomes move apart by the middle of S-phase. The independent structures are then distinct and easy to resolve, providing a popular means of marking G₁/S transition in Toxoplasma.

- Lines 187-189: The mentioned checkpoint is unclear and so is the "specific cell cycle population". Fig. 2B analyses budding, but as the final step in the cell cycle, the knock down parasites may have arrested at various other stages of the cell cycle. In addition, it is unclear on which primary data Fig. 2B is based. It appears these may be at least partially shown in Fig. 3. If so, please reorganize as this is highly misleading.

"A checkpoint" in the indicated lines refers to G₂/M and SAC, which are regulated by TgCrk4 and TgCrk6, respectively. We refer to "specific cell cycle population" since each transgenic parasite that is subject to G₂/M or SAC arrest can allow us to isolate very different cell cycle stages. TgCrk6-dependent arrest had been confirmed by the presence of unresolved centrocone (not shown but was previously reported in Hawkins et al., 2022), while we thoroughly examined the novel TgCrk4-dependent block by focusing on many parameters, such as joint centrosomes, single-bud assembly, or unresolved apicoplast. Fig. 2 and Fig. S2 summarize our rigorous quantifications of these phenotypes. For convenience, we used budding efficiency as a readout to compare arrest and release of G₂/M and SAC, which was incorporated

in Fig. 2B. Table S4 contains the primary data used in all figures in the manuscript, including Fig. 2B.

- Line 246-254: It is unclear how many biological replicates were performed and how many cells were analyzed to conclude that TgCrk4 deficient parasites cannot form a bipolar spindle (Fig. 2H, S3B). This, together with the possibility that the developmental arrest occurs pre-S phase (Fig. 3H), does not support the statement, that the G2/M transition is regulated by the novel TgCrk4-TgCyc4 complex.

We have indicated our replicates in the M&M. As addressed for Fig. 3H above, these IFA experiments were performed in at least three independent experiments.

Minor comments:

- Throughout the manuscript, please reorganize and present the figures in order of appearance in the text. Also, Fig. 1G summarizes data that are only presented in Fig. 1H. Please reorder.

Similarly, Fig. 2C appears to summarize data that are only presented later.

Thank you for the suggestion, however we must abide by the standards of the publishers. The order of the figures must be maintained, but there is a substantial degree of freedom in organizing panels within figures. Fig. 1G summarizes data shown in Fig. 1F, H, while Fig. 2C summarizes many panels including preceding Fig. 2B and Fig. S2. Most of our schematics are placed at the top of figures to provide guidance for the relevant experiments.

- Why was only the "G1" timepoint quantified in Fig. 1H? Do the other images shown in F and H represent the majority of cells analyzed?

You are correct, we indicated the percentage of factor-positive parasites only when the factor emerges during a specific cell cycle phase. For example, the TgCyc4-positive parasites with 1 centrin dot were quantified to show that TgCyc4 emerges in the middle of G₁ phase. The lack of a number indicates that the image represents all the parasites progressing through this phase; we have added this explanation to the figure legends.

- Several micrographs lack scale bars (Fig. 1B, D; 2E, F, H, I; 6D; 7F, H and S2G, S3A, B; S5A, B, D).

Thank you, we have added the scale bars to indicated images.

- Lines 83-85 and 93-95: Recently several publications investigated the cell cycle of the apicomplexan parasite Plasmodium and data are accumulating, showing that there may be a gap between the last S phase and segmentation (e.g., PMID: 35731838; PMID: 35353560), which may be interpreted as a G2 phase. Thus, these statements could be revised to reflect the current literature.

The studies mentioned provide very valuable insights into S-phase dynamics; the gap that was detected between S-phase and segmentation includes mitotic events such as prophase, metaphase, and anaphase prior to telophase (karyokinesis to segmentation). However, studies using means like stage-specific markers could help resolve the composition and order of events

in the apicomplexan cell cycle. We used processes specific to G₂ (repression of DNA and centrosome reduplication) and identified TgCrk4/TgCyc4 as the first G₂ markers in apicomplexans.

- Fig. 4 shows the effect on protein abundance and phosphorylation upon TgCrk4 depletion. Fig. 4B seems somewhat redundant as a more detailed analysis with two timepoints is shown in the rest of the figure.

Fig. 4B is provided in contrast to the plot in Fig. 4A. It demonstrates that TgCrk4 depletion results in a far more pronounced effect on global phosphorylation rather than on proteolysis. While Fig. 4B highlights the checkpoint arrest, panels C and D are dedicated to the search for TgCrk4 substrates: the phospho-sites that immediately lost intensity of phosphorylation and remained low during the 4h block.

- Lines 146-148: This statement is confusing in light of the expression data in Fig.1 F and H. If they stabilize each other, how is TgCrk4 stabilized in G₁, when TgCyc4 is absent?

We believe that multiple mechanisms contribute to the stability and function of TgCrk4. We tested one and found that depleting the cyclin partner led to reduced expression of TgCrk4, and were able to conclude that the complex is stable when both subunits are expressed. Please note that we probed the mixed cell cycle populations by WB, and our proteomics data show that TgCrk4 interacts with many partners (Fig. 1E). Thus, it is likely that G₁ stability may have been mediated by other partners, or by a higher transcription/translation rate, which could be evaluated in further experiments that focus on the regulation of TgCrk4/TgCyc4 complex.

- Fig. 2D, and G: Please provide representative images of what has been quantified, as E/F and H/I are apparently UxEM images.

The corresponding images are included in Fig. S2.

- Line 236-243: This statement seems to be based on a single IFA shown in Fig. 2K. If so, the manuscript would benefit from clearly stating that this is a singular observation.

Thank you, we have provided clarification as described in previous points.

- Lines 301-304: In the cited publication, the TgOTUD3A knockout could not be complemented, which raises the possibility that other factors are involved. Thus, this statement would benefit from revision.

The lack of TgOTUD3A KO complementation is an example of the unappreciated complexity of apicomplexan cell cycle regulation by controlled proteolysis. We highlighted the similarity of TgCrk4 and TgOTUD3A deficiencies, which indirectly confirms their partnerships in the G₂ network. Fig. 8A shows that, in addition to TgOTUD3A, the G₂ network contains numerous factors.

- Lines 421-422: PfCdt1 was annotated in PlasmoDB some time ago and this statement needs to be revised.

Please see our response to comments made by Reviewer 2. Briefly, we agree with Reviewer 2 comment that TgCdt1 does not function as conventional DNA replication licensing factor CDT1. Therefore, we named TGME49_247040 TgiRD1 – inhibitor of DNA and centrosome ReDuplication 1.

- Lines 448-450 and Fig. 6F: Are these data from a single biological replicate and how many cells were analyzed for the different time points? Given the insufficient data on the DNA content, the paper would benefit from more conservative conclusions on the role of TgCdt1.

The numbers of biological replicates were added throughout the text, also please refer to our response to Reviewer 2 and the comment above.

Reviewer #1 (Significance (Required)):

- This manuscript investigates the role of TgCrk3, TgCyc4 and TgCdt1s and provides a large amount of data.
- These data will contribute to our understanding of the unusual division modes of Apicomplexa, a field of research that recently gained momentum.
- These data will be interesting to the community of cell and molecular biologist, which work on the fundamental biology of eukaryotic microorganisms.
- My field of expertise is the cell biology of Apicomplexa.

Reviewer #2 (Evidence, reproducibility and clarity (Required)):

Summary:

In this Manuscript, Hawkins et. al. describe advances in the apicomplexan parasite cell cycle, which is reminiscent but distinct from mammalian cell cycle regulation. These differences include a presumed lack of G2 phase and the ability to replicate in either a multinuclear (schizogony) or binary (endodyogeny) manner. Using *Toxoplasma gondii* (TG) as a model, the authors seek to expand the current understanding of how these highly variable parasitic cell cycles are regulated by describing a previously unreported G2 phase. Building on the authors earlier work, this manuscript defines the function of TgCrk4 and identifies a novel binding partner, TgCyc4. Crk4 and Cyc4 control a G2/M checkpoint by regulating centrosome duplication and separation.

The authors also identify 247040, a protein with previously no known function, as a binding partner and substrate of TgCrk4/TgCyc4 and several replication fork proteins such as MCM and PCNA. Results indicate that the protein negatively regulates replication and centrosome duplication. The authors propose to rename this protein TgCDT1 despite "low sequence similarity" and having a completely opposite function to eukaryotic CDT1. Using Swiss-Prot modeling the authors claim 247040 bears a "partial resemblance" to mammalian CDT1. Indeed,

both of these proteins show high intrinsic disorder and have 2 folded domains. While 247040, like hCDT1, does contain cyclin interacting motifs (Cy), a collection degrons (not all shared with other CDT1 orthologs), and an NLS, the list of nuclear cell cycle proteins that also contain Cy and degron motifs would be very long. Further, 247040 is regulated in an opposite manner to all other CDT1 orthologs because it is absent in TG G₁ and present in TG S phase; eukaryotic CDT1 is either degraded or relocated to the cytoplasm in S phase, and evidence for degradation via APC/C is minimal. Crucially, loss of 247040 resulted in inappropriate replication ("re-replication"), whereas all other eukaryotic CDT1 orthologs are essential for replication. Re-replication in eukaryotic cells can be caused by excess or hyper-active CDT1, not by loss of CDT1 activity as shown here for 247040. Clearly 247040 is a negative regulator of DNA replication, and as such, is not a candidate for the TgCDT1 ortholog. If anything, it is functionally analogous to metazoan geminin, the negative regulator of metazoan CDT1; of note, geminin also has centrosome-related phenotypes. We cannot support naming 247040 TgCDT1 because it will cause confusion in the field.

Aside from this major issue, the study is well-executed, rigorous, quantitative, and thorough; it has many strengths from the unbiased interaction screens. The authors' sequence analysis also suggests broader possibilities for cyclin structures than had previously been appreciated. We appreciate the legend in Figure 2 to the organism-specific terminology.

Major comments:

The spatiotemporal dynamics of 247040, its role in repressing TG DNA replication, lack of PIP motif and winged helix domain indicate that some other nomenclature, other than TgCdt1 will be a better name for this protein of previous unknown function.

We would like to thank Reviewer 2 for this highly insightful comment. We agree that TGME49_247040 functions as a CDT1 inhibitor rather than as CDT1 itself, so conserving the name would produce confusion in the cell cycle field. Based on TGME49_247040 protein function we decided to name this factor TgiRD1 – inhibitor of DNA and centrosome ReDuplication 1. We revisited our data, looked deeper into the protein structure, and adjusted our conclusions. Our new Figure S5 shows differences in the predicted folding of HsCDT1 and TgiRD1. We could not ignore the fact that TgiRD1 is phylogenetically related to CDT1 in ancestral branches and metazoans (Fig. 6B), but we identified substantial differences that may indicate a selective loss (or inheritance) of protein features. For example, TgiRD1 does not interact with ORCs that are critical for the licensing step, but TgiRD1 retained an MCM binding domain (winged helix-turn-helix) that plays a role in licensing and firing. Rather than CRL4^{Cdt2} degrons, TgiRD1 contains APC/C degrons that would be activated late in mitosis (similar to regulation of Geminin). Together with the lack of DNA licensing control in G₁ and its opposing expression profile, we concluded that TgiRD1 represents a Cdt1-related protein that controls DNA and centrosome reduplication in S and G₂ phases.

Minor comments:

1. For clarity, please include the number of replicates in the figure legends where appropriate.

We added the requested information.

2. For microscopy/imaging, how were representative cells/images chosen?
The representative images constituted the most common phenotype of the feature we aimed to highlight, and most are accompanied by quantifications.

3. In addition to the ELM analysis, the authors could also employ fold recognition software (such as Promal) to analyze 247040 structural models to show similarity to known protein structures.
We use a variety of folding prediction software, including AlphaFold2, PyMol, and template-based SWISS-PRO module to examine protein structures in our study, indicated in the text and figure legends. Our new TgiRD1 (former TgCdt1) analysis is based on an AlphaFold2 prediction (Fig. S5). All the software we used is listed in the M&M section.

4. Line 107: missing words "TgCdt1"
We corrected the sentence.

5. Line 141: the interpretation that the C terminus is "unstable" is misleading if it is simply that the protein cannot tolerate a fusion to the C-terminus.
We successfully incorporated a tag at the C-terminus (confirmed by sequencing across the recombinant gene) but could not detect protein expression. If our protein could not tolerate a recombinant tag, the transgenic parasites would not survive because TgCyc4 is essential protein. Therefore, since the parasites survived, we concluded that the lack of TgCyc4-AID-HA expression was due to native truncation at the C-tail (instability).

6. Line 221: word choice "reminisced"
We have changed the wording.

7. Line 348 refers to Orc4 expression in Figure 4A, but the data point is not labelled.
Fig. 4A references GO group (DNA replication/licensing factors), and the raw data is included in Table S6, which is now indicated in the text.

8. Lines 407-8 and 510-11: Reference Fig 1E
We added the reference.

Line 408: please define what is meant by "dominant interactor"
We meant that TgiRD1 is the most prominent interactor of TgCrk4 and TgCyc4. To clarify the confusion, we changed the wording to "primary interactor".

Reviewer #2 (Significance (Required)):

This manuscript makes great strides in defining apicomplexan cell cycle control and genome replication. These strides include defining a previously unrecognized G2/M checkpoint controlled by TgCrk4 and the novel TgCyc4. Further, the authors identify a binding partner and

substrate of the novel Crk4/Cyc4 kinase complex, 247040 that acts as a repressor of replication.

Reviewer #3 (Evidence, reproducibility and clarity (Required)):

Summary

The present study Hawkins et al have described the important role of Cyclin-CDK complex in an apicomplexan parasite *Toxoplasma*(Tg) which exhibit binary mode of cell division like many other eukaryotes. In the apicomplexan field it is generally shown that G₂ phase of cell cycle is either absent or has very little role. The authors here demonstrate that the combination of Tg CRK4 and Tg Cyclin4 works during the G₂ phase of cell cycle such as chromosome rereplication and centrosome reduplication. In order to show the function of Cyclin-CRK function they used Auxin degradation system to down regulate or deplete the protein and study parasite growth during cell cycle as well as they used tagged parasite to identify the protein complex with these two molecules. In the study they showed that these two molecules Cyc4 and cRK4 formed the complex in protein pulldown method and show identical function in the cell cycle. In addition to these two proteins they also found another interacting partner Cdt1 that was further analysed to be involved in controlling Chromosome rereplication and centrosome. So overall the study is nicely performed and three molecules of Cyclin4-Crk4-Cdt1 and their role is illustrated in the binary mode of cell division in *Toxoplasma*.

Comments

1. Though no new experiments need to be performed but it will be good if some details are given as to which stage of tachyzoite cycle the protein complex were performed and if there is difference in the various phases of cell cycle especially the S phase and the M phase. Are these period changed. Since G₂ is suppose to be absent in many apicomplexan do the authors suggest that G₂ phase is only coupled to binary mode of cell division. Please discuss how it is then linked to the other part of cell cycle.

*You are correct, we propose that the presence of G₂ phase is linked to binary division in apicomplexans and our hypothesis is supported by the overall evolution of the cell cycle (see Discussion section). We also entertained the hypothesis that G₂ operates in multinuclear division since all apicomplexans encode TgiRD1 orthologs (please, see the Discussion section). For the first time, we identified the major functions of G₂ functions (repression of the DNA and centrosome reduplication) in the apicomplexan cell cycle. However, given the unresolved organization of the *Toxoplasma* (or any apicomplexan) cell cycle, it is currently impossible to define the boundaries of G₂. According to our study, TgCrk4 and TgCyc4 control G₂/M transition or the end of G₂ phase, and we still lack markers of G₂ entry. In our comparative synchronization study (Fig 2), we uncovered the temporal link between G₂/M and SAC regulatory points, which is discussed in the results section.*

2. Ganter et al have studied CRK4 in *Plasmodium* previously and they do find in their phosphoproteome study the similar association with the DNA replication machinery with CRK4 but no cyclin was identified in their study. In the cyclin study by Roques et al it has been shown

that no cell cycle cyclins are found in Apicomplexan so can the author discuss more how these complex can be different in two apicomplexan species. They describe that Crk4 is novel cell cycle kinase though this has been studied earlier. Authors have almost not discussed these previous finding with respect to their in this study.

We would like to clarify this confusion. We have not discussed Ganter et al. studies because PfCRK4 is not orthologous to TgCrk4, but rather it is related to TgCrk6. Unfortunately, the Plasmodium and Toxoplasma Crk nomenclature was published almost concurrently. Our previous (Alvarez & Suvorova, 2017) and current study show that Plasmodium and other apicomplexans that divide by multinuclear division do not encode TgCrk4 orthologs (and/or TgCyc4). Additionally, the mentioned studies by Roques and Ganter were released prior to newer genome annotations that include additional cyclin-domain proteins, including 10 Toxoplasma cyclins (5 new) that we categorized in our recent publication (Hawkins et al., 2022). Although the newly annotated cyclins are not related to conventional cell cycle cyclins, we had proven empirically that TgCyc1 together with TgCrk6 controls SAC, and now, the specific interaction of TgCyc4 with TgCrk4 controls G₂ processes. Lastly, we call TgCrk4 “a novel” kinase only in the meaning that it is a novel cyclin-dependent kinase that is not related to known CDKs in other eukaryotes. The identification of TgCrk4 in our previous study (Alvarez & Suvorova, 2017) is described in the Introduction section and at the opening of the Results.

3. The manuscript is too dense, in terms of both figures and text. At times loses the focus and hence can be organised with most important finding in the figure and text. Especially Fig2, Fig4 and Fig7. Fig5 does not give too much in terms of the real finding an in fact take away from the focus. Some parts of these figures can be simplified or moved to supplementary. Some of the figures in Fig2 and 7 are missing the scale bars.

We respectfully disagree with some conclusions made by the Reviewer. Our study contains ample material that is intended to guide the reader through the complexity of the Toxoplasma cell cycle and the intricate structures contained in the parasite. We have also introduced a few novel approaches that require additional schematics and dedicated discussions.

- **Fig 2.** *The G₂/M block, as well as the G₂ phase, had never been detected in apicomplexans. We created a new approach to determine the timing of the G₂/M checkpoint, which involves comparison to a known cell cycle block. Panels A, B, and C provide visuals and summarize our findings. The main events are highlighted with arrows (Panel C), while graphs (panel B) show differences in responses. The rest of the figure is devoted to quantification of the primary events caused by TgCrk4 deficiency, since the G₂ block had never been examined. While the U-ExM images of the entire vacuole (2-4 parasites) may seem overwhelming, they represent that the deficiency is consistent.*
- **Fig 7** *is devoted to the major Crk4/Cyc4 interactor TgiRD1 (former TgCdt1). This is one of the first mechanistic studies of central cell cycle regulators in Toxoplasma. This Cdt1-related protein was examined at the molecular level to support the main claims of its control of G₂ function. Nevertheless, we moved two panels from Fig. 7 into the supplement.*
- **Fig. 4** *is organized as follows. Top row: panels A, B visualize the G₂/M checkpoint block at the protein level. Middle row: panels C, D, and E represent the workflow to find*

TgCrk4 substrates. Bottom row: panels F, G highlight TgCrk4 substrates of interest that are discussed in the paper.

- *Fig. 5 is an in-depth analysis of the central cell cycle regulators across Apicomplexa phylum, a key figure of the study. Its comparative nature supports our main message: binary division is regulated by TgCrk4/TgCyc4, which are only expressed in a subgroup of apicomplexans that divide in a binary mode.*

4. May be bit more discussion of ORC in relation to their Cyclin-CRK complex as they did find upregulation of the ORC in their genome profiling. So may be instead of CDT1 these are more important in the licensing of DNA replication.

Our choice to focus on Cdt1-related protein was driven by the fact this protein is a major component of the TgCrk4/TgCyc4 complex, while the ORCs act downstream (as TgCrk4 substrates). Shifting focus to ORCs opens an entire new project, which will be explored in the future.

5 The model in Fig8B does not take Cyc4 into consideration and I feel is bit oversimplified as there are many factors that may be responsible for centrosome non separation. The S and G2 are not separated in the Cell cycle as given in this Fig.

Referring to comment 3, we focused on empirically supported, central findings and created the first model of centrosome cycle regulation in T. gondii. We intentionally drew focus to TgCrk4, which was extensively studied, while TgCyc4 received less attention due to difficulties in modulating its expression. We have used transcriptional downregulation to evaluate TgCyc4 (tet-OFF model), which is unfavorable for cell cycle studies because it exceeds the duration of the cell cycle. The unclear cell cycle borders are addressed in the introduction to this response. Briefly, the organization of apicomplexan cell cycle is currently unclear, thus most of the schematics are approximate.

6. It is not clear from the data with Cdt1 if this linking the inner and outer centrosome or its down regulation breaks the bipartite centrosome. May be some reflection it will be useful.

Our model suggests that both TgCrk4 and TgiRD1 (former TgCdt1) affect only the inner core of the centrosome, which we propose is comprised of two types of linkers. The arrows in Fig. 8 point specifically to the linkers whose stability depends on the expression of TgCrk4 or TgiRD1.

Minor comments

1. What is SAINT analysis as it is not described in methods.

We added the description of our SAINT analysis to M&M.

2. How was budding quantified

We supplemented the figure legend with the required information.

3. Western blot can have predicted size

Due to density of the figures, we did not supply the predicted MW of the proteins when they display the proper PAGE motility.

4. what does red star mean in Blot 1C

We added the description to the figure legend.

5. What does the number in Fig1H means please explain in the legends and same for Fig6F. In fig 1, removing the inhibition for 5 hours led to very less budding, but in fig 3, removing inhibition showed increased budding (50% in 2 hours). Please explain

Please see our response to the reviewer 1 minor comment regarding Fig. 1H and 6F.

We presume that there is some confusion regarding figure numbers. Perhaps the Reviewer refers to Fig. 2B. Indeed, the 4h block at G₂/M led to reduced budding (Fig. 2B), while release from the block for 2 hours (Fig. 3C, post-recovery) allows parasites to continue cell cycle progression and reach the next stage –budding. The numbers over the Fig. 3A, B, and C panels are from the plots in Fig. 2B to help give a comprehensive representation of the analyzed timepoint.

6. Fig2 has no scale bars -please add- this figure is too dense. May be fig2A, B,C can be in supplementary, legend in the figure can be in the figure legend.

Please see our response to comment 3. We have included scale bars.

7. Also this figure2 H and I in not quoted in line 231. Also this figure2 has no panel J but goes directly from I to K

The alphabetical order was corrected, and the reference added.

8. Fig3 the FigG can be more relevant in the Figure 8 while describing about the Crk4 and Cyc4 and CDt1 in binary mode of cell division. Also please define what stars mean either in legend or methods section in terms of significance.

Thank you for the suggestion. The Fig. 3G schematics summarize the overall findings of the Figure and acts as an intermediate conclusion in this study. We added the meaning of the stars in the M&M section.

Line 107 the sentence is incomplete

We have corrected the sentence.

Line 217 may be the figure could be referred as then it is not clear about the description.

Due to the density of the figures and well-established dynamics of the centrocone and basal rings, we included the reference to a publication rather than as a figure panel.

****Referees cross-commenting****

The study is quite rigorous and with analyses of CRK4-CYC4 and CDT. However it will be better if authors please revisit their conclusions on G2 phase of cell cycle in Toxoplasma based on their findings. The study will have important bearing on the community studying apicomplexan

Full Revision

parasites and DNA replication as well as who work on eukaryotic cell cycle.

Reviewer #3 (Significance (Required)):

Significance

In the manuscript by Hawkins et al have illustrated that in the apicomplexan parasite that have binary mode of cell division present a Cyclin-Crk complex with detailed analysis of Tg Crk4-Cyc4 that are novel in these group of parasite infect humans and animal alike like malaria parasite and ones affecting cattle and chicken. So these finding are novel as very little is known about this interaction. The significant finding is to show how the G2 phase of cell cycle may be regulated in these parasites and how DNA licencing factor Cdt1 is highly divergent but part of this CRK-Cyclin complex.

So though it discusses more on the Toxoplasma but it may be of interest to the scientist working on eukaryotes with divergent mode of cell cycle.

General Assessment - The findings are novel but the manuscript is too dense and at time loses the focus. May be both text and Figures could be made less dense so that important finding are revealed in better way.

Advance - It does give important insight into the cell cycle in apicomplexan parasite and how even though there are no cell cycle cyclin in Apicomplexa. The findings here suggest how different complexes can substitute for the function. It does extend the knowledge in the field of Cell division in divergent parasites both in terms of mechanistic, functional and technical way.

Dr. Elena Suvorova
University of South Florida
Department of Internal Medicine, Division of Infectious Diseases and International Medicine, Morsani College of Medicine
Tampa

28th Feb 2024

Re: EMBOJ-2024-116635
The G2 phase controls binary division of *Toxoplasma gondii*

Dear Dr. Suvorova,

Thank you for submitting your revised Review Commons manuscript for consideration by The EMBO Journal. In light of potential interest of the work, and the constructive transferred referee comments and your responses to them, I decided to treat the study like a regular revision, and returned it directly to the three original reviewers. After some delay, for which I apologize, we have in the meantime received their feedback, copied below for your information. Since two of the referees are now overall supportive, I would be interested in pursuing this work further for EMBO Journal publication. Nevertheless, especially referee 1 still retains several reservations, which would need to be clarified in an additional round of revision. I would therefore invite you to consider the included reports, and get back to me with a tentative point-by-point response within the next 1-2 weeks, so that we could discuss how the remaining issues might best be addressed at this stage.

In addition to the scientific points, please note that final revision for our journal will also require adjustments according to our journal's revisions guidelines copied below and in our online Guide to Authors. In particular:

- Please download and complete our author checklist (link provided below).
- Please upload the manuscript text as an editable DOCX file.
- On the abstract page of the manuscript, please include 4-5 general keyword terms to enhance searchability.
- Please adjust the format of the reference list and of the in-text citations according to EMBO Journal format (alphabetical order, author name et al + year, first up to 10 authors should be listed, followed by 'et al' ...). Please also adjust the format for citation of preprints as specified in our author guidelines. The citation in the text should be: "(preprint: NAME1 et al, YEAR)"; in the reference list: "Author NAME1, Author NAME2, ... (YEAR) article title. bioRxiv doi: XXX"
- Please include a dedicated "Data Availability" section at the end of the Material and Methods (suggested wording: "The [structural coordinates | microarray | mass spectrometry] data from this publication have been deposited to the [name of the database] database [URL] and assigned the identifier [accession | permalink | hashtag]."); should there no data deposition to public repositories linked to the study, this should still be stated as "This study includes no data deposited in external repositories."
- Please include a Disclosure and competing interests statement prior to the references - for details, see <https://www.embopress.org/competing-interests>
- "Supplementary Figures": These should either be uploaded as separate Expanded View figures (called: "Figure EV1/2/3...") with legends after the main figure legends; or collated in a single Appendix PDF (called: "Appendix Figure S1/2/3...") with each legend underneath the respective figure. An Appendix would further need to be headed by a brief Table of Contents.
- "Supplementary Tables": Simple ones (e.g. transgenic strains and primers) could become "Appendix Table S1/2/3...) included again in an Appendix PDF (and listed in its ToC); more complex ones may be more useful when uploaded as Expanded View XLSX spreadsheets, in which case they should be renamed to "Table EV1..." or -esp. when containing several tabs in the spreadsheet- to "Dataset EV1...". For the latter two types, the legend/header needs to be moved from the main text file and into a separate "Legend" tab of each spreadsheet.
- Please provide suggestions for a short 'blurb' text prefacing and summing up the conceptual aspect of the study in two sentences (max. 250 characters), followed by 3-5 one-sentence 'bullet points' with brief factual statements of key results of the paper; they will form the basis of an editor-written 'Synopsis' accompanying the online version of the article. Please also upload a synopsis image, which can be used as a "visual title" for the synopsis section of your paper. The image should be in PNG or JPG format, and please make sure that it remains in the modest dimensions of (exactly) 550 pixels wide and 300-600 pixels high.

- Finally, you shall also receive a separate message from our Source Data curation team, with instructions on how to prepare and upload relevant image and numerical raw data.

Thank you again for the opportunity to consider this work for The EMBO Journal. I look forward to hearing from you in due time.

Yours sincerely,

Hartmut Vodermaier

We realize that it is difficult to revise to a specific deadline. In the interest of protecting the conceptual advance provided by the work, we recommend a revision within 3 months (28th May 2024). Please discuss the revision progress ahead of this time with the editor if you require more time to complete the revisions. Use the link below to submit your revision:

Link Not Available

Referee #1:

General summary

The revised version of the manuscript is largely unchanged and while the authors provide additional explanation in the rebuttal, it is unclear if this information are also provided to the future reader.

(For example, response to initial comment: Lines 146-148: This statement is confusing in light of the expression data in Fig.1 F and H. If they stabilize each other, how is TgCrk4 stabilized in G1, when TgCyc4 is absent?)

Also, the figures are largely unchanged (except a smaller Fig. 7) and the point-by-point response leaves the impression that most comments and concerns are discussed away, rather than being used to improve the manuscript. This notion is further enhanced as I was not able to find a version of the revised manuscript where the changes were highlighted.

The general statement of the authors appears, in part, somewhat unscientific, i.e., to focus "on the processes that are associated with each cell cycle phase rather than on their temporal order". It is quite clear, also from studies on apicomplexan parasites (e.g., pmid: 35731838; pmid: 35353560) that the genome has to be replicated (S-phase) before nuclei can divide to segregate the genomes (G2/M). Hence, the temporal order does matter and the majority of mutant parasites should arrest at 2N if the developmental arrest of the mutants indeed occurs in G2 as claimed by the authors. Without a scientifically sound analysis of the DNA content (see initial comments and below), I do not see this manuscript fit for publication in The EMBO Journal.

Specific major concerns

As mentioned, the order of cell cycle events matters and, to me, the (unchanged) DNA content analysis is not convincing to support major claims of the manuscript. The authors do not show a wild type control (only arrested and released parasites). How can they claim that there is an enrichment of mutant parasites with re-replicated DNA? Especially since the culture is asynchronous as mentioned by the authors. Showing the DNA content of wild type parasites over time is critical to place the mutant data in context.

Also, the authors claim in the rebuttal that "removal of either TgCrk4 or TgiRD1 led to substantial decrease of the G1 population (reduction of 1N peak) accompanied by increase of parasites in the process of replication, completed replication (increase of 1.8 N peak)". Again, without an age-matched comparison to wild type, this is not supported.

The increase of parasites with > 1 N after the addition of IAA suggest that the parasites' development was indeed arrested prior to S-phase and not at G2/M. The now only 14/16, 14/11, and 6/6% of parasites in 3H/7A with higher DNA content are likely a consequence of the asynchronous culture as mentioned by the authors.

Thus, my initial concern remains that it seems more parsimonious to interpret the provided data as a DNA replication phenotype rather than a phenotype in G2. To address this, a scientifically sound DNA content analysis of wild type and mutant parasites is key.

Minor concerns

In addition, I still find it highly irritating that the panels of the figures are not in order of appearance in the text. And since schizogony is also part of the *T. gondii* life cycle (e.g., Antunes et al., 2023), an extended discussion of these data would improve the manuscript.

Referee #2:

general summary and opinion about the principal significance of the study, its questions and findings

This revised draft assesses how TgCrk4-TgCyc4 is important for binary cell division of the limited apicomplexans that progress through the cell cycle in this manner of endodyogeny. Despite the difficulties of differing and complex cell cycle modes in apicomplexans and low conservation of known cell cycle regulators, this manuscript identifies the TgCyc4 as the cyclin for TgCrk4 and how this interaction regulates *T. Gondii* G2 phase. Further, the newly re-named TgiRD1 is identified as an inhibitor of replication and a substrate of TgCrk4-TgCyc4.

- specific major concerns essential to be addressed to support the conclusions
none-remaining

- minor concerns that should be addressed

line 107-108- we suggest this wording: genotypically but *distantly* related to CDT1 and *phenotypically* related to Geminin as an inhibitor of replication.

Referee #3:

In the manuscript by Hawkins et al have illustrated detailed analysis of novel *Toxoplasma* CRK4-Cyclin4 complex that have

binary mode of cell division in apicomplexan group of parasites that infect humans and animal alike like malaria parasite and ones affecting cattle and chicken. So these finding are new as very little is known about this interaction. In the apicomplexan field it is generally shown that G2 phase of cell cycle is either absent or has very little role. The authors here demonstrate that the combination of Tg CRK4 and Tg Cyclin4 works during G2 phase of cell cycle such as chromosome rereplication and centrosome reduplication The significant finding is to show how the putative G2 phase of cell cycle may be regulated in these parasites and how DNA licencing factor Cdt1 is highly divergent but part of this CRK-Cyclin complex.

So though it discusses more on the Toxoplasma but it may be of interest to the scientist working on eukaryotes with divergent mode of cell cycle.

Advance - It does give important insight into the cell cycle in apicomplexan parasite and how even though there are no cell cycle cyclin in Apicomplexa. The findings here suggest how different complexes can substitute for the function. It does extend the knowledge in the field of Cell division in divergent parasites both in terms of mechanistic, functional and technical way.

In my previous review I had suggested that the manuscript is too dense but not much is taken on board. But I appreciate it is authors prerogative. However most other comments are addressed.

My suggestions and comments are as following:

1. In the present version is that authors have used different nomenclature for divergent CDT1 that they described. It will be better if they continue to use divergent CDT1 or CDT1 like rather than this terminology. Many proteins in Apicomplexa are divergent but if we keep changing terminology then it gets very confusing.

[EDITOR NOTE: SINCE IT IS NOT JUST FUNCTIONAL DIVERGENCE, BUT AS DISCUSSED BY REFEREE 2 AND IN THE PBP RESPONSE ALSO STRUCTURAL DIFFERENCES TO "CDT1", I AGREE WITH REFEREE 2 THAT THE NEW NAMING IS PREFERABLE IN THIS CASE]

2. In the discussion section authors use that they show "first time cyclin 4, first time G2 etc "and I feel this could be avoided or toned down as the cyclin4 was described much before Hawkins et al showed it as shown by Le Roch et al from Doerig group and Roques et al 2019 earlier discussed about this cyclin. These references were not quoted in the discussion or elsewhere.

3 I think the present title for the manuscript is bit too vague and it will be better if they use the title that they had used in their BioRxiv submission and since the G2 boundary are not clear in the manuscript and not proves with what factor really control in this study. It will better if they are bit careful and may be use "putative G2" rather being so forthright as G2. My suggestion below is based on their previous title of BioRxiv

"The CRK4-Cyc4 complex regulates putative G2 phase of binary division in Toxoplasma gondii"

[EDITOR NOTE: I AGREE WITH THE REFEREE THAT A MORE EXPLICIT TITLE IS PREFERABLE]

So overall the study is rigorously performed about three molecules of Cyclin4-CRK4-Cdt1 and their functional role is in the binary mode of cell division in Toxoplasma as well as suggesting the presence of G2 phase in apicomplexan parasites. The study will have important for the community studying apicomplexan parasites and divergent cell division as well as who work on eukaryotic cell cycle.

Rev_Com_number: RC-2023-02245

New_manu_number: EMBOJ-2024-116635

Corr_author: Suvorova

Title: The G2 phase controls binary division of Toxoplasma gondii

We would like to thank the editor for considering our work for EMBO Journal and our reviewers for overall support of the study. In this new response, we sought to clarify misunderstandings raised by reviewers 1 and 3 and made the appropriate adjustments. We included DNA analysis of the parental strain (new Fig. S2H) and proposed an alternative title for our study.

Referee #1:

General summary

The revised version of the manuscript is largely unchanged and while the authors provide additional explanation in the rebuttal, it is unclear if this information are also provided to the future reader.

(For example, response to initial comment: Lines 146-148: This statement is confusing in light of the expression data in Fig.1 F and H. If they stabilize each other, how is TgCrk4 stabilized in G₁, when TgCyc4 is absent?)

We apologize for the confusion. Because we did not explore the mechanics of the Crk4-Cyc4 complex, we reported the finding but refrained from including unsupported speculations in the text. However, the dynamic partnerships of Crk4 offers the most likely explanation: Crk4 interaction with Cyc4 is critical for G₂/M transition, but Crk4 interacts with many other molecules, the activities of some may be limited to G₁ phase. Such interaction may support Crk4 stability and likely keeps Crk4 inactivated. We believe that Fig. 1C data independently confirms the Crk4-Cyc4 complex, and it is a well-known phenomenon that degradation of one complex component often leads to the destabilization and/or degradation of the entire complex.

Also, the figures are largely unchanged (except a smaller Fig. 7) and the point-by-point response leaves the impression that most comments and concerns are discussed away, rather than being used to improve the manuscript. This notion is further enhanced as I was not able to find a version of the revised manuscript where the changes were highlighted.

A version of the manuscript with highlighted changes had not been requested. Our aim in addressing concerns in point-by-point responses was to provide our rationale and to help clarify any potential misunderstandings.

The general statement of the authors appears, in part, somewhat unscientific, i.e., to focus "on the processes that are associated with each cell cycle phase rather than on their temporal order". It is quite clear, also from studies on apicomplexan parasites (e.g., pmid: 35731838; pmid: 35353560) that the genome has to be replicated (S-phase) before nuclei can divide to segregate the genomes (G₂/M). Hence, the temporal order does matter and the majority of mutant parasites should arrest at 2N if the developmental arrest of the mutants indeed occurs in G₂ as claimed by the authors. Without a scientifically sound analysis of the DNA content (see initial comments and below), I do not see this manuscript fit for publication in The EMBO Journal.

We agree that the order of individual processes does matter, since DNA replication has to be completed before segregation. However, it does not apply to the cell cycle stage *per se*, which often incorporates multiple processes (e.g. DNA replication and parallel centrosome duplication). We will give two scenarios of possible overlaps of S/M and S/G₂ in *Toxoplasma*.

- S/M overlap: Mitosis has 4 subphases. The alignment of chromosomes prior to segregation takes place in metaphase, but the preceding prophase can run concurrently with DNA replication, because *Toxoplasma* does not have a chromatin condensation step and does not need to dissolve a nuclear membrane.
- G₂/M overlap: G₂ processes could also run concurrently with DNA replication. For example, spindle assembly initiated at G₂/M takes place in the centrocone, which is isolated from the chromatin compartment. In fact, spindle assembly initiation was reported in pre-mitotic parasites (tubulin accumulation in the centrocone, 26466679).

Lastly, the concept of concurrent cell cycle processes had been brought up by pioneers of cell cycle studies: "*Another peculiarity of budding yeast is that cells progress simultaneously through S and M phases (DNA synthesis, spindle formation, and chromosome alignment), without any noticeable condensation of chromosomes. In this case, completion of DNA synthesis is not required for the early events of M phase but is required for the metaphase–anaphase transition (Nasmyth, 1995).*" (from 10637314).

Specific major concerns

As mentioned, the order of cell cycle events matters and, to me, the (unchanged) DNA content analysis is not convincing to support major claims of the manuscript. The authors do not show a wild type control (only arrested and released parasites). How can they claim that there is an enrichment of mutant parasites with re-replicated DNA? Especially since the culture is asynchronous as mentioned by the authors. Showing the DNA content of wild type parasites over time is critical to place the mutant data in context.

Also, the authors claim in the rebuttal that "removal of either TgCrk4 or TgiRD1 led to substantial decrease of the G1 population (reduction of 1N peak) accompanied by increase of parasites in the process of replication, completed replication (increase of 1.8 N peak)". Again, without an age-matched comparison to wild type, this is not supported.

The temporal changes of DNA content in asynchronously growing *T. gondii* tachyzoites has been reported in multiple studies since the original work published by Dr. White group (11420103). However, we share the reviewer's concern and included the FACScan analysis of the parental strain (RH TIR1) after 0, 4, and 7h auxin treatment, which show negligible effects on DNA content. Due to the concern raised regarding the density of content in figures, these results are shown in Fig. EV2H and referenced in the text (lines 482-483).

The increase of parasites with > 1 N after the addition of IAA suggest that the parasites' development was indeed arrested prior to S-phase and not at G₂/M. The now only 14/16, 14/11, and 6/6% of parasites in 3H/7A with higher DNA content are likely a consequence of the asynchronous culture as mentioned by the authors.

Unfortunately, we are finding it difficult to understand this conclusion and can only offer our reasoning in hopes it can provide clarity. Arresting the cell cycle prior to S-phase should lead to an increase of 1N populations (DNA is not replicated). On the contrary, our data (for example, Fig. 3H) shows an increase of S-phase (1-1.8N: 16/20/24), G₂ and M (1.8N: 14/14/27) and re-replicated (>2N: 6/6/8) populations. At the same time, Crk4-deficient parasites were neither able to complete cell division nor initiate a new round of DNA replication, as reflected in 1N decrease (63/58/40). A similar dynamic applies to iRD1, as shown in Fig. 7H. These data align well with a post-G₁ block: Detecting re-replication (>2N increase) indicates a malfunction in G₂, since this process should be under the control of G₂ machinery.

Thus, my initial concern remains that it seems more parsimonious to interpret the provided data as a DNA replication phenotype rather than a phenotype in G₂. To address this, a scientifically sound DNA content analysis of wild type and mutant parasites is key.

We would like to assure the reviewer that we did not intend to refrain from performing or showing our analyses of DNA replication. Several lines of evidence have contributed to our final conclusion where we identified the G₂/M block in *Toxoplasma* endodyogeny, and FACScan DNA analyses being one of them. Specifically, FACScan detected DNA rereplication, supporting centrosome reduplication (quantitative IFA and ultra-expansion microscopy) and the control of spindle extension (ultra-expansion microscopy). The identity of G₂ was further elucidated by discovery of the Crk4 network (phospho-proteome), global proteome of Crk4-dependent arrest, and the positioning of G₂/M upstream of SAC (block-release synchronization). We collected FACScan data in three or more independent experiments, showed quantifications of the gated cell cycle populations, and now provided parallel analyses of the parental strain. The latter should provide scientifically sound DNA content analysis, and together with the other reported findings, contribute to the proof of a functional G₂/M checkpoint in *Toxoplasma*.

Minor concerns

In addition, I still find it highly irritating that the panels of the figures are not in order of appearance in the text. And since schizogony is also part of the *T. gondii* life cycle (e.g., Antunes et al., 2023), an extended discussion of these data would improve the manuscript.

We believe that the reviewer is referring to limited endopolygeny (as opposed to schizogony, since *Toxoplasma* forms internal buds, 32582569), which *Toxoplasma* uses to replicate in cat's intestine. Considering the broader readership audience of EMBO J, we opted to use terms that are more commonly accepted across cell cycle studies (binary and multinuclear division) rather than terms that are more specialized and limited to the parasitology field (endodyogeny, endopolygeny and schizogony). Endodyogeny is binary division, while endopolygeny/schizogony are multinuclear divisions. Our view of the role of G₂ in multinuclear divisions can be found in the last paragraph of the discussion.

Referee #2:

general summary and opinion about the principal significance of the study, its questions and findings

This revised draft assesses how TgCrk4-TgCyc4 is important for binary cell division of the limited apicomplexans that progress through the cell cycle in this manner of endodyogeny. Despite the difficulties of differing and complex cell cycle modes in apicomplexans and low conservation of known cell cycle regulators, this manuscript identifies the TgCyc4 as the cyclin for TgCrk4 and how this interaction regulates *T. Gondii* G₂ phase. Further, the newly re-named TgiRD1 is identified as an inhibitor of replication and a substrate of TgCrk4-TgCyc4.

- specific major concerns essential to be addressed to support the conclusions
none-remaining

- minor concerns that should be addressed

line 107-108- we suggest this wording: genotypically but *distantly* related to CDT1 and *phenotypically* related to Geminin as an inhibitor of replication.

Thank you, we have followed this suggestion and modified that sentence.

Referee #3:

In the manuscript by Hawkins et al have illustrated detailed analysis of novel Toxoplasma CRK4-Cyclin4 complex that have binary mode of cell division in apicomplexan group of parasites that infect humans and animal alike like malaria parasite and ones affecting cattle and chicken. So these finding are new as very little is known about this interaction. In the apicomplexan field it is generally shown that G2 phase of cell cycle is either absent or has very little role. The authors here demonstrate that the combination of Tg CRK4 and Tg Cyclin4 works during G2 phase of cell cycle such as chromosome rereplication and centrosome reduplication. The significant finding is to show how the putative G2 phase of cell cycle may be regulated in these parasites and how DNA licencing factor Cdt1 is highly divergent but part of this CRK-Cyclin complex.

So though it discusses more on the Toxoplasma but it may be of interest to the scientist working on eukaryotes with divergent mode of cell cycle.

Advance - It does give important insight into the cell cycle in apicomplexan parasite and how even though there are no cell cycle cyclin in Apicomplexa. The findings here suggest how different complexes can substitute for the function. It does extend the knowledge in the field of Cell division in divergent parasites both in terms of mechanistic, functional and technical way.

In my previous review I had suggested that the manuscript is too dense but not much is taken on board. But I appreciate it is authors prerogative. However most other comments are addressed.

My suggestions and comments are as following:

1. In the present version is that authors have used different nomenclature for divergent CDT1 that they described. It will be better if they continue to use divergent CDT1 or CDT1 like rather than this terminology. Many proteins in Apicomplexa are divergent but if we keep changing terminology then it gets very confusing.

[EDITOR NOTE: SINCE IT IS NOT JUST FUNCTIONAL DIVERGENCE, BUT AS DISCUSSED BY REFEREE 2 AND IN THE PBP RESPONSE ALSO STRUCTURAL DIFFERENCES TO "CDT1", I AGREE WITH REFEREE 2 THAT THE NEW NAMING IS PREFERABLE IN THIS CASE]

Resolved.

2. In the discussion section authors use that they show "first time cyclin 4, first time G2 etc "and I feel this could be avoided or toned down as the cyclin4 was described much before Hawkins et al showed it as shown by Le Roch et al from Doerig group and Roques et al 2019 earlier discussed about this cyclin. These references were not quoted in the discussion or elsewhere.

We would like to clarify the apicomplexan cyclins' nomenclature; *Plasmodium* cyclin 4 is not related to the *Toxoplasma* Cyc4 that had not been annotated in *Toxoplasma* genomes prior to 2020. As we mentioned in our previous rebuttal, apicomplexan genomes undergo constant refinement and revised annotations are available a few times a year (VEupathDB). For this reason, we performed a new phylogenetic analysis of *Toxoplasma* cyclins (5 known and 5 newly annotated) in 2022 (Hawkins et al, 2022). TgCyc4 was among previously unidentified cyclins (indicated in the current manuscript, lines 128-132) and had not been examined in the previous studies. *Plasmodium* cyclins had not been re-evaluated since 2013.

3 I think the present title for the manuscript is bit too vague and it will be better if they use the title that they had used in their BioRxiv submission and since the G2 boundary are not clear in the manuscript and not proves with what factor really control in this study. It will better if they are bit careful and may be use "putative G2" rather being so forthright as G2. My suggestion below is based on their previous title of BioRxiv

"The CRK4-Cyc4 complex regulates putative G2 phase of binary division in *Toxoplasma gondii*"

[EDITOR NOTE: I AGREE WITH THE REFEREE THAT A MORE EXPLICIT TITLE IS PREFERABLE]

We would like to maintain the certainty of a G₂ phase in *Toxoplasma*, as we had shown the roles of Crk4 and iRD1 in centrosome re-duplication, DNA re-replication, and spindle formation, all conventional G₂ processes. Furthermore, we determined that the Crk4 block functions upstream of the spindle assembly checkpoint. The fact that Apicomplexan cell cycle phases can overlap (please see reviewer 1 comments) makes the identification of G₂ onset an impossible task. Thus, we suggest a compromise to instead focus on the G₂/M checkpoint. We are proposing a new title: "The Crk4-Cyc4 complex regulates G₂/M transition in *Toxoplasma gondii*".

So overall the study is rigorously performed about three molecules of Cyclin4-CRK4-Cdt1 and their functional role is in the binary mode of cell division in *Toxoplasma* as well as suggesting the presence of G2 phase in apicomplexan parasites.

The study will have important for the community studying apicomplexan parasites and divergent cell division as well as who work on eukaryotic cell cycle.

Dr. Elena Suvorova
University of South Florida
Department of Internal Medicine, Division of Infectious Diseases and International Medicine, Morsani College of Medicine
Tampa

26th Mar 2024

Re: EMBOJ-2024-116635R
The Crk4-Cyc4 complex regulates G2/M transition in *Toxoplasma gondii*.

Dear Dr. Suvorova,

Thank you for submitting your final revised manuscript for our consideration. I am pleased to inform you that we have now accepted it for publication in The EMBO Journal.

Yours sincerely,

Hartmut Vodermaier
